# Can riparian vegetation shade mitigate the expected rise in stream temperatures due to climate change during heat waves in a human impacted pre-alpine river?

Heidelinde Trimmel[1], Philipp Weihs[1], David Leidinger[1], Herbert Formayer[1], Gerda Kalny[2], Andreas Melcher[3]

[1]Institute of Meteorology, University of Natural Resources and Life Science (BOKU), Vienna, 1190, Vienna, Austria
[2]Institute of Soil Bioengineering and Landscape Construction (IBLB), University of Natural Resources and Life Science (BOKU), Vienna, 1190, Austria
[3]Institute of Hydrobiology and Aquatic Ecosystem Management (IHG), University of Natural Resources and Life Science (BOKU), Vienna, 1190, Vienna, Austria

*Correspondence to*: Heidelinde Trimmel (heidelinde.trimmel@boku.ac.at)

**Abstract.** Global warming has already affected European rivers and their aquatic biota, and climate models predict an increase of temperature in Central Europe over all seasons. We simulated the influence of expected changes in heat wave intensity during the 21st century on water temperatures of a heavily impacted pre-alpine Austrian river and analysed future mitigating effects of riparian vegetation shade on radiant and turbulent energy fluxes using the deterministic model *Heat Source*. Modelled stream water temperature increased less than 1.5 °C within the first half of the century. Until 2100 a more significant increase of around 3 °C in minimum, maximum and mean stream temperature was predicted for a 20 year return period heat event. The result showed clearly that in a highly altered river system riparian vegetation was not able to fully mitigate the predicted temperature rise caused by climate change, but would be able to reduce water temperature by 1 to 2 °C. The removal of riparian vegetation amplified stream temperature increases. Maximum stream temperatures could increase by more than 4 °C even in annual heat events. Such a dramatic water temperature shift of some degrees, especially in summer, would indicate a total shift of aquatic biodiversity. The results demonstrate that effective river restoration and mitigation requires re-establishing riparian vegetation and emphasizes the importance of land-water interfaces and their ecological functioning in aquatic environments.

**Keywords:** stream temperature, modelling, riparian vegetation, shade, climate change

## 1 Introduction

Stream temperature is an important factor influencing the physical, chemical and biological properties of rivers and thus the habitat use of aquatic organisms (Davies-Colley and Quinn 1998; Heino et al. 2009; Magnuson et al. 1979). Heino et al. (2009) suggest that freshwater biodiversity is highly vulnerable to climate change with extinction rates exceeding those of terrestrial taxa. Stream temperature is highly correlated with the assemblages of fish and benthic invertebrates along the river course (Dossi et al. 2015; Melcher et al. 2015). The duration and magnitude of the maximum summer stream temperatures in particular are limiting factors for the occurrence of fish many species. High temperatures may produce high

physiological demands and stress while also reducing the oxygen saturation in the water column. The increased metabolic requirements together with the decreased oxygen availability can prove to be a limiting or even be lethal in combination; the average optimum temperature for cold water species is below 16 °C (Matulla et al. 2007; Pletterbauer et al. 2015).

Continuous warming of water temperatures induces changes from cold water to warm water fish species assemblages and slow altitudinal shifts of species, if the habitat is suitable and no migration barriers exist. River continuum disruption and river dimension reduce the fish zone extent significantly (Matulla et al. 2007; Bloisa et al. 2013). Extreme events where lethal thresholds of stream temperature are exceeded can cause a disruption of animal communities or even extinction of (cold water) species (Melcher et al. 2013; Pletterbauer et al. 2015). The largest uncertainties in forecasts of total suitable habitat are climate uncertainty (Wenger et al. 2013). All 230 stations of the Austrian hydrographic central office, with different elevations, distances from source and catchment areas recorded increases in stream temperature of 1.5 °C during summer (Jun - Aug)  and 0.7 °C during winter (Dec – Feb) between 1980 and 2011 (0.48 °C / decade) (BMLFUW 2011). This change is not likely to be due to natural climatic cycles, but is part of a long term trend caused by anthropogenic changes in the atmosphere (APCC 2014).

Air temperatures have been rising and are expected to continue to rise globally within the next century (IPCC 2013). In eastern Austria mean air temperature has risen by 2 °C since 1880, which is more than double the 0.85 °C rise recorded globally (Auer et al. 2014). A further temperature increase within the 21st century is very likely (APCC 2014). If emission scenario A1B is assumed, mean air temperature increases of 3.5 °C over the level of the reference period 1961-1990 by the end of the 21$^{st}$ century are expected in Ausria (APCC 2014; Gobiet et al. 2014).

Temperature extremes have changed markedly and extreme high temperature events i.e. heat waves are very likely to increase in the 21st century (APCC 2014). Soil temperature is also expected to increase due to climate change and will influence stream temperatures via substrate heat conduction and groundwater flux (Kurylyk et al. 2015). For example, in Austria, near surface groundwater body temperature is expected to rise by 0.5 to 1 °C on average by 2050 (BMLFUW 2011). Austria lies between two zones of opposing precipitation trends (IPCC 2013). Northern Europe shows an increasing trend, while the Mediterranean has a decreasing trend (Böhm 2006). In southeastern Austria a precipitation decrease of about 10–15 % has been recorded over the last 150 years (APCC 2014; Böhm 2012). Low flow discharge rates of rivers are likely to decrease by 10 to 15 % by 2021–2050 compared to 1976–2007 during all seasons (Nachtnebel et al. 2014; Mader et al. 1996; APCC 2014).

 For the study region during summer heat waves neither change in groundwater nor snow melt contributions are expected (APCC 2014). Heavy and extreme precipitation shows no clear increasing signal on average, but it is likely to increase from October to March (APCC 2014). No clear trend of increasing wind speed (Matulla et al. 2008; Beniston 2007) or increase in sunshine hours (Ahrens et al. 2014) has been detected but changes in the climate system may also include changes in those parameters (APCC 2014).

Stream temperature is controlled by advection of heat, dispersion and the net energy fluxes acting on the surface and river bed. Net short wave radiation is the dominant energy input causing diurnal and seasonal water temperature variability. Long wave radiation flux (Benyaha et al. 2012) as well as the turbulent fluxes evaporation and convection, which are controlled by air temperature, vapour pressure, wind speed and net radiation, play an important role (Caissie et al. 2007; Garner et al 2014; Hannah et al. 2008; Johnson 2004).

One of the most influential factors regulating stream temperature is riparian vegetation (Caissie 2006; Groom et al. 2011; Johnson 2004; Moore et al. 2005; Rutherford et al. 1997). The streamside vegetation buffer width (Clark et al. 1999), vegetation density and average tree height all have a strong influence on stream temperature (Sridhar et al. 2004). Vegetation affects the sky view of the river and thereby short (Holzapfel et al. 2013) and long wave radiation flux, evaporation and convection heat flux, which are highly correlated to the openness of the sky.  The reduction of short wave radiation can contribute significantly to reducing the heating of rivers during warmer summers (Sinokrot and Stefan 1993; Parker and Krenkel 1969; Rutherford et al. 1997).

There are different approaches to predicting stream temperature. Water temperature can be predicted using statistical functions (stochastic models) and its correlation (regression models) to known variables (e.g. air temperature, water temperature of the previous days or streamflow). Use of air temperature as a surrogate for future water temperature can lead to errors when linear (Erickson and Stefan 2000; Webb and Nobilis 1997) or non-linear (Mohseni et al. 1998) regression models are applied (Arismendi et al. 2014). Stochastic models used to determine the long term annual component of temperatures and their short term residuals separately yield good results (Caissie et al. 2001). Including a discharge term in the regression model can improve the model's performance during heat wave and drought (low flow) conditions, when water temperatures are most sensitive to air temperature (van Vliet et al. 2011).  Energy balance models resolving all energy fluxes affecting a river system are the best suited to predict stream temperature (Caissie et al. 2007) but demand the most input data. Only these models are able to simulate energy flux changes caused by increased or decreased river shade.

Though the influence of vegetation on water temperature is evident, its ability to mitigate climate change is not yet sufficiently understood. Latent and sensible heat flux as well as long wave radiation balance are non-linearly dependent on air temperature. It is not obvious whether the same level of shade will always lead to the same rate of heat reduction. Shading caused by tall but less dense trees may allow exchange of air, while lower riparian vegetation may cause the same level of shade but would reduce air movement. Vegetation can reduce warming but may also reduce nightly cooling by altering the energy fluxes on a local scale, which can only be modelled using deterministic methods.

The conclusion may be drawn that many studies have already addressed the influence of riparian vegetation on stream water temperature using field measurements. Other studies used different methods to make short-term forecasts of stream temperature and few tried to answer the question on how climate change might increase stream water temperature. One result or trend may however not be transferred from one river to another. Particular statements about the riparian vegetation's potential to mitigate the influence of climate change are only reliably valid for a given type of stream with its unique combination of morphologic and hydrologic parameters, local climate (Sinokrot and Stefan 1993;  Johnson 2003; Steel and Fullerton 2017) and regional climate change (Johnson and Wilby 2015). Air temperature was normally used as a surrogate for stream temperature and energy flux variations in different river sections were not considered. The novel aspect of the present study is to investigate the influence of climate change and of riparian vegetation on the same river and attempt to make a realistic forecast of the riparian vegetation's potential to mitigate climate change in a specific river using a deterministic model.

The aims of the present study are therefore (1) to estimate the magnitude of stream temperature rise during extreme heat events caused by the expected rise in air temperature by the end of this century and (2) to investigate the ability of riparian

vegetation to mitigate the expected water temperature rise within the habitat optimum of the site specific aquatic fauna and (3) to analyse the possible variation of vegetation and potential interaction of vegetation and discharge with respect to climate change and their impact on water temperature.

## 2 Methods

Stream temperature was simulated with the 1D energy balance and hydraulic model *Heat Source* (Boyd and Kasper 2003) for 51 km along a section of river including upstream forested regions and tributaries. Temperature was simulated for each 500m section of the river, which amounted to a total of 103 sites. First the longitudinal changes of energy fluxes were analysed during the maximum heat wave, which took place in eastern Austria during summer 2013. Future heat wave episodes that are likely to occur during the climate periods 2016-2045, 2036-1065 and 2071-2100 in the study region, were

selected. Regional climate scenarios produced by the ENSEMBLE project (Hewitt et al. 2004) were further processed and the meteorological data extracted. The future upstream model water temperature was simulated by the methodology of Caissie et al. (2001). *Heat Source* was used to simulate the stream temperature of the River Pinka for 12 future episodes and eight vegetation scenarios.

### 2.1 Study region

The River Pinka originates at 1480 meters above sea level (m.a.s.l.) in the eastern Austrian Alps and discharges about 100 km downstream at 200 m.a.s.l. into the River Raab. The catchment of the Pinka is 664 km². According to Muhar et al (2004), who categorized all Austrian rivers with catchment areas > 500 km² corresponding to their annual discharge, the Pinka falls in the smallest of the five categories with $0 – 5$ m³ s$^{-1}$ mean annual discharge. The study region covers a 51 km stretch of the river Pinka from distance from source (DFS) 11 km (559 m.a.s.l) near its most upstream gauge in Pinggau to

DFS 62 km (240 m.a.s.l.) close to the gauge at Burg (Fig. 1). For the first 10km the river has a slope of 0.017 m m$^{-1}$ whereas in the remaining section the slope is only 0.004 m m$^{-1}$. The river bankfull width varied from 4 to 10 m (Fig. 2c). The maximum depth of the different river sections varied between 0.1 and 0.5 m and was 0.17 m on average. Only 4 % of the reaches presently fall into into the most natural or the second category according to Ledochowski (2014) (Fig. 2c). On the other hand, 60 % of reaches are classed as continuously influenced with no or very few natural sections (Fig. 2c).

Close to the source (DFS 0-12.5) the vegetation consists of commercial spruce forests (*Picea abies*) which undergo management. In the middle and downstream sections of the river, the near-natural deciduous riparian vegetation includes typical floodplain species of the region (willows (*Salix* sp.) and alders *Alnus glutinosa* and *incana*). In the downstream 80 % of the river (from DFS 34 to 61), riparian vegetation is reduced to one- or two-sided sparse tree plantations lining the river course for decorative purposes. These areas are mowed on a regular basis to prevent scrub growth. Other frequent trees like

ash (*Fraxinus excelsior*), hazel (*Corylus avellana*), wild cherry (*Prunus avium*) and Elder (*Sambucus nigra*) can be found along the whole river course. In this region air temperature has risen by 2 °C since 1880 (Auer et al. 2014). Precipitation has declined by 10-15 % in our study region, the largest reduction in precipitation in Austria (Auer et al. 2007; Böhm et al. 2009; Böhm et al. 2012).

**Potential changes in vegetation cover**

Changes in vegetation height and density in floodplain forests in natural systems are mainly due to succession (Primack 2000; Garssen et al. 2014; Rivaes et al. 2014 ). The present potential natural floodplain forest is in many areas reduced to narrow fringes accompanying the river, which are flooded at least annually. The river has been continuously straightened and regulated throughout the 20th century. Flood protection measures and land use pressure has further altered the river and riparian vegetation dynamics. The vegetation behind these fringes is in the transition zone between softwood and hardwood wetland and a further change towards upland or zonal vegetation is expected via terrestrialization processes, well known in the Danube region (Birkel and Mayer 1992; Egger et al. 2007). The dominant tree species present along the River Pinka, *Salix alba, Alnus glutinosa* and *Fraxinus excelsior* have a European-wide distribution (San-Miguel-Ayanz et al. 2016) so they are likely to defend their habitat. Some autochthonous species (*Populus alba, Prunus avium, Salix caprea, Fraxinus excelsior, Carpinus betulus*) which were present in 2013 are favoured by warmer climates (Kiermeyer 1995; Roloff and Bärtels 2006). Non-native species like *Robinia pseudoacacia* and *Acer negundo* are already present in the study region and might enlarge their habitat at the expense of native species (Kiermeyer 1995; Roloff and Bärtels 2006). Changes in tree species in favour of warmth-loving plants from downstream regions of the Raab/Danube catchment are possible (Lexer et al. 2014). Generally changes are likely to be not only driven by climatic but also anthropogenic factors as plantation of foreign species, which is not foreseeable.

**2.2 Modelling vegetation influence on energy fluxes and stream temperature along the river**

Using the deterministic model *Heat Source* version 9 (Boyd and Kasper, 2003; Garner 2007) the energy fluxes, hydraulics and stream temperature were simulated along the River Pinka. The generation of the input  data sets is described in the following section 2.3 below. Vegetation affects water temperature directly by reducing short wave radiation input and reducing the view to sky which affects long wave radiation balance and the turbulent heat fluxes. Long wave radiation and the turbulent heat fluxes are non-linearly dependent on air temperature. Short and long wave energy flux, latent and sensible heat flux and conduction are taken into account:

$$\Phi_{Total} = \Phi_{Latent} - \Phi_{Sensible} - \Phi_{Longwave} - \Phi_{Solar} - \Phi_{Conduction} \quad (1)$$

where $\Phi_{Total}$ is the energy balance, $\Phi_{Latent}$ the latent heat flux, $\Phi_{Sensible}$ the sensible flux and $\Phi_{Longwave}$ is the long wave radiation balance, all of which refer to the stream surface. $\Phi_{Solar}$ is the short wave energy which is absorbed by the water column and $\Phi_{Conduction}$ is the conduction flux to the stream bed.

**Short wave radiation**

The amount of radiation entering the stream $\Phi_{SolarEnter}$ is the radiation unobstructed by shading $\Phi_{AboveTopo}$ reduced by topographic shade $\Phi_{TopoShade}$, bank shade $\Phi_{BankShade}$, vegetation shade $\Phi_{VegShade}$ and reflected from river surface $\Phi_{SolarRef}$.

$$\Phi_{SolarEnter} = \Phi_{AboveTopo} - \Phi_{TopoShade} - \Phi_{BankShade} - \Phi_{VegShade} - \Phi_{SolarRef} \quad (2)$$

If topographic or bank shade is present, the direct radiation fraction is reduced by the radiation entering in the affected angles. If vegetation shade is present the direct radiation is reduced dependent on the vegetation density using a formulation of Beer's law by the term $\Phi_{SolarExtinct}$.

$$RE = -\log\left(\frac{1-VD}{10}\right) \quad (3) \qquad \Phi_{\text{SolarExtinct}} = 1 - \exp\left(-RE\left(\frac{LD}{\cos\left(\text{rad}\left(\theta_s\right)\right)}\right)\right) \quad (4)$$

Where $RE$ is the riparian extinction, $VD$ is vegetation density, $LD$ is the distance from the river centre and $\theta_s$ is the solar elevation angle. $\Phi_{\text{Solar}}$ which is finally absorbed by the water column is the amount of solar radiation entering the stream $\Phi_{\text{SolarEnter}}$ (2) minus the amount that is absorbed in the river bed $\Phi_{\text{SolarAbsob}}$ and reflected $\Phi_{\text{SolarBedRef}}$.

$$\Phi_{\text{Solar}} = \Phi_{\text{SolarEnter}} - \Phi_{\text{SolarAbsorb}} - \Phi_{\text{SolarBedRefl}} \quad (5)$$

**VTS and long wave radiation balance**

The view to sky VTS is calculated using modified vegetation density $VD_{\text{mod}}$ and the vegetation angle $\theta_v$. VTS is used to calculate the diffuse radiation below vegetation height, atmospheric longwave radiation $\Phi_{\text{LongwaveAtm}}$(7), longwave radiation emitted from vegetation $\Phi_{\text{LongwaveVeg}}$ (6) and the reduction of wind speed at the river surface (11).

$$\text{VTS} = 1 - \frac{\max\theta_v * VD_{\text{mod}}}{7*90} \quad (6)$$

Longwave radiation balance $\Phi_{\text{Longwave}}$ is the sum of all long wave components:

$$\Phi_{\text{LongwaveAtm}} = 0.96 * \text{VTS} * em * \sigma * \left(T_{\text{airK}}\right)^4 \quad (7)$$

$$\Phi_{\text{LongwaveVeg}} = 0.96 * (1\text{-VTS}) * 0.96 * \sigma * \left(T_{\text{airK}}\right)^4 \quad (8)$$

$$\Phi_{\text{LongwaveStream}} = -0.96 * \sigma * \left(T_{\text{prevK}}\right)^4 \quad (9)$$

$$\Phi_{\text{Longwave}} = \Phi_{\text{LongwaveAtm}} + \Phi_{\text{LongwaveVeg}} + \Phi_{\text{LongwaveSream}} \quad (10)$$

where $em$ is the emissivity of the atmosphere, $\sigma$ the Stefan Bolzmann constant and $T_{\text{airK}}$ the air temperature and $T_{\text{prevK}}$ the stream temperature of the advected water in degree Kelvin.

**Latent and sensible heat flux**

Latent heat flux $\Phi_{\text{Latent}}$ was calculated using the Penman method, which included the radiation balance:

$$E_a = 1.51\text{E-9} + 1.6\text{E-9} * \left(w * \text{VTS}\right) * \left(e_s - e_a\right) \quad (11)$$

$$E = \frac{\left(\left(\frac{\Phi_{\text{Rad}} * \Delta}{\left(\rho * LHV\right)}\right) + E_a * \gamma\right)}{\left(\Delta + \gamma\right)} \quad (12), \qquad \Phi_{\text{Latent}} = -E * LHV * \rho \quad (13)$$

where $E_a$ is the aerodynamic evaporation, $w$ the wind speed [ms$^{-1}$], $E$ is the evaporation rate [ms$^{-1}$], $\Phi_{\text{Rad}}$ the sum of $\Phi_{\text{Longwave}}$ and $\Phi_{\text{Solar\_enter}}$, $\Delta$ the slope of the saturation vapour vs. air temperature curve, $\rho$ is the density of water [kg m$^{-3}$], $LHV$ the latent heat of vaporization [Jkg$^{-1}$ ] and $\gamma$ is the psychrometric constant [mb°C$^{-1}$].

Sensible heat flux is calculated from evaporation via the Bowen ratio $\beta$:

$$\beta = \frac{\gamma * (T_{prev} - T_{air})}{(e_s - e_a)} \quad (14); \quad \Phi_{Sensible} = \Phi_{Latent} * \beta \quad (15)$$

where $T_{prev}$ is the stream temperature, $T_{air}$ is air temperature, $e_s$ is the saturated vapor pressure and $e_a$ the air vapor pressure.

**Conduction heat flux**

Conduction $\Phi_{Conduction}$ is dependent on the thermal conductivity of the sediment $TC_{sed}$, the sediment depth $d_{sed}$ and sediment temperature $T_{sed}$ and water temperature $T_{prev}$ :

$$\Phi_{Conduction} = \frac{TC_{sed} * (T_{sed} - T_{prev})}{(d_{sed}/2)} \quad (16)$$

**Water temperature**

The effect of the energy balance of the water column on stream temperature was calculated taking into account flow velocity and river morphology. The stream temperature increase ΔT caused by $\Phi_{Total}$ (1) was calculated using:

$$\Delta T = \frac{\Phi_{Total} * dt}{\left(\frac{A}{W_w}\right) * c_{H2O} * m} \quad (17)$$

where $A$ is the cross sectional area of the river, $W_w$ is the wetted width, the $c_{H2O}$ is the specific heat capacity of water (4182 J kg$^{-1}$ C$^{-1}$), $m$ the mass of 1 m³ water which is 998.2 kg.

Conclusively *Heat Source* includes all aspect of vegetation changes on stream temperature during future episodes and the main processes needed to answer the research questions can be modelled with *Heat Source*.

A first model set up and validation for usage at the River Pinka during heat wave conditions was done by Trimmel et al. (2016). By fine-tuning the morphological input (bottom width, roughness parameter Manning's n and sediment hyporheic thickness) and the wind parameterisation, the model's validity could be considerably improved for the simulations used here. Tuning increased the coefficients of determination R² for water temperature stations of different vegetation height and density at DFS 31, 35, 37, 39 and 48 km to 0.96–0.98 (daily minimum), 0.96–0.99 (daily mean) and 0.94–0.98 (daily maximum). The measurements fitted the simulation very well (hourly RMSE was 0.88 °C averaged for all stream measurement stations) so we concluded that all assumptions were met and the model was appropriate to be used for predictions.

**2.3 Preparation of input**

**2.3.1 Meteorological input**

During the maximum heat wave event of 2013, field measurements were collected at the study site. Global radiation, air temperature, air humidity and wind speed were measured at a reference station located at DFS 39 km 47° 16' 11.055" N 16° 13' 47.892" E, 300 m.a.s.l. To link the measured microscale meteorological data to topological scale meteorological data a systematic intercomparison between the local meteorological stations of the Austrian Weather Service (ZAMG) and the 1x1 km gridded observational data set INCA (Haiden et al., 2011) was made. Since the local permanent meteorological stations

of ZAMG were used to produce the gridded INCA data set, they are highly consistent. The comparison of the INCA data with the air temperature measured at our reference station close to the river showed an RMSE of 0.67°C and an $R^2$ of 0.99 for consecutive hourly measurements during summer half-year 2013 (1 Apr – 30 Sept). So the INCA data set was used as a proxy to represent the local meteorological conditions within the catchment.

To obtain future meteorological conditions at the reference station, data were extracted from the regional climate models (RCM) Aladin (driven by the global climate model ARPEGE (Déqué et al., 1994)), Remo, and RegCM3 (both ECHAM 5 driven (Roeckner et al., 2003, 2004)). The aim was to estimate possible maximum temperature values; therefore, data from Aladin, the climate model with the most extreme dry and hot summers, were selected. The RCMs were bias-corrected using the quantile mapping technique (Déqué 2007) based on the E-OBS data set (Haylock et al., 2008) and scaled. In a second

step the data were spatially localized to a 1 km x 1 km grid encompassing the area under investigation using the Austrian INCA data set (Haiden et al. 2011). In a third step the data were temporally disaggregated from a resolution of one day to one hour. Temperature was disaggregated based on the daily maximum and minimum temperatures using three piecewise continuous cosine curves (Koutsoyiannis 2003; Goler & Formayer 2012). The temperature data were elevation corrected with a lapse rate of 0.65 °C per 100 m.

**Selection of extreme heat events**

The period chosen as past reference period ("OBS") was an extreme heat wave that ran from 4 – 8 August 2013, which was the most intense heat wave of 2013. The mean air temperature of this episode was comparable to a 20 year return period 5 day event (Table 1) for the period 1981–2010. Future episodes were selected by choosing future heat wave events in three periods (2016–2045: "2030", 2036–2065: "2050", 2071–2100: "2085") in the summer months (June–August) that were

simulated for the emission scenario A1B by the climate model Aladin (Radu et al. 2008). The events were chosen by selecting periods when the 5 day mean air temperature exceeded different thresholds using the percentiles of the 5 day mean air temperature of the three periods, which corresponded to an event with a 1 year (1a), 5 year (5a) or 20 year (20a) return period as well as the maximum heat wave event of the period (Max). The selection criteria are shown in Table 1. The start was 14 days prior to the end of the episode to allow spin-up of the *Heat Source* model, so all episodes have equal length of

14 days.

**2.3.2 Vegetation and morphology**

The riparian vegetation cover and river morphology of this region was investigated by Ledochowski (2014). First, aerial photographs were used to define the river centre line and a 50 m buffer on both sides, because the influence of riparian vegetation on the river is negligible beyond this point. Within this zone, areas of homogeneous structure, land use and

ecological function were mapped by hand. Additional information such as height, density and dominant vegetation type were recorded as attributes of mapped features on. To verify and complete the attributes field mapping was done using custom-built checklists. The checklists included two tree levels, one shrub and one herb level. The recorded parameters for each level were height, density, overhang and dominant species. Vegetation height was estimated with a precision of +/- 5m, overhang with a precision of +/- 1m, and density with a precision of +/- 20 %. The inclination of the river slope as well as

the roughness of the section (type of regulation, whether sinuous or straight) and type of substrate were noted. From these data sources VTS (see equation (4)) and percent shade were calculated (Fig 2a, 2b). The river morphology parameters river bankfull width (Fig. 2c), wetted width, average water depth and height of river to slope top were also measured.

The riparian vegetation data were obtained after the phenological phase of leaf development was finished and leaves were fully developed (Ellenberg 2012). The river investigated here is strongly influenced anthropogenically and highly regulated. The degree of anthropogenic influence was categorized by Ledochowski (2014) according to Mühlmann (2010) into five categories: entirely natural (1), slightly or not influenced (2), strongly influenced but with natural areas (3), continuously influenced with few natural areas (4) and completely regulated (5) (see Fig. 2c). This categorization mainly describes constraints on bank and riverbed dynamics. The structure and substrate composition of stream bed and vegetation were additional parameters recorded by Ledochowski (2014). The entirely natural class is endowed with riparian vegetation of above 30m height, vegetation densities of 76 to 100 % and a riparian zone of more than 49 m in width. The continuously influenced areas coincide with reduced riparian vegetation strips and reduced vegetation height.

**Vegetation scenarios**

Taking into account all likely changes in tree species, no change in maximum vegetation height or density is predictable. Potential changes can only be induced by different vegetation management strategies as intentional clearings, plantations or mowing. Seven vegetation management scenarios were chosen to estimate the impact of different levels of vegetation shade on future heat waves. This also makes it possible to quantify potential changes to warmth-loving species of reduced height and density. The following scenarios have been considered:

STQ used the best available status quo input data for vegetation, bank and topographic shade as described in Ledochowski (2014). The average density including all land cover types was 66 % (standard deviation = 17 %) and the average height was 9.4m. Only considering areas including trees larger than 15m height the average density rose to 76 % (standard deviation = 11 %), ranging from 2 to 90 %. At the sheltered headwaters (DFS 20) the vegetation density reached 0.89. For V0 within a 50m buffer all vegetation parameters (vegetation height, density and overhang) were set to 0 so that no vegetation shading occurred. The V0 scenario corresponds to intentional clearings and mowing. V100 was defined as: 30 m height, 8 m overhang and 90 % vegetation density within a 50m buffer which is representative for the most dense riparian forests  of existing riparian vegetation (STQ) located in the Pinka catchment (Ledochowski 2014). The V100 scenario represented the maximum possible level of vegetation shade. It is achievable by suspension of clearing and mowing activities as well as additional plantations of local tree and shrub species, who grow to different heights and form a well structured shrub and tree layer. To maintain this scenario management measures like replacement plantatings and well-directed cuttings are necessary. An intermediate height scenario (V50) was defined as 15 m vegetation height and 90 % vegetation density. A reduced density scenario (V70) was defined as 30m vegetation height and vegetation density of 70 %. Additionally scenarios of vegetation density 50 %  and full vegetation height (VD50,VH100),  and vegetation height reduced by 50 % and vegetation density 70 % (VD70,VH50) and vegetation density 50 % (VD50,VH50)were considered. River bank and topography were not changed in the vegetation scenarios. No river restauration in terms restoring natural river bank and allowing natural river dynamics was assumed.

**2.3.3 Definition of sediment layer and conduction flux**

*Heat Source* uses only one substrate temperature, which is representative for the whole sediment layer. The depth of the sediment layer is set to 1m, which corresponds to the available geological information of the River Pinka (Pahr 1984). The substrate temperature used in the model is set equal to the stream temperature at the uppermost model point. For each

consecutive model point the substrate temperature is calculated depending on the local thermal conductivity, thermal diffusivity, layer depth, hyporheic exchange, the river morphological profile and the solar radiation received at the river bed. The sediment of this region is very inhomogeneous and the spatial distribution of the groundwater level is unknown (Pahr 1984). For low flow conditions it was assumed that there was no deep groundwater influence.

### 2.3.4 Definition of discharge

During the analysed period 4 – 8 August 2013 low flow conditions prevailed. The river flow volume increased from 0.18 $m^3s^{-1}$ close to the upstream model boundary at DFS 13 km to 0.76 $m^3s^{-1}$ at the downstream model boundary (DFS 62 km). The mean flow velocity was 0.46 $ms^{-1}$ and it took the river water about 30 hours to traverse the studied length of the river. The model was not sensitive to discharge rates. A decrease in discharge of the upstream boundary station of 0.01 $m^3s^{-1}$ (6 %) led to an increase in average stream temperature from DFS 26 km to 48 km of 0.04 °C (0.2 %) (Trimmel et al. 2016). Because the aim was to estimate the influence of vegetation shade, clear sky periods were chosen where no or only minor precipitation events occurred so discharge was fixed at mean low flow conditions (MLF). MLF was defined as the average of all daily discharges below the 5 % percentile discharge of the climate period 1981 – 2010. The mean low flow conditions (MLF) of the gauging station at Pinggau, DFS 13 km (MLF = 0.143 $m^3s^{-1}$), which is maintained by the Hydrographischer Dienst Österreich was used in the model. At the other end of the study region at DFS 62 km the corresponding flow volume was 0.795 $m^3s^{-1}$. To take into account potential reductions of discharge a scenario of MLF discharge – 15 % (MLF-15 = 0.122 $m^3s^{-1}$), which is a 5 % reduction of the mean annual discharge, was calculated .

### 2.3.5 Upstream boundary stream temperature

Stream temperature and discharge were used as upstream boundary conditions. For the 2013 episode these values rely on observations of the gauging station at Pinggau which is maintained by the Hydrographischer Dienst Österreich and a stream temperature measurement station maintained by the authors. To obtain equivalent data for future conditions, the maximum water temperature was first modelled at DFS 11 km using the expected air temperature as input (Mohseni et al., 1998). The water temperature was split into two components: the long-term seasonal component (or annual component) and the short-term non-seasonal component (or residuals series) (Caissie et al. 2001). The annual component was calculated according to the method of Kothandaraman (1971) and the residuals were calculated with a stochastic second-order Markov model after Cluis (1972) and Salas et al. (1980). Observed hourly water temperatures (N = 12.537) over the period 7 July 2012 to 9 September 2014 were used to fit the model. The coefficient of determination $R^2$ between observed and predicted water temperature for this period was 0.96 and the RMSE was 0.68 °C. For the summer half-year 2013 (Apr – Sept), the $R^2$ was 0.89 and the RMSE was 0.80 °C. To take into account the climatic trend caused by the warming of the land surface (Kurylyk et al. 2015) the difference between the moving average over a 30 year climate period and the reference period 1981–2010 was added to the annual component.

### 2.3.6 Input data of tributaries

The discharge levels and water temperature of the River Pinka at the upstream model boundary and its main 5 tributaries were measured during the 2013 episode in the field by the authors and by two permanent gauging stations. The remaining

unmeasured tributaries added less than 5 % discharge each. Their future water temperatures were synthesized using the daily fluctuations of the water temperature at the upstream model boundary with the adding of a fixed offset depending on the distance of the inflow to the upstream model boundary. Missing discharge information was supplemented using proportions of the discharge levels of the gauge at Burg (DFS 62 km) as  measured during 2013.

**3 Results**

**3.1 Influence of vegetation shade and energy fluxes on stream temperatures during the heat episode 2013 along the river**

In order to interpret the influence of vegetation shade on future water temperature it is important to understand the influence of vegetation shade on the present conditions first. The mean view to sky (VTS) for the study region under current
conditions (STQ) was 0.55. If all vegetation were to be removed (V0) there would still be some remaining shade caused by topography and the river bank, which reduces the maximum VTS value to 0.89. If maximum vegetation was assumed (V100), the value of VTS is strongly reduced, but still amounts to 0.16 on average because a 90 % vegetation density was assumed. Peaks in VTS were found at broader river sections or sections oriented East-West (Fig. 2a). The percentage shade is similar to the inverse of VTS but differs, as the south orientation is of importance (Fig. 2b).
During the STQ scenario the most important energy inputs on the river surface during the study period were short wave radiation flux with an average of 101.6 W m$^{-2}$ (Fig. 3a), sensible heat flux with an average of 39.9 W m$^{-2}$ (Fig. 3d) and long wave radiation with an average of 17.2 W m$^{-2}$ (Fig. 3b). Conduction only amounted to 1.3 W m$^{-2}$ on average (Fig. 3e). The relative percentage of short wave radiation balance, long wave radiation balance and sensible heat flux were 64 %, 11 % and 25 % of the inputs respectively that heated the water column.  The main energy output was latent heat flux (Fig. 3c).
For the V0 and V100 scenario the characteristic of the longitudinal energy fluxes remained the same. During the V0 scenario the relative percentage of short wave radiation balance increased (73 %), while long wave radiation balance (7 %) and sensible heat flux (18 %) decreased. During the V100 scenario the trend was opposite. Short wave radiation balance decreased (47 %) and long wave radiation balance (21 %) and sensible heat flux (32 %) increased (Fig. 3a-e).
Looking at the longitudinal distribution of energy fluxes along the river it can be seen that sensible heat flux and long wave
radiation flux as well as conduction showed their highest values close to the source during all vegetation scenarios. This leads to a rapid increase in the water temperature of the cool spring water, which is clearly seen in both measured and simulated data (Fig. 3g). All energy fluxes were dependent on the degree of openness to the sky, and showed the same pattern along the river (Fig. 3a - f). Short wave radiation and latent heat flux in particular were strongly influenced by the value of the VTS and showed distinct reductions of up to 70 % where shading occurred (Fig. 3a, 3c).
The energy balance was positive on average along the whole river (Fig. 3f). The V0 scenario showed the highest and the V100 scenario the lowest   net energy  with mean values of 55, 40 and 22 W m$^{-2}$ for the V0, STQ and V100 scenarios respectively (Fig. 3f). The greatest differences between the different vegetation scenarios were found close to the source, where during the V0 scenario up to 200 W m$^{-2}$ net energy were available to heat the water column (Fig. 3f), while during the V100 scenario the corresponding figure was only 91 W m$^{-2}$. The positive energy balance can explain the gradual warming of
the stream temperature along the river (Garner et al. 2014) which can be seen in Fig. 3g. The continuous downstream

warming is reversed at about DFSs 16, 22, 26.5, 32, 43.5 and 53.5km by about of 0.5 °C for short distances caused by the addition of cooler water from tributaries (Fig. 3g).

## 3.2 Future climate and advective input

The selection criteria mean air temperature of modelled scenarios increased depending on the return period of the event (Table 1, 2). Apart from the 1a and 5a events of 2030 and the 1a event of 2050, all modelled events were warmer than the 2013 heat wave. Air humidity during the selected events decreased slightly by the end of the century(Table 2). In the 20 year return period event of 2050, wind speeds were higher (1.1 m s$^{-1}$) than in 2030 (0.9 m s$^{-1}$) and 2085 (0.8 m s$^{-1}$) (Table 2). The average global radiation received during each event per day was different for each event as well. For the 20 year return event in 2030, global radiation was 28 MJ m$^{-2}$ d$^{-1}$ i.e. higher than the same scenario in 2050 (23.1 MJ m$^{-2}$ d$^{-1}$) and 2085 (23.1 MJ m$^{-2}$ d$^{-1}$). During the 20 year return event of 2085 on the other hand global radiation was higher than the Max event (20.9 MJ m$^{-2}$ d$^{-1}$) of this climate period (Table 2).

For the mean water temperature at the model boundary an increase of +4.1 °C for a 20 year return event of 2085 with respect to 2013 levels was simulated (Table 2). For the Max event of 2085, which had 2.2 MJ m$^{-2}$ d$^{-1}$ lower global radiation input, a slightly lower temperature increase (+4.0 °C) was simulated (Table 2).

The extraction of future climate data was based on the location of the INCA grid. INCA data for the heat event in 2013 was compared with data measured directly at the river. The INCA data assume a greater distance to the river surface and show higher mean and maximum air temperatures, but also lower air humidity and higher wind speed. This difference in meteorological input data resulted in a 0.1 °C higher measured mean water temperature (Table 3). Maximum water temperature was affected as well, with INCA showing a reduction of 0.3 °C below measured values. Minimum water temperature was 0.6 °C warmer when INCA data input were used. In order to directly compare the 2013 event with the future scenarios, the simulation using the INCA data of 2013 is referred to as "20a OBS" hereafter.

## 3.3 Future stream temperatures

**At DFS 39 km**

To analyse future changes, the initial focus was upon the reference station in the centre of the study region at DFS 39 km. As a temporal reference, the focus was placed on the 20 year return period events of the 2071–2100 climate period as it represents the maximum expected temperature rise.

The mean water temperature of the River Pinka under MLF conditions with unchanged riparian vegetation (STQ) at DFS 39 km during the 20a heat wave event for the periods 2016–2045, 2036–2065 and 2071–2100 was predicted to be 22.4 °C, 22.6 °C and 25.5 °C respectively (Fig. 4, Table 3). The corresponding predicted maximum water temperatures were 25.0 °C, 24.8 °C and 27.3 °C. These predictions represent a significant increase over the mean temperatures of the 20a event of the OBS period of 22.5 °C (maximum temperature: 24.4 °C) by the end of the century.

For mean temperatures, a minor increase in water temperature was predicted for the first half of the century even for extreme heat events with a 20 year return period (Table 4). However, by the end of the century (2071–2100) a remarkable increase in minimum temperatures of +3 °C was modelled. Maximum water temperatures also showed increases. For the period 2016–2045, maximum temperatures increased more rapidly than mean temperatures with a change over baseline

conditions of +0.6 °C. By 2071–2100 the increase in maximum temperatures was predicted to be 2.9 °C compared to the OBS period, which was similar to the predicted increase in mean and minimum water temperatures (Table 4).

Supposing the existing vegetation were removed (V0), the mean water temperature reached 26.7 °C during 20 year return period heat events at the end of the century, which was 4.2 °C above the level of the STQ scenario of the OBS period.

Maximum temperatures reached 28.9 °C, which is 4.5 °C more than in the STQ scenario of the OBS period (Fig. 4, Table 3, 4). Under conditions of maximum riverine vegetation (V100), the expected mean water temperature was predicted to reach only 23.9 °C, which is 1.4 °C above the level of the STQ scenario during 2013 (Fig. 4, Table 3,4). The maximum temperature reached in this scenario is 25.5 °C which is only 1.1 °C above the maximum event of the OBS period (Fig. 4, Table 3, 4).

Vegetation was not able to compensate fully for the temperature increase expected by the end of the century. For the climate period 2036–2065 though, riverine vegetation had the potential to more than compensate for climate change during extreme events and could even cause a cooling of –1.2 °C on average and –1.4 °C with respect to maximum temperatures (Table 4).

**Longitudinal distribution**

During the 2013 heat wave event for the STQ scenario, the stream temperatures increased between the upstream model boundary at DFS 11 km and DFS 62 km by about 7° C (Fig. 3). Looking at the longitudinal distribution of water temperature along the river it can be seen that increases in mean stream temperature caused by increases of future air temperature affected all parts of the river (Fig. 5a-c).

The maximum values showed a similar pattern to the mean values on a higher level. The average difference between mean

and maximum values of the STQ scenario was 3.92 °C, 3.35 °C and 3.91 °C, the maximum difference between maximum values was 5.51 °C, 4.89 °C and 5.51 °C and the standard deviation of this difference was 0.71, 0.66 and 0.71 for 2030, 2050 and 2085 respectively Fig. 5a).

V0 scenarios were always warmer than STQ scenarios and V100 scenarios were always cooler than the STQ scenarios. The mean differences along the river between V0 and STQ were 1.25 °C, 1.26 °C and 1.13 °C, the maximum difference was

1.81 °C, 1.85 C and  1.66 °C, the standard deviation was 0.35,  0.36 and 0.32 for 2030, 2050 and 2085 respectively. The mean difference between STQ and V100 was 1.42 °C,  1.52 °C,  and 1.26 °C, the maximum difference was 1.92 °C, 2.05 °C and  1.72 °C, the standard deviation of this difference was 0.46, 0.49 and  0.41 for 2030, 2050 and 2085 respectively Fig. 5c).

Water temperature was especially sensitive to the removal of vegetation within the first 10 km (DFS 11 – 21 km) where

there were dense forests which prevented the cool headwaters from warming (Fig. 5d). In this region temperatures increased by 1.4 °C under the no-vegetation scenario (V0-STQ). Additional tree cover (V100) caused a temperature reduction of 0.9 °C compared to the STQ scenario (Fig. 5d).  This can be explained by the slower flow velocities in the lower reaches (last 30 km - DFS 32-62: 0.003 m m$^{-1}$, 0.4 m s$^{-1}$ ) in comparison to the steeper upstream sections (first 10 km - DFS 11-21: 0.017 m m$^{-1}$, 0.6 m s$^{-1}$), which gives short wave radiation in unshaded sections more time to heat the water column. For the Pinka

the benefit of additional tree cover maximizing riparian shade became more distinct in the downstream sections (DFS 25-55) where the additional tree cover caused a change of -1.75 °C, while removal only caused a change of around +1.25°C (Fig 5).

**Diurnal ranges**

For aquatic species the mean stream temperature is not the only relevant temperature parameter. The daily temperature range, the absolute minima and maxima as well as the timing when extremes take place are of importance as well. These vary along the river and change depending on the different vegetation shade intensities and discharge volumes (Fig. 6). In the contour plot shown as Figure 6 the warming along the longitudinal gradient is clearly visible, but it is also obvious that the stream is warming to a higher peak each day until the end of the heat episode. In Figure 6's lower panel the daily water temperature amplitude is plotted, along with the energy balance components acting on the river surface for the two locations marked by the black bars in the contour plots. Here the absolute values, amplitude and timing of extremes can be seen. While the energy balance shows the energy input taking place directly at the location, the water temperature includes the energy input of the whole water volume upstream. An upstream site (DFS 20 km) is compared to a downstream site (DSF 61 km). They are both open (VTS of V0 = 0.9, 1) but differ in average water depth (0.09 m, 0.31 m) and discharge levels (0.34 $m^3s^{-1}$, 0.8 $m^3s^{-1}$).

The daily amplitude of the water temperature is strongly damped by the larger flow volume which can be seen in the comparison of the upstream and downstream sites (Fig. 6). A decrease in discharge of -15 % can also be seen to affect the daily minima and maxima of stream temperature in open sections (V0). During the V100 scenario the 15 % discharge reduction has no visible effect (<<0.1 °C).

The daily amplitude of the energy fluxes is not affected by flow volume, but is reduced by vegetation shade. The hourly values of all energy fluxes are reduced synchronously. Decreased solar input and wind access close to the river surface caused by an increase in vegetation density lowers the energy fluxes. From V0 to V100 the maxima can increase more than 2 °C (Fig. 6 and 7). But changes in vegetation density of as little as 20 % can cause an increase of maximum water temperature of more than 0.5 °C (Fig. 7). A change from e.g. 100 % to 70 % raises the heat input by short wave radiation (+17 $Wm^{-2}$ ) convection (+5.6 $Wm^{-2}$ ), and long wave radiation (+3.7 $Wm^{-2}$ ) but only increases heat loss by evaporation from the river surface (-21 $Wm^{-2}$) (Fig. 7). The shading affects the maximum as well as the minimum water temperature and leads to a reduction of the daily amplitude (Fig. 6 and 7). An interesting aspect is that the peak of stream temperature occurs about 1h later when vegetation is included. With a vegetation density reduction of 50 % (VD50) the diurnal range and especially the maximum temperatures are further increased (Fig. 7). It is interesting to note, that halving vegetation height has a similar or less significant effect as reducing vegetation density by 20 % (Fig. 7).

**Trends**

The trend lines were calculated by minimizing the square error. An ANCOVA (analysis of covariance) showed significant interactions between vegetation and air temperature ($p < 0.001$). The equal slope assumption failed, the equal variance test was passed. Mean, maximum and minimum stream temperatures increase as air temperature increases (Fig. 8). Under the assumption of full vegetation, the intercept of the regression line is lowest for the mean and maxima, while under the assumption of no vegetation it is lowest for the minima. The difference between the vegetation scenarios is greatest for the maxima and smallest for the minima. The slope on the other hand is smallest for the maxima and greatest for the minima. All scenarios and values show a squared Spearman's rank correlation coefficient between 0.78 and 0.93. For mean and maximum temperatures the trend line of V0 is steeper than V100 (17 %), which means, that supposing no vegetation the maximum temperatures will increase at a higher rate. For the daily minima the difference in slope is even greater (30 %).

The regression lines of the halved vegetation height scenario (V50) and the reduced vegetation density scenario (V70) cross for minima, mean and maxima values. The change in slope though is small (3.6 %, 1.4 % and 5.8 % for the mean, minima and maxima respectively) and statistically not significant.

## 4 Discussion

### 4.1 Energy fluxes during heat waves

In the present article evaporative heat flux was responsible for 100 % of heat loss from river water on average. Short wave radiation balance, long wave radiation balance and sensible heat flux were 64 %, 11 % and 25 % of the total energy input respectively.

During summer periods of high air temperature the difference between air and water temperature increases, which can trigger intensified evaporative flux that cools the river, but can also cause sensible heat flux to heat the water column (Benyahya et al. 2012). Benyahya et al. 2012 found that evaporative heat flux accounted for 100 % of energy outputs during 7-23 June 2008 while short wave radiation balance, long wave radiation balance and sensible heat flux were 72.53 %, 24.05 % and 2.03 % of the energy input respectively.

### 4.2 Magnitude of stream temperature rise

The modelled 20 year return period heat wave (20a) in the climate period 2071–2100 showed a +3.8 °C increase in air temperature with respect to the observed period. Increases in maximum, mean and minimum stream temperatures of close to +3 °C with respect to the observed period were simulated for this episode. During the Max event, the modelled increases of maximum, mean and minimum temperatures were 3.4 °C, 3.5 °C and 4 °C respectively. When looking at the whole river, mean changes of 3.3 °C for the maximum and 3.9 °C mean temperatures were calculated. Melcher et al (2014) also found that average and maximum temperatures show similar warming trends. An increase of 3.9 °C from the OBS period to 2085 corresponds to an increase of 0.43 °C / decade. An increase of 3 °C equates to an increase of 0.33 °C / decade.

The relatively low values of water temperature predicted for the 20a 2050 heat wave might be explained by higher wind speeds and lower air humidity causing higher evaporation rates and lower solar radiation energy input compared to 2013. The relatively low modelled temperatures were most evident in maximum water temperatures. For the V0 scenario relatively low water temperatures were also predicted, which was caused by increased evaporation. The maximum vegetation scenario (V100) shows similar stream temperatures to 2013.

Temperature increase in Austrian stream waters is well-documented and ubiquitous. All 230 stations of the Austrian hydrographic central office, with different elevations, distances from source and catchment areas recorded increases of stream temperature of an average of 1.5 °C (0.48 °C / decade) from 1980 to 2011 (BMLFUW 2011). (The data were elevation-corrected using External Drift Top Kringing (Skøien et al. 2006) and a mean trend was calculated using the Mann-Kendall Test (Burn and Hag Elnur, 2002)). Melcher et al. (2013) analysed 60 stations and found a similar trend of 1 °C within the last 35 years for mean August temperatures, which was independent of the river type (0.29 °C / decade). The annual mean temperature of the River Danube has been rising (Webb and Nobilis 1995) and is likely to continue to rise to reach a value between 11.1 and 12.2 °C by 2050 compared to around 9 °C at the beginning of the 20th century at the border

with Slovakia (Nachtnebel et al. 2014). Dokulil (2013) extrapolated the quadratic regression of the period 1900-2006 of the river Danube near Vienna and predicted an increase of up to 3.2 °C by 2050 with respect to 1900 (0.21 °C / decade). Using linear regression the increase was only 2.3 (0.15 °C / decade), but using the linear trend beginning from 1970 the increase was 3.4 °C (0.23 °C / decade). Due to the size of the River Danube, daily amplitudes and extremes are not comparable to the Pinka, but trends in mean water temperature values are comparable though. The temperature values predicted by this study were clearly greater than the model uncertainty and lie in the upper region of the values published by other studies (BMLFUW 2001; Dokulil 2013; Melcher et al. 2013, 2014).

Considering a likely discharge decrease (Nachnebel et al. 2014), a slightly higher temperature rise might be expected. Van Vliet et al. (2011) analysed 157 river temperature stations globally for the 1980–1999 period and predicted increases of annual mean river temperature of 1.3 °C, 2.6 °C and 3.8 °C under air temperature increases of 2 °C, 4 °C and 5 °C respectively. Discharge decreases of 20 % and 40 % increased the modelled water temperature rises by 0.3 °C and 0.8 °C on average (Van Vliet et al. 2011).

### 4.3 Ability of riparian vegetation to mitigate the expected stream temperature rise

How will riparian vegetation systems behave in the future, what are the feedback mechanisms of increased shading under a warmer heat wave scenario? Decrease in discharge caused by increased evaporation from the river surfaces caused by missing riparian vegetation (V0 compared to V100) was calculated to be -0.001 $m^3s^{-1}$ at the lower boundary of the river (DFS 61). Also during an MLF reduced by 15 % the loss of water to evaporation was only -0.001 $m^3s^{-1}$. Therefore mass loss was not found to be a significant driver of temperature rise in a river of this size. Further there might be a potential decrease of discharge caused by increased withdrawal of river water by the riparian vegetation under warmer climates. As species of the floodplain forest are "spender" type plants that do not economise their water use, this needs to be considered. In this study a simulation is included with a discharge decrease of 15 %, a level that is presently expected from past observations. This estimation includes precipitation losses as well as increased evapotranspiration by the soil-vegetation system of the catchment area and increased evapotranspiration by the riparian vegetation via rises in air temperature. Different discharge scenarios were not simulated for all episodes, because the fact that low flow situation was chosen was more dominant than the expected reduction by 15 %.

The increased air humidity and reduced air temperature caused by transpiration of riparian vegetation close to the river reduces air humidity and air temperature gradients. The effect on water temperature was calculated to be a maximum of around 0.2 °C. More directly vegetation affects water temperature by reducing short wave radiation input, but also it reduces the view to sky which affects long wave radiation balance and the turbulent heat fluxes. Community changes which might affect vegetation height and density are possible within the next century though changes in vegetation height and density in floodplain forests in natural systems are mainly due to succession. Primack (2000), Garssen et al. (2014), Rivaes et al. (2014) studied the effect of climate change on natural riparian vegetation cover via changes in the hydrological regime including inundation periods and intensity, days since rain and the decline of water table. As the River Pinka is anthropogenically influenced and will be regulated for the foreseeable future no dynamical changes and no natural succession dynamics are expected which could cause an extreme change in vegetation cover.

Different vegetation scenarios were simulated in this study to quantify the potential effects of shading and wind reduction caused by vegetation. Compared to the status quo (STQ) scenario, additional riparian vegetation (V100) could reduce

maximum stream temperatures during extreme heat waves by 2.2 °C, mean temperatures by 1.6 °C and minimum temperatures by 0.9 °C (Table 4). Removal of existing vegetation (V0) amplified stream temperature increases, and could cause an average increase of maximum, mean and minimum stream temperatures of 1.8 °C, 1.3 °C and 1.0 °C respectively in comparison with the STQ vegetation scenario (Table 4).

Removal of vegetation (V0) magnified stream temperatures during 20 year return period events by the end of the century by up to 4.2 °C (mean) and 4.5 °C (daily maximum). Additional riparian vegetation (V100) on the other hand mitigated part of the rise in maximum temperatures, so there was only a 1.1 °C increase. Although the increase of mean temperatures was reduced to about 1.4 °C, riparian vegetation management alone was not enough to compensate for the predicted warming caused by climate change.  The water temperature reduction rates predicted in the present article lie within the range of

observed changes of pre- and post harvest situations found in literature (Cole and Newton 2013; Moore et al. 2005).

The maximum water temperatures during heat waves in particular could be reduced significantly by vegetation shade. The daily mean and daily maximum temperature tends to increase more strongly for higher air temperatures if less vegetation is present. Daily minimum temperatures increase at an even higher rate. These trends are in agreement with findings about experimental data analysed by Kalny et al. (2017).

Vegetation height and density can alter the slope of the temperature trend line. For example with dense low vegetation, water temperature starts lower and ends higher for the same air temperature compared to the high and less dense vegetation scenario, which indicates that there is some impeding of cooling during the night by lower vegetation compared to higher vegetation.  Water temperatures rise more rapidly for dense low vegetation than high vegetation of reduced density. High vegetation of lower density cannot compete with dense high vegetation in terms of reduction of stream water temperature

though.

During heat wave situations the reduction in air exchange causes an important lag in temperature rise, so the time of maximum solar exposure does not coincide with the maximum heat stress caused by water temperature. This lag is known in the literature (Brown and Krygier 1970). Apart from its influence on stream temperature, vegetation can cast spatially differentiated shade, which results in areas of different sun exposure and energy balance. This heterogeneity can provide

ecological niches which are important for different development stages of river fauna (Clark et al. 1999).

**4.4 Limitations**

Vegetation mainly causes lower maximum stream temperatures by reducing the solar radiation input at the river surface by shading. This effect is strong during times of clear skies and high solar irradiation. Under cloudy conditions this effect is less pronounced and during night time it is absent, but outgoing long wave radiation is still impeded. This in turn could lead

to higher mean and minimum temperatures, which can be also seen in the simulated low global radiation scenarios.

Although vegetation can have important effects on stream temperature, there will be river sections which will not be affected by the addition (or removal) of vegetation due to upstream or lateral, surface or subsurface advection of heat or topographic shade (Johnson and Wilby 2015). Ground water influence was not measured and no ground water influence was assumed in the model. Although the model performed well (RMSE 0.88 °C) there might be some ground water influence

between DFS 45 km and 55 km where the measurements lie below the simulation results. Other possible future alterations to the river via development or climate change were not considered here. These include potential anthropogenic heat sources or sinks like discharges of tempered waste water, possible changes in stream velocity and shading, sediment changes caused

by impoundments, regulation and canalization, or discharge changes such as withdrawal of water for irrigation. The climate input used only one possible emission scenario simulated by one regional climate model. The percentage contributions of surface, subsurface, groundwater and/or snow melt still have to be analysed in more detail (Johnson and Wilby 2015). Apart from rising air temperatures and discharge changes, anthropogenic influences like discharges from waste water treatment plants and cooling water can influence stream temperatures in a negative way and are therefore presently illegal in Austria (WRG 1959). Other possible consequences of climate change are changes in sediment loads in river systems due to changes in mobilization, transport and deposition of sediment, which is expected to be very likely (APCC 2014). Sediment changes might alter the bed conduction flow as well as flow velocity, which can influence the magnitude and variability of stream temperature. Artificial changes which deteriorate the situation are presently illegal in Austria as well (WRG 1959).

**5 Conclusions**

In this study the influence of expected changes in heat wave intensity during the 21st century on stream temperature in the rithron to upper potamal section of the human impacted eastern Austrian River Pinka were simulated and the mitigating effect of riparian vegetation shade on the radiant and turbulent energy fluxes was analysed. By the end of the century (2071–2100) in the study region an air temperature increase of 3.8 °C to 5.6 °C was predicted during annual or less frequent extreme heat waves in comparison to the observed period of 1981–2010. Stream water temperature increases of less than 1.5 °C were modelled for the first half of the century. For the period 2071–2100 a more significant increase of 3 °C in maximum, mean and minimum stream temperatures was predicted for a 20 year return period heat event.

Discharge changes caused by increased evaporation due reduced shade was not found to be significant. Discharge changes caused by precipitation and increased evapotranspiration in the catchment area as expected from past observations was found to be insignificant compared to the changes caused by vegetation shade.

Vegetation could reduce stream temperature during heat waves when conditions of high solar radiation predominate. Even when maximum vegetation extent with maximum height and density including plantations and replacement plantings was assumed, the additional riparian vegetation was not able to fully mitigate the expected temperature rise caused by climate change. But during extreme heat waves maximum stream temperatures could be reduced by 2.2 °C, and mean temperatures by 1.6 °C. Removal of existing vegetation amplified stream temperature increases, and could cause an increase of maximum and mean stream temperatures by 1.8 °C and 1.3 °C respectively in comparison with the status quo vegetation scenario. With complete vegetation removal, maximum stream temperatures in annual heat events at the end of the century could increase by more than 4 °C compared to the present time.

Daily amplitudes were reduced by riparian vegetation and the timing of the peak temperature was delayed by about one hour. A reduction of vegetation density by 20 % had shown a similar effect as a 50 % reduction of vegetation height. Vegetation can reduce maximum temperatures more effectively on an absolute scale but also reduced the trends significantly compared to the no vegetation scenario. Minimum temperatures increased most.

This study shows that it is very likely that during extreme events a temperature increase of 2 °C will be exceeded during this century. This is the magnitude of an average of 2 °C which is the temperature differentiation of fish zones and in particular for the occurrence of native cold water and warm water preferring fish species (Logez et al. 2013; Melcher et al. 2013; Pletterbauer et al. 2015). At a stream temperature of 20 °C, cold water adapted species begin to experience temperature-

induced mortality (Melcher et al. 2014; Schaufler 2015). During a simulated annual heat wave event in the period 2016–2035 this threshold was never exceeded in the most upstream region (DFS 13 km), which is presently populated by the cold adapted species brown trout (Guldenschuh 2015). At the end of the century during a heat wave event of a 20 year return period the threshold was likely to be exceeded for 72 of 120 h. At the lower boundary of the trout zone (DFS 20 km), the 20 °C mark was exceeded for 70 of the 120 h during heat waves at the beginning of the century, but riparian vegetation shade could reduce this period to 9 h in total. The mitigation possibilities of vegetation were limited though, and could not fully compensate for the whole predicted temperatures rise. At the end of the century in heat waves of a 5 year or shorter return period, even if maximum vegetation was assumed, 20 °C was exceeded during the whole heat wave event.

Global warming has multiple impacts on changes in aquatic ecosystems, whereas in combination with loss of habitat and other human pressures, this is leading to a deadly anthropogenic induced cocktail (Schinegger et al., 2011). The study affirmed the importance of shading and riparian vegetation along river banks for aquatic biodiversity and indicates the added value of riparian vegetation to mitigate climate change effects on water temperature. During this study no economic evaluation of the vegetation scenarios could be done. While maximum vegetation height and densities of 50 % can easily be reached without external efforts, this process can certainly be accelerated as well as high densities assured by planting additional trees. This comes at a certain cost, but it might be worth to invest. The used method provides a model for weighting of interactions of environmental parameters especially during heat wave events. The findings and recommendations gained with this methodology can help key decision makers choosing the right restoration measures. The study in general emphasizes the importance of land-water interfaces and their ecological functioning in aquatic environments.

## 6 Appendix

### 6.1 Abbreviations

DFS            distance from source

INCA          integrated nowcasting through comprehensive analysis

VTS            view to sky

*climate episodes:*

1a, 5a, 20a       episodes of 1 year, 5 year, 20 year return period within a 30 year climate period

Max            maximum event of a 30 year climate period

OBS           observed period (1981 - 2010)

2030, 2050, 2085    30 year climate period centred on 2030 (2016 – 2045), 2050 (2036-2065), 2085 (2071 - 2100)

*discharge scenarios:*

MLF           mean low flow of the gauging station at DFS 13 km: 0.143 $m^3s^{-1,}$ DFS 62 km: 0.795 $m^3s^{-1}$

MLF-15        MLF minus 15 % discharge

*vegetation scenarios:*

STQ           "status quo", exisiting/actual vegetation

V100          "maximum vegetation" - vegetation height 30 m, vegetation density 90 % (VD90, VH100)

| | |
|---|---|
| V70 | "reduced density" - vegetation height 30 m, vegetation density 70 % (VD70, VH100) |
| V50 | "intermediate vegetation height" - vegetation height 15 m, vegetation density 90 % (VD90. VH50) |
| V0 | "no vegetation" |
| VD50, VH100 | vegetation density 50 %, vegetation height 100% (30m) |
| VD70, VH50 | vegetation density 70 %, vegetation height 50 % (15 m) |
| VD50, VH50 | vegetation density 50 %, vegetation height 50% (15m) |

**Team list** (alphabetical order): Herbert Formayer, Clement Gangneux, Gerda Kalny, Valeria Ledochowski, David Leidinger, Andreas Melcher, Imran Nadeem, Hans Peter Rauch, Heidelinde Trimmel, Philipp Weihs, David Whittaker

**Code availability**: The last official version of *Heat Source* software used are available online at: http://www.deq.state.or.us/WQ/TMDLs/tools.htm
The changes included in *Heat Source* within this study will be implemented in the next version, which will be available at the same location.

**Data availability**: The simulation input and result data sets for the present and future heat wave episodes used in this article are published on the freshwater biodiversity data portal (https://doi.org/10.13148/BFFWM8). As they are part of the research project BIO_CLIC the metadata is published together with the other vegetation, morphological and biological data sets produced in the project in the Freshwater Metadata Journal (https://doi.org/10.15504/fmj.2017.22).

**Authors contributions**: Melcher A. was in charge of the hydrobiological aspects. Weihs P. helped to better understand the energy fluxes of the riverine system. Formayer H. selected the climate episodes and helped to interpret the significance of the results. David L. produced the climate episode data and the upstream boundary water temperature. Kalny G. organized the field campaigns and helped Valeria Ledochowski to built the basic vegetation and morphology data set. Trimmel H. organized and executed the water temperature measurements further processed the all input data for the use of *Heat Source*, adapted and validated the model. She ran the *Heat Source* simulations for all selected episodes and prepared the manuscript.

**Acknowledgements**: This research was part of the project BIO_CLIC and LOWFLOW+ both funded within the Austrian Climate Research Programme (ACRP) by the Klima und Energiefond. The regional climate model data sets used to produce the climate episodes were developed in the ENSEMBLES project supported by the European Commission's 6th Framework Programme through contract GOCE-CT-2003-505539. The INCA data set was created by the Zentralanstalt für Meteorologie und Geodynamik (ZAMG). Hydrological data and the digital elevation model were provided by hydrographic services, which are part of the Federal Ministry of Agriculture, Forestry, Environment and Water management and the federal state governmental geoinformation service authorities of Styria and Burgenland. Bernhard Spangl from the Institute of Applied Statistics and Computing (IASCBOKU) gave advice regarding statistics. Special thanks are given to the Oregon Department of Environmental Quality, who maintain the model *Heat Source* and opened the source code for scientific use.

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

**Table 1: Mean 5 day air temperatures of modelled future heat wave episodes used as selection criteria, shown with equivalent values from the observed period for comparison.**

|  | 1a | 5a | 20a | Max |
|---|---|---|---|---|
| 1981-2010 ("OBS") | 23.1 | 25.0 | 27.2 | 27.4 |
| 2016-2045 ("2030") | 23.4 | 26.6 | 27.2 | 29.0 |
| 2036-2065 ("2050") | 24.2 | 27.2 | 28.4 | 28.8 |
| 2071-2100 ("2085") | 28.1 | 30.6 | 31.0 | 32.0 |

5 **Table 2: Mean and daily maximum air temperature, air humidity, wind speed, global radiation at the reference station and water temperature at the upstream model boundary averaged for the selected 5 day heat episodes in 2013 and the 1a, 5a, 20a and Max events of the climate periods centred on 2030, 2050 and 2085. For 2013 (OBS) measured values of the reference station 2 m above the river (M.) and interpolated measurement data from the INCA (I.) data set are shown.**

|  | OBS | | 2030 | | | | 2050 | | | | 2085 | | | |
|---|---|---|---|---|---|---|---|---|---|---|---|---|---|---|
|  | M. | I. | 1a | 5a | 20a | max | 1a | 5a | 20a | max | 1a | 5a | 20a | max |
| Air temp. (mean) [°C] | 26.2 | 27.2 | 23.3 | 26.6 | 27.2 | 29.0 | 24.2 | 27.2 | 28.4 | 28.8 | 28.1 | 30.6 | 31.0 | 32.0 |
| Air temp (mean daily max) [°C] | 34.5 | 35.7 | 30.0 | 33.7 | 34.6 | 37.5 | 29.5 | 33.7 | 35.9 | 36.9 | 34.8 | 38.2 | 39.6 | 39.0 |
| Air humidity [%] | 62 | 55 | 73 | 57 | 55 | 53 | 54 | 56 | 56 | 60 | 58 | 51 | 48 | 52 |
| Wind speed [m s-1] | 0.6 | 1.4 | 0.7 | 0.9 | 0.9 | 1.0 | 1.3 | 1.1 | 1.1 | 0.8 | 1.3 | 1.2 | 0.8 | 0.9 |
| Global rad. [MJ m$^{-2}$ d$^{-1}$] | 24.6 | 24.6 | 23.4 | 25.0 | 28.0 | 29.0 | 24.9 | 28.7 | 23.1 | 21.7 | 27.3 | 24.5 | 23.8 | 20.9 |
| Boundary water temperature [°C] | 16.3 | 16.3 | 14.1 | 15.9 | 16.0 | 16.8 | 15.6 | 16.2 | 17.0 | 17.5 | 17.5 | 19.4 | 20.4 | 20.3 |

**Table 3: Daily minimum, mean and maximum 5 day mean water temperatures of the 5 day episodes averaged over the River Pinka during the 1a, 5a and 20a episodes for the climate periods centred on 2030, 2050 and 2085 and mean low flow discharge at DFS 39. For 2013 (OBS), the measured values of the reference station 2 m above the river (Meas.) and interpolated measurement data from the INCA data set are compared.**

|            | (a) max. |      |      | (b) mean |      |      | (c) min. |      |      |
|------------|----------|------|------|----------|------|------|----------|------|------|
|            | V0       | STQ  | V100 | V0       | STQ  | V100 | V0       | STQ  | V100 |
| OBS Meas.  | 26.6     | 24.7 | 22.4 | 23.8     | 22.4 | 20.7 | 20.2     | 19.5 | 18.5 |
| OBS INCA   | 26.1     | 24.4 | 22.1 | 23.7     | 22.5 | 20.8 | 21.0     | 20.1 | 19.2 |
| 2030_1a    | 24.5     | 23.1 | 20.7 | 21.5     | 20.4 | 18.6 | 16.5     | 16.5 | 16.3 |
| 2030_5a    | 25.9     | 24.3 | 22.1 | 22.5     | 21.3 | 19.7 | 17.8     | 17.2 | 16.5 |
| 2030_20a   | 27.0     | 25.0 | 22.5 | 22.2     | 22.4 | 20.2 | 19.4     | 18.2 | 17.2 |
| 2030_Max   | 27.2     | 25.7 | 23.5 | 24.8     | 23.4 | 21.6 | 21.9     | 20.8 | 19.5 |
| 2050_1a    | 24.3     | 22.6 | 20.0 | 21.6     | 20.4 | 18.9 | 19.0     | 18.2 | 17.3 |
| 2050_5a    | 26.5     | 24.8 | 22.2 | 23.7     | 22.3 | 20.5 | 20.4     | 19.5 | 18.4 |
| 2050_20a   | 26.6     | 24.8 | 23.0 | 23.7     | 22.6 | 21.3 | 20.2     | 19.9 | 18.9 |
| 2050_Max   | 27.5     | 25.9 | 23.7 | 25.1     | 23.9 | 22.2 | 22.5     | 21.5 | 20.4 |
| 2085_1a    | 28.6     | 24.9 | 23.1 | 26.2     | 22.5 | 21.7 | 22.3     | 18.8 | 18.8 |
| 2085_5a    | 29.0     | 27.3 | 25.0 | 26.5     | 25.3 | 23.7 | 24.1     | 23.0 | 21.7 |
| 2085_20a   | 28.9     | 27.3 | 25.5 | 26.7     | 25.5 | 23.9 | 23.6     | 22.9 | 21.7 |
| 2085_Max   | 29.3     | 27.8 | 25.7 | 27.1     | 26.0 | 24.6 | 25.0     | 24.1 | 23.0 |

**Table 4: Differences between the 20a event of the OBS period (2013) (with mean low flow discharge) of predicted maximum (a), mean (b) and minimum (c) water temperatures for the 1a, 5a, 20a and Max event at DFS 39 km for the climate periods centred on 2030, 2050 and 2085 for vegetation scenario V0 (no vegetation), STQ (vegetation unchanged), V100 (maximum vegetation).**

|          | (a) max. |      |       | (b) mean |      |       | (c) min. |      |       |
|----------|------|------|-------|------|------|-------|------|------|-------|
|          | V0   | STQ  | V100  | V0   | STQ  | V100  | V0   | STQ  | V100  |
| OBS INCA | 1.7  | 0    | -2.3  | 1.2  | 0    | -1.7  | 0.9  | 0    | 0.9   |
| 2030_1a  | 0.1  | -1.3 | -3.7  | -1   | -2.1 | -3.9  | -3.6 | -3.6 | -3.8  |
| 2030_5a  | 1.5  | -0.1 | -2.3  | 0    | -1.2 | -2.8  | -2.3 | -2.9 | -3.6  |
| 2030_20a | 2.6  | 0.6  | -1.9  | 0.3  | -0.1 | -2.3  | -0.7 | -1.9 | -2.9  |
| 2030_Max | 2.8  | 1.3  | -0.9  | 2.3  | 0.9  | -0.9  | 1.8  | 0.7  | -0.6  |
| 2050_1a  | -0.1 | -1.8 | -4.4  | -0.9 | -2.1 | -3.6  | -1.1 | -1.9 | -2.8  |
| 2050_5a  | 2.1  | 0.4  | -2.2  | 1.2  | -0.2 | -2    | 0.3  | -0.6 | -1.7  |
| 2050_20a | 2.2  | 0.4  | -1.4  | 1.2  | 0.1  | -1.2  | 0.1  | -0.2 | -1.2  |
| 2050_Max | 3.1  | 1.5  | -0.7  | 2.6  | 1.4  | -0.3  | 2.4  | 1.4  | 0.3   |
| 2085_1a  | 4.2  | 0.5  | -1.3  | 3.7  | 0    | -0.8  | 2.2  | -1.3 | -1.3  |
| 2085_5a  | 4.6  | 2.9  | 0.6   | 4    | 2.8  | 1.2   | 4    | 2.9  | 1.6   |
| 2085_20a | 4.5  | 2.9  | 1.1   | 4.2  | 3    | 1.4   | 3.5  | 2.7  | 1.6   |
| 2085_Max | 4.9  | 3.4  | 1.3   | 4.7  | 3.5  | 2.1   | 4.9  | 4    | 2.9   |

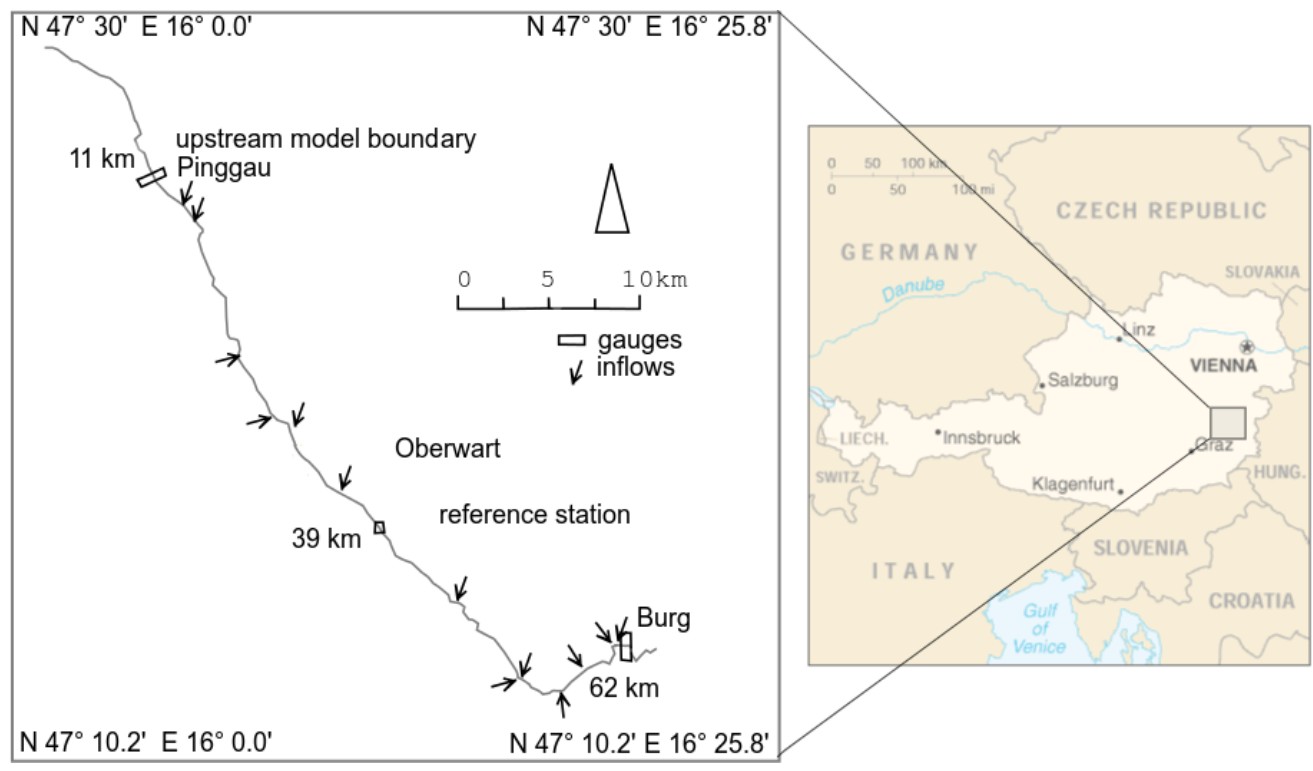

**Figure 1: The study region in Pinka showing gauges, tributaries and the reference station (km markers shown as distance from source).**

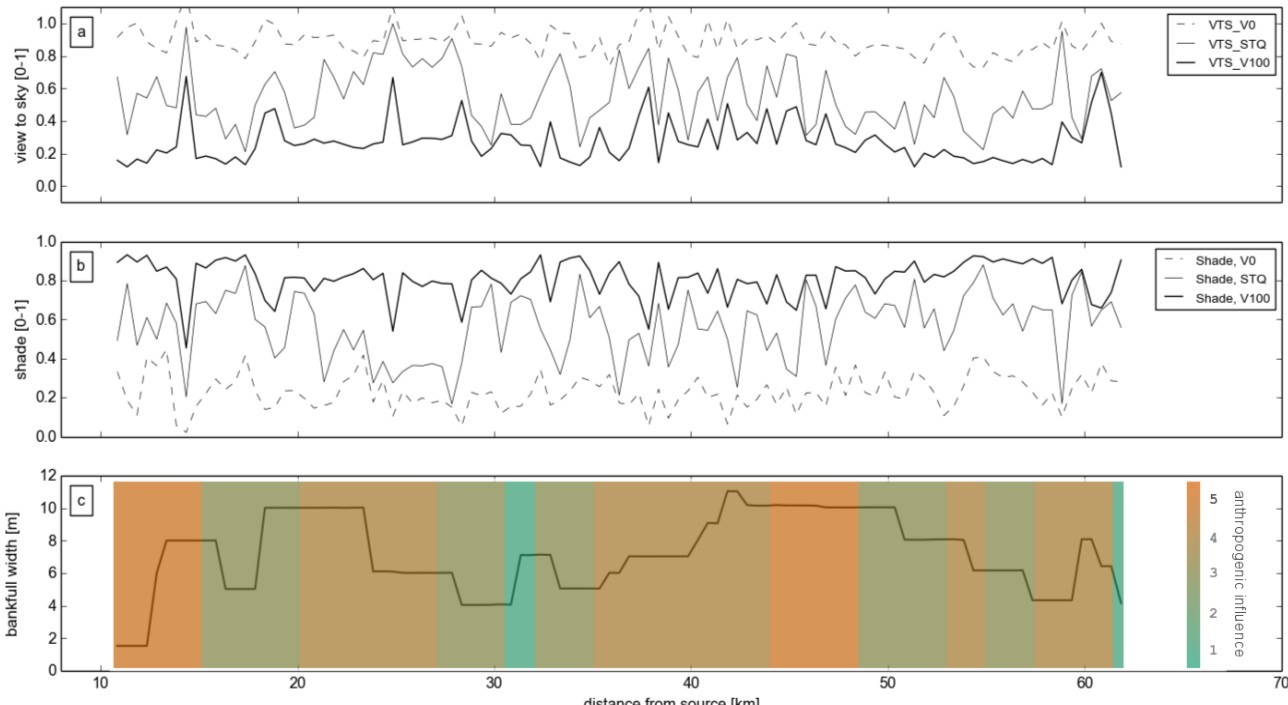

**Figure 2: Characteristics of the River Pinka. (a) The longitudinal distribution of view to sky (VTS) and (b) shade at the river's surface, (c) the bankfull width and the level of anthropogenic influence on the river (legend on the right: entirely natural: 1, slightly or not influenced: 2, strongly influenced but with natural areas: 3, continuously influenced with few natural areas: 4 and completely regulated: 5).**

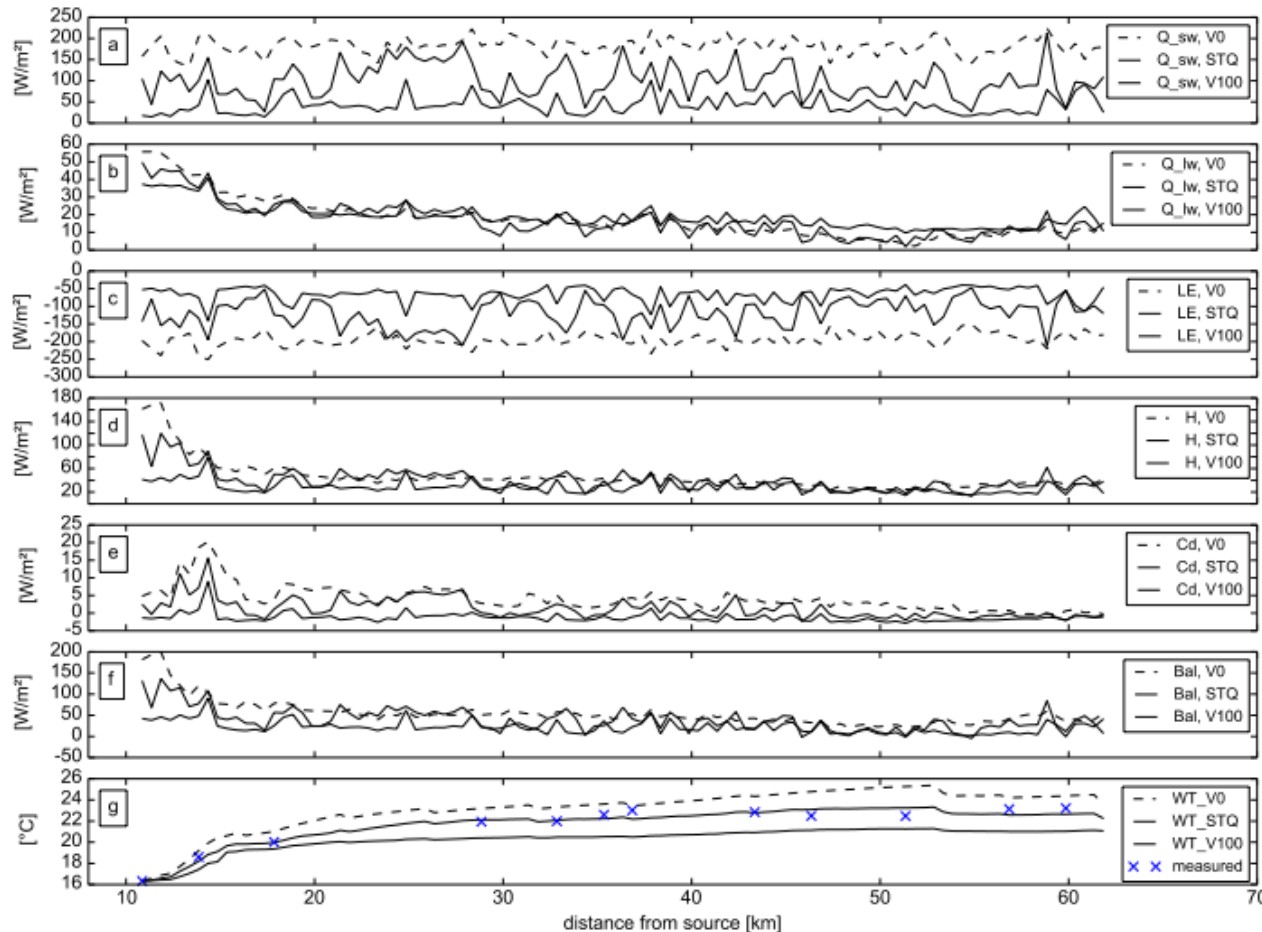

**Figure 3: Comparison of (a) the calculated short wave, (b) long wave radiation balance, (c) latent and (d) sensible heat flux, (e) conduction heat flux, (f) total energy balance and (g) measured (measured) and simulated (WT) water temperature for the heat wave episode 4 – 8 August 2013 along the River Pinka for three vegetation scenarios: no vegetation (V0), existing vegetation (STQ) and maximum vegetation (V100).**

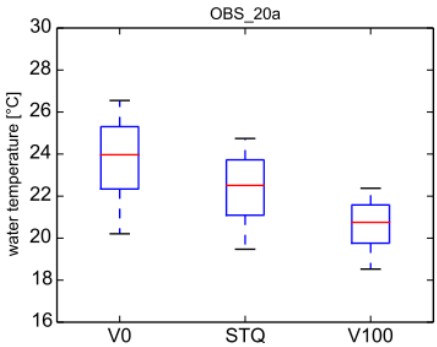

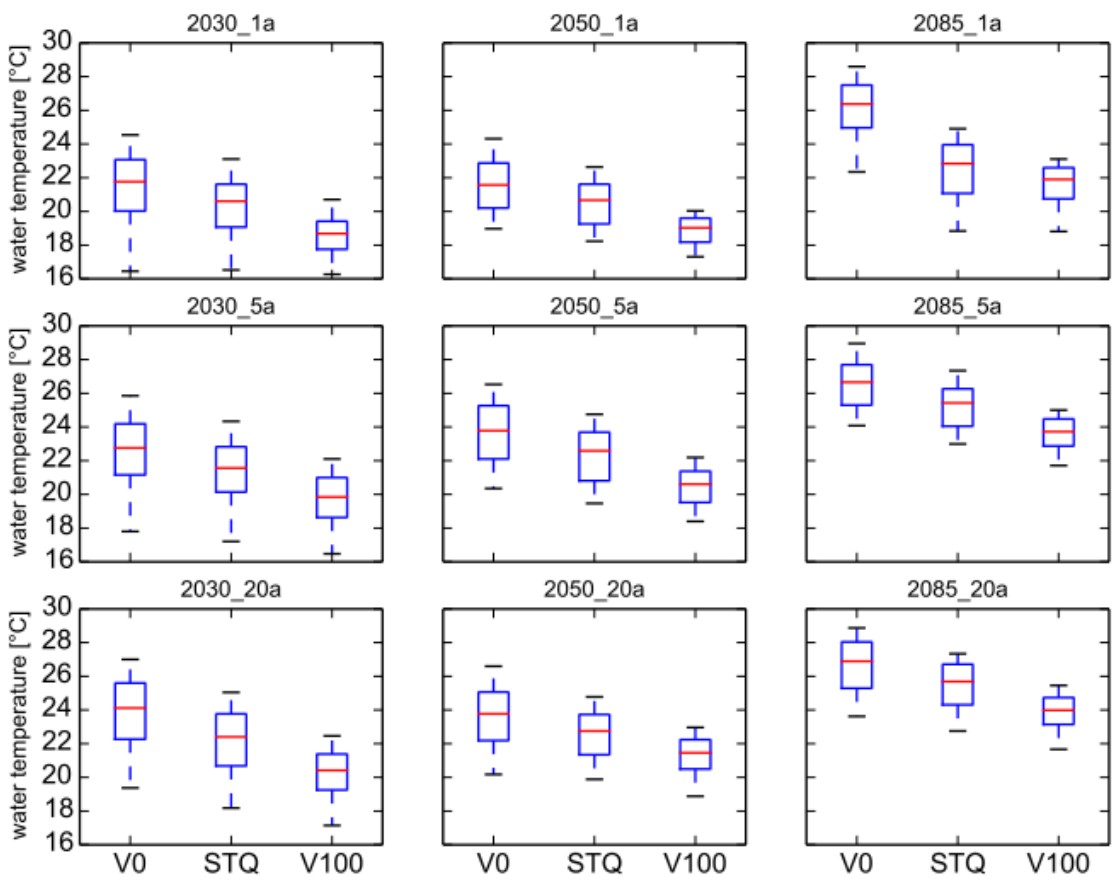

**Figure 4: Box and whiskers charts showing the 5 day mean water temperature distribution during the 1a, 5a and 20a episodes for the climate periods centred on 2030, 2050, 2085 with mean low flow discharge at DFS 39 km. The hourly values of V0 (no vegetation) and V100 (full vegetation) are significantly different from STQ in all episodes (p<0.0001).**

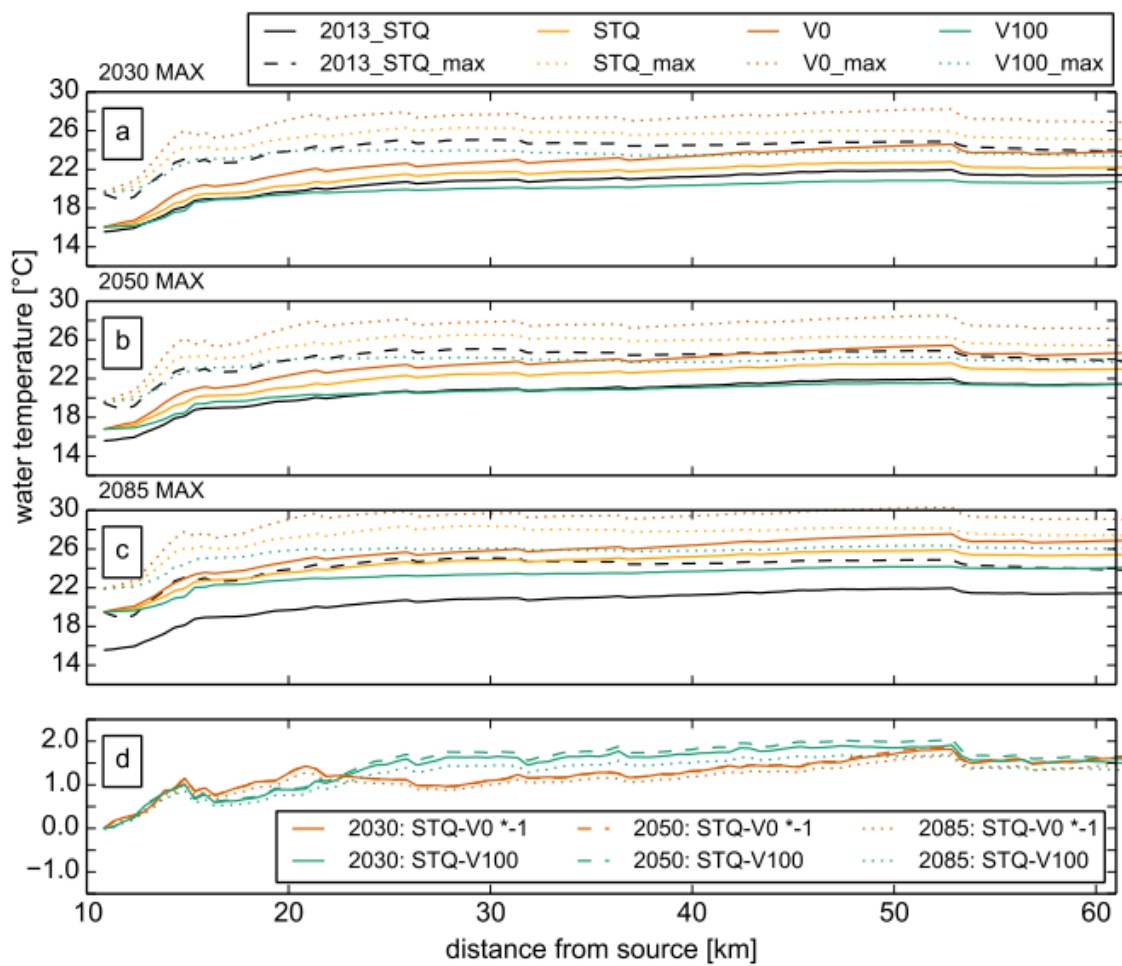

**Figure 5: Mean and maximum water temperature averaged during the maximum events predicted for the climate periods centred on (a) 2030, (b) 2050 and (c) 2085 along the River Pinka using vegetation scenarios V0 (no vegetation), STQ and V100 (full vegetation) in comparison to the maximum event recorded in 2013. (d) The difference between STQ and V100 (green) and STQ and V0 (*-1) (red).**

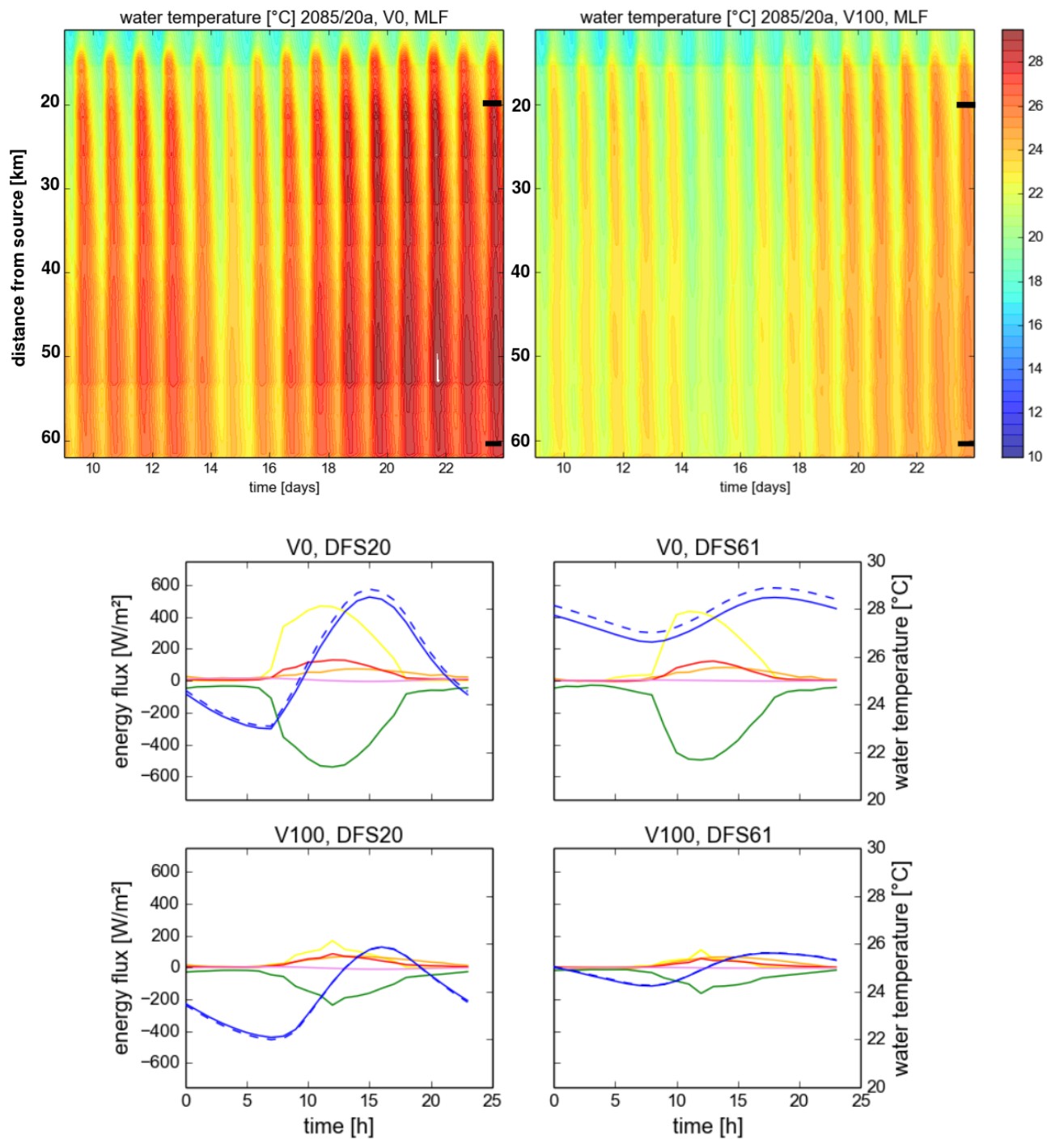

**Figure 6: The effect of a 15% discharge reduction (MLF– 15%) of the mean low flow conditions (blue, dashed) on stream temperature compared to MLF (blue, solid) for an upstream (DFS 20 km) and downstream location (DFS 61 km) for the 20 year**
5   **return period event centred on 2085 for no vegetation (V0) and maximum vegetation (V100). Diurnal amplitude of all energy fluxes (short wave radiation balance = yellow, latent heat flux = green, long wave radiation balance = red, sensible heat flux = orange, conduction heat flux =v iolett) .**

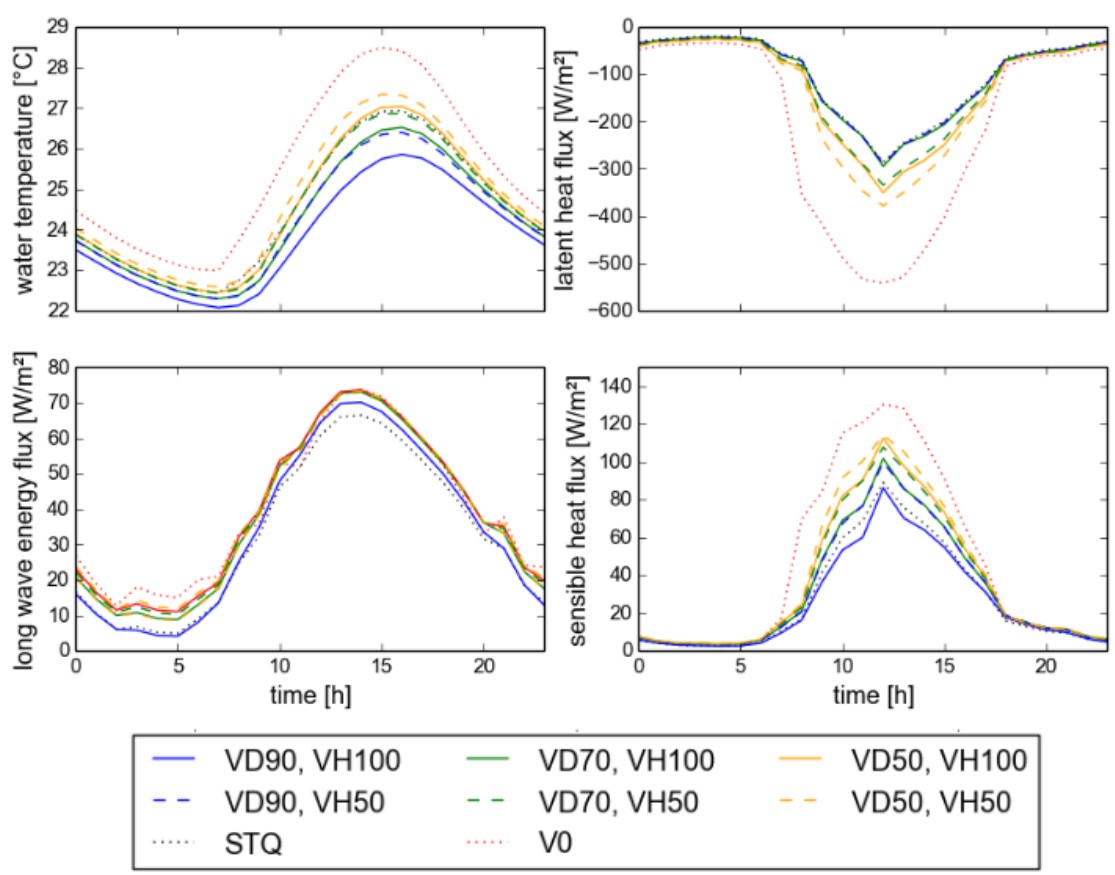

**Figure 7: The effect of the vegetation scenarios of maximum vegetation height (VH100) and 50% vegetation height (VH50), natural dense vegetation (VD90), natural light vegetation (VD70), sparse vegetation (VD50), V0 (no vegetation), STQ (actual vegetation) on the diurnal amplitude of water temperature and the air temperature dependent energy fluxes longwave radiation, sensible and latent heat flux for the 20 year return period events of the final day of the climate periods centred on 2085, for mean low flow conditions (MLF) for an upstream location (DFS 20).**

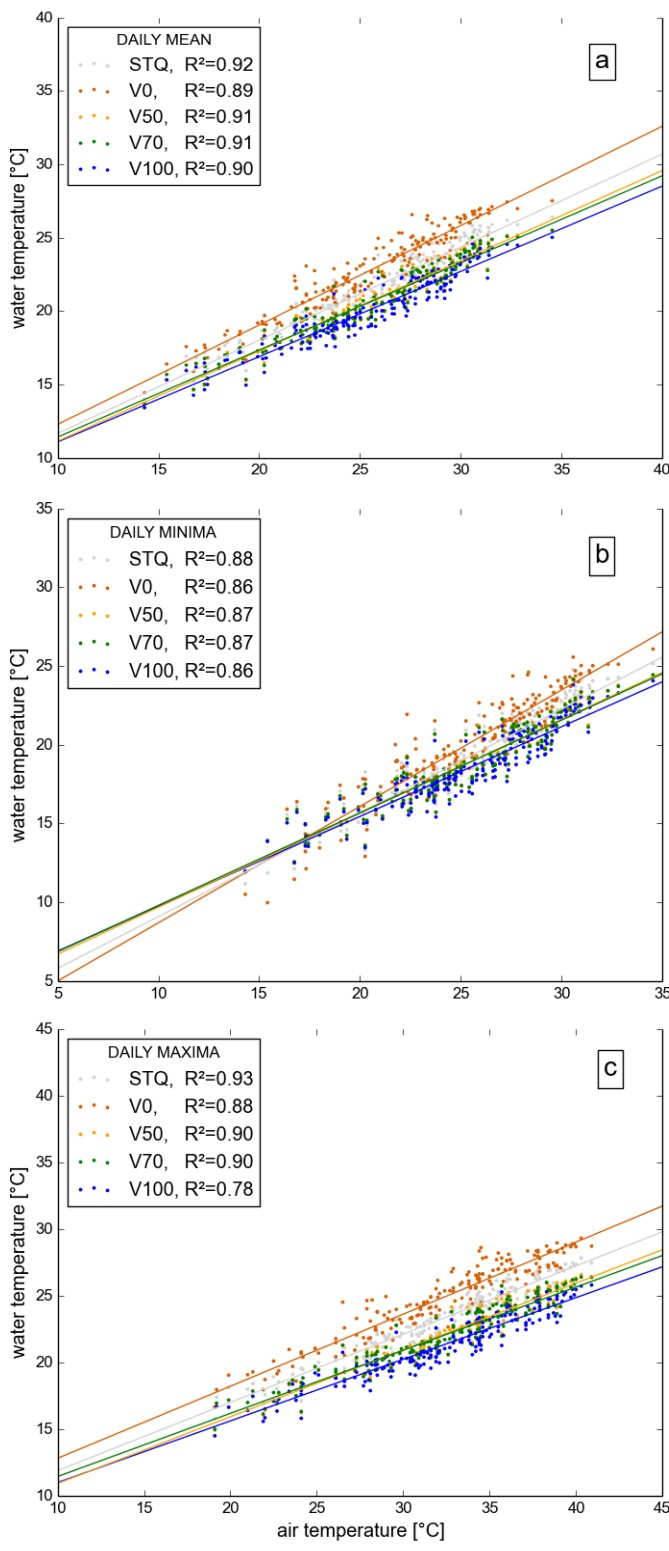

**Figure 8: Correlations between water temperature and the daily (a) mean, (b) minima and (c) maxima air temperatures for the 1a, 5a, 20a and Max episodes of the climate periods centred at 2030, 2050 and 2085 for existing vegetation (STQ), no vegetation (V0), vegetation height 50% (V50), vegetation of 70% vegetation density (V70) and full vegetation (V100) reported with the squared Spearman's rank correlation coefficient. ANCOVA showed significant interactions between vegetation and air temperature (p < 0.001).**