# Peer review of "Can riparian vegetation shade mitigate the expected rise in stream temperatures due to climate change during heat waves in a human impacted pre-alpine river?"

_Hydrology and Earth System Sciences, 2016_

## Referee Comment (RC1) · Anonymous Referee #1 · 28 Jul 2016

General comments:

This paper looks at different forest canopy (no vegetation, maximum vegetation and actual vegetation) on river water temperatures within a 49 km reach of the Pinka River (Austria). A 5-day heat wave period was studied under observed condition (2013), and under climate change (three different periods; 2016-2045, 2036-2065 and 2071-2100). In general, this paper was difficult to read and review because it lacked focus and a clear explanation of the research actually carried out. The objectives of study were not clearly presented; this was a climate change type study (and the title should have reflected this). A lot of results and discussion material focused around a model which was not described within the present study. As such, this aspect was not evaluated.

Nevertheless, authors presented R2, RMSE and model uncertainties, etc., which was somewhat confusing to the reviewer. The heat fluxes presented in this study were all positive, (section 3.1) which is clearly not the case in reality (not sure how authors could model river temperatures with such fluxes). The result section was difficult to follow, as authors presented both results from the present study and results from Trimmel et al. (2016). The text was almost presented as a part 2 of that paper, so reviewer was not always able to follow in the results presented were from this study or from the previous study. The results section also contained discussion material, and then separate discussion section was also presented. Finally, the reviewer is not sure of the scientific novel contribution which this present brings.

Specific comments:

Pg. 1, line 13-14: "and turbulent energy fluxes analysed". Not clear, something is missing here.

Pg. 1, line 14: "Minor stream water temperature increases are modelled within". Authors are presenting result in the present tense; it should be in the past tense. This applies throughout the document.

Pg. 1, line 14-15: "Minor stream water temperature increases are modelled within the first half of the century, but a more significant increase is predicted for the period 2071–2100". Sentence which is not saying anything, please be more specific.

Pg. 1, line 16: "to be in the region of 3 °C". In the range of 3°C?

Pg. 1, line 16: "Additional riparian vegetation". Not clear how this will be accomplished, regrowth, re-vegetation, etc., please clarify.

Pg. 2, line 4: "riparian ecosystems play a superior role in climate change". Riparian ecosystem plays a superior role in climate change to what?

Pg. 2, line 11: "21st century are nearly certain". Not sure about this level of certainty.

Pg. 2, line 19: "winter half-year". Not sure about the meaning of this term winter half-year, please clarify.

Pg. 2, line 23: "Long term increases of wind speeds or storm activity cannot be detected." Not clear.

Pg. 2, line 26: "dominant energy input causing diurnal fluctuations". Energy inputs are contributing to both diel and seasonal water temperature variability.

Pg. 3, line 1: "Since 1980 Austrian river temperatures have increased on average by 1.5 °C during". Here I would be more specific, one, XX or all Austrian rivers.

Pg. 3, line 6: "affect discharge volume and velocity". I would delete velocity, as it is implied.

Pg. 3, line 10-12: The information related to changes in sediment transport and climate change is not important, unless authors are implying that is has an impact on water temperatures, and clearly this study is not addressing this.

Pg. 3, line 33: "microthermal gradients in the river profile". Not clear.

Pg. 4, line 16-24: Too many vague statements within this paragraph. Be more specific, how many sites within the Pinka River, which regional climate scenarios?

Pg. 4, line 26-32: No need to describe the upcoming sections. Delete this whole section.

Pg. 5, line 9-10: "In this region the highest temperature increases and the largest precipitation reductions in Austria have been observed (Böhm et al. 2009)." Be more specific, by how much?

Pg. 5, line 30-33: "The average difference in stream temperature between no vegetation and maximum vegetation during the maximum heat wave of 2013 was calculated to be 3.81 °C by Trimmel et al. (2016).". Here the reviewer is confused, results from the present study are being reported or this analysis has been carried out before, not

clear.

Pg. 6, line 1-6: This information does not belong here. This information should have been presented in the introduction or in the discussion section.

Pg. 6, line 7-9: Is this what is new in the present study compared to Trimmel et al. (2016), i.e., studying a reach of 49 km rather 22.5 km?

Pg. 6, line 27-28: "These comparisons showed a high consistency, so the INCA data set was used". Vague statement, please be more specific and quantitative.

Pg. 7, line 11: "Stream temperature and flow volume were used as upstream boundary condition.". Authors should use the term discharge or river discharge, rather than flow volume.

Pg. 7, line 17-20: It would be better if authors would have presented root mean square errors (RMSE) rather the R2, or presenting both, as the R2 is not very informative on a model's performance. Also, not sure about the reported RMSE of 0.08 (°C? maybe). If it is an RMSE of 0.08°C, it does not fit with R2 values of 0.92 to 0.96.

Pg. 7, line 22: "The substrate temperature was initialized with the upstream model boundary temperature". Not clear about the substrate temperatures, where and at which depth?

Pg. 7, line 27: "Tributaries are defined by their water temperature and discharge values." Vague statement. Were they measured and then used in the model?

Pg. 7, line 28-29: Not exactly clear on what the boundary station means.

Pg. 8, line 9-10: Information presented within these two lines and related to the climate change aspect of this study should have been clearly stated in the introduction.

Pg. 8, line 26-27: "The most important influences of atmospheric energy fluxes and vegetation shade on stream temperatures are depicted in Fig. 2.". There is an issue with this figure, as three different vegetation scenarios were presented in Figure 2a

and only one heat flux scenario is presented in Figure 2b. Also, all heat fluxes presented in Figure 2b are positive, which is not possible. Generally, some fluxes will be positive (incoming shortwave radiation); however, other will be negative (longwave radiation/evaporative flux), while sensible heat, for instance, will be both positive and negative.

Pg. 9, line 1-11: All reported fluxes are positive within this paragraph (see comment above). How can authors have possibly fitted river temperatures, with such fluxes?

Pg. 9, line 13-14: "This leads to a rapid increase in the water temperature of the cool spring water." Authors do not have the data to support such statement.

Pg. 9, line 32-33: "Future boundary water temperature increases by the end of the century by up to 4.1 °C (Table 2)". Not clear.

Pg. 10, line 9-10: "The stream temperatures increase from the upstream model boundary at DFS 13 to DFS 62 during the 2013 heat wave event was about 7 °C (Fig. 2).". Was the water temperature increase due to tributary inflows (with different water temperatures) or due to the surrounding meteorological conditions (most likely tributary inflow)?

Pg. 10, line 14-15: Not sure why water temperature would drop from 25.0°C to 24.8°C (middle period) when the climate is warming from 22.4°C to 22.6°C.

Pg. 11, line 5: "additional vegetation becomes more distinct in the downstream sections". Not clear about additional vegetation, please clarify.

Pg. 11, line 12-32: This whole section on model uncertainties does not seem to belong in this paper. How can a reviewer assess a model uncertainties when no information was presented on the model?

Pg. 11, line 30-32: "overhang caused changes in water temperature of +/–0.40 °C, +0.44 /–0.46 °C and +0.01 /–0.05 °C respectively". It is at times difficult for the reviewer to understand which data come from the present study or Trimmel et al. (2016).

Authors should remember that this section is the results section and most of the information presented here seems to be discussion material.

Pg. 12, line 12-13: "As the air–water temperature difference – unlike the absolute temperature level – is not expected to increase, no increase in sensible heat flux can be predicted.". Not sure what authors mean, please clarify.

---

## Short Comment (SC1) · 30 Jul 2016

The authors thank the reviewer for the constructive and useful comments and for his valuable time spent reviewing this manuscript. All the comments will be addressed within a author comment.

However, we would like to directly comment on the reviewers remark concerning the positive sign of all energy fluxes in Figure 2B. In this figure the latent, sensible, short- and long wave energy flux averaged over the heat wave episode 4-8. August 2013 are shown. Extreme heat events as treated in this study are outlined by high minimum and maximum air temperatures. High minimum air temperatures limit radiative cooling at night, also higher air temperatures increase the sensible heat flux from the atmosphere

towards the river. Under these extreme conditions long wave radiation and sensible heat flux became positive on average. Evaporation was the only energy flux, which was negative on average. The intention of Figure 2B was to better compare the magnitude of the negative latent heat flux with the magnitude of the short wave radiation balance, the magnitude of the other energy fluxes and the view to sky. This is why the latent energy flux was multiplied with (-1) and a minus sign added in the legend. In the text the term "input" and "output" was used to indicate the positive or negative direction of the energy flux. The authors however understand that this representation of the energy fluxes is misleading and will clarify this aspect.

---

## Short Comment (SC2) · 4 Aug 2016

All comments of Referee#1 will be addressed within an author comment. Prior to that the authors would like to address the general comments regarding scientific novelty, aim and scope of this study. We agree that a more precise delimitation to other studies, presentation of the scientific novelty and a clear overview of this study would greatly facilitate readability. The aim and scope of this study were summed up at the end of the Introduction Pg.4, line 16-24, but this section will be enlarged and clarified.

We propose following alternative formulation:

[revised manuscript text omitted]

---

## Author Comment (AC1) · 21 Aug 2016

Dear Referee 1

The authors thank you for the constructive and useful comments and for your valuable time spent reviewing our manuscript "Can riparian vegetation shade mitigate the expected rise in stream temperatures during heat waves in a pre-alpine river?" by H. Trimmel, P. Weihs, H. Formayer, D. Leidinger and G. Kalny. You have been addressing many issues to greatly improve readability of the manuscript.

Below we address all your general and specific comments.

General remarks

| No. | Comment | Response |
|-----|---------|----------|
| 1+2 | In general, this paper was difficult to read and review because it lacked focus and clear explanation of the research actually carried out.

The objectives of the study were not clearly presented; | We agree that a more precise delimitation to other studies and a clear overview of this study would greatly facilitate readability. The aim and scope of this study were summed up at the end of the Introduction Pg.4, line 16-24, but this section will be enlarged and clarified.
We propose following alternative formulation (as already posted within our second short comment):

"During the project BIO_CLIC vegetation cover and river morphology was recorded continuously along the river, stream temperatures were recorded at 11 sites as well as main tributaries of the eastern Austrian river Pinka (Holzapfel and Rauch 2015, Holzapfel et al. 2015). This data was used to setup and validate the 1D energy balance and hydraulic model Heat Source (Boyd and Kasper 2003) for the river Pinka.
Further Heat Source was used to analyse the mean influence of different meteorological, hydrological and shading parameters during heat wave conditions along a 22.5 km long uniform reach. Existing vegetation was found to be responsible for 4 times as much influence on temperatures as topographic or bank shade on average (1.68°C). This was reported during a different article by Trimmel et al. 2016.

The aim of the present article is (1) to estimate the magnitude of stream temperature rise during extreme heat events caused by the expected rise in air temperature until the end of this century compared to the last observed period and (2) to investigate the ability of riparian vegetation to mitigate the expected water temperature rise.

In the present article stream temperature was simulated with the 1D energy balance and hydraulic model Heat Source (Boyd and Kasper 2003) for 49km along a diverse section including upstream forested regions and tributaries for each 500m along the river, which amounts to a total of 103 sites. First the longitudinal changes of energy fluxes were analysed during the maximum heat wave, which took place in eastern Austria during summer 2013. Future heat wave episodes, which are likely to occur during the climate periods 2016-2045, 2036-1065 and 2071-2100 in the study region, were selected. Regional climate scenarios, which have been produced within the ENSEMBLE project (Hewitt et al. 2004) were further processed and the meteorological data extracted. The future upstream model water temperature was simulated according the methodology of Caissie et al. (2001). Heat Source was used to simulate the stream temperature of the river Pinka for 12 future episodes and three vegetation scenarios." |

Boyd, M. and Kasper, B.: Analytical methods for dynamic open channel heat and mass transfer: Methodology for heat source model Version 7.0, available at: http://www.deq.state.or.us/wg/TMDLs/tools.htm, 2003.

Caissie, D., Nassir, E.-J and Mysore, G. S.: Modelling of maximum daily water temperatures in a small stream using air temperatures, J. Hydology, 251, 14–28, 2001.

Hewitt, C. D. and D. J. Griggs, 2004: Ensembles-based Predictions of Climate Changes and their Impacts. Eos, 85, p566

Holzapfel, G. and Rauch H.P.: Der Einfluss der Ufervegetation auf die Wassertemperatur der Lafnitz und Pinka, Mitteilungsblatt für die Mitglieder des Vereins für Ingenieurbiologie, Ingenieurbiologie: Neue Entwicklungen an Fließgewässern, Hängen und Böschungen, 1/2015, 4–10, 2015.

Holzapfel, G., Rauch, H.P., Weihs, P. and Trimmel H.: The interrelationship of riparian vegetation and water temperature demonstrated with field data measurements and analysis of the rivers Pinka and Lafnitz, in: Geophysical Research Abstracts, 17, EGU General Assembly, Vienna, 12–17 April 2015, 11653–11653, 2015.

Trimmel, H., Gangneux, C., Kalny, G. and Weihs P.: Application of the model 'Heat Source' to assess the influence of meteorological components on stream temperature and simulation accuracy under heat wave conditions, Meteorol. Z. 25/4, PrePub doi:10.1127/metz/2016/0695, 2016.

| | | |
|---|---|---|
| 3 | this was a climate change type study (and the title should have reflected this) | The formulation "expected rise in stream temperature" was chosen in the title and considered to point out that this study is coping with climate change. To make it clearer the title will be extended to: "Can riparian vegetation mitigate the expected rise in stream temperature due to climate change during heat waves in a pre-alpine river?" |
| 4 | A lot of results and discussion material focused around a model which was not described within the present study. As such, this aspect was not evaluated. Nevertheless, authors presented R2, RMSE and model uncertainties, etc., which was somewhat confusing to the reviewer. | The authors agree that the methods section is not well organized and difficult to follow. We suggest so include several subheadings and restructure some paragraphs to make it easier to read. We will take great care, that all aspects necessary to understand the results and discussion, if not mentioned yet will be included. An exhaustive description of Heat Source cannot be given though, because this is not the scope of the article. A short description will be included in the Introduction as well (→ see Response to general remarks, comment #1+2). |
| 5 | The heat fluxes presented in this study were all positive, (section 3.1) which is clearly not the case in reality (not sure how authors could model river temperatures with such fluxes) | On this issue we responded within our first short comment: "In this figure the latent, sensible, short- and long wave energy flux averaged over the heat wave episode 4-8. August 2013 are shown. Extreme heat events as treated in this study are outlined by high minimum and maximum air temperatures. High minimum air temperatures limit radiative cooling at night, also higher air temperatures increase the sensible heat flux from the atmosphere towards the river. Under these extreme conditions long wave radiation and sensible heat flux became positive on average. Evaporation was the only energy flux, which was negative on average. The intention of Figure 2B was to better compare the magnitude of the negative latent heat flux with the magnitude of the short wave radiation balance, the magnitude of the other energy fluxes and the view to sky. This is why the latent energy flux was multiplied with (-1) and a minus sign added in the legend. In the text the term "input" and "output" was used to indicate the positive or negative direction of the energy flux. The authors however understand that this representation of the energy fluxes is misleading and will clarify this aspect." |

| 6 | The result section was difficult to follow, as authors presented both results from the present study and results from Trimmel et al. (2016). The text was almost presented as a part 2 of that paper, so reviewer was not always able to follow in the results presented were from this study or from the previous study. | Trimmel et al. (2016) was never cited within the Results (Section 3).

Section 4 (starting at Pg 11. line 12-32) is a separate section treating the uncertainties of the results (which are the predicted stream temperatures) and was not intended to be a results section. We consider the issue of uncertainties important, because we aimed to forecast the water temperature as precise as possible. We didn't want to include it in the Discussion section itself, because it should help to better discuss the results. Seemingly this was misleading therefore this part will be shortened and integrated into the Introduction and Discussion. |
|---|---|---|
| 7 | The results section also contained discussion material, and then separate discussion section was also presented. | We agree that on Pg. 9 line 30 and 32 two explanatory sentences should be moved to the Discussion. Some sentences in Section 3.1 may read like a discussion, but they actually are descriptions of the facts visible in Figure 2. Figure 2 should be referenced more often in Section 3.1 In Section 3.1 three sentences were found on Pg 9, line 20-25 which could be moved to the Discussion. They were placed there intentionally, because as the reviewer stated, it is quite uncommon to see both the sensible as the long wave heat flux becomes positive. The authors considered it necessary to explain this phenomena directly next to results. If other reviewers also object this, than this part can be moved to the discussion. The same applies to Pg 11. line 5-8 in Section 3.3 were Figure 4 is explained. |
| 8 | Finally, the reviewer is not sure of the scientific novel contribution which this present brings | We agree that a clear presentation of the scientific novelty would enhance the article. We propose to insert following paragraph at Pg. 4, line 16 (as already posted within our second short comment):

"Many studies have already addressed the influence of riparian vegetation on stream water temperature using field measurements. Other studies cope with different methods to predict stream temperature and few try to answer the question on how climate change might increase stream water temperature. Mainly air temperature is used as a surrogate for stream temperature and energy flux variations at different river sections are not considered. One result or trend may however not be transferred from one river to other. Statements of the riparian vegetation's potential to mitigate influence of climate change are only reliably valid for a given type of stream and for a given climate zone. The novel aspect of the present study is to investigate the influence of climate change and of riparian vegetation on the same river and attempt to make a realistic forecast of the riparian vegetation's potential to mitigate climate change in a specific river." |

Specific comments:

| No. | Comment | Response |
|---|---|---|
| 9 | Pg. 1, line 13-14: "and turbulent energy fluxes analysed". Not clear, something is missing here. | Referee 1 is correct, here is one word missing. It should be "and turbulent energy fluxes **are** analysed" |
| 10 | Pg. 1, line 14: "Minor stream water temperature increases are modelled | This will be corrected. |

| | | |
|---|---|---|
| | within". Authors are presenting result in the present tense; it should be in the past tense. This applies throughout the document. | |
| 11 | Pg. 1, line 14-15: "Minor stream water temperature increases are modelled within the first half of the century, but a more significant increase is predicted for the period 2071–2100". Sentence which is not saying anything, please be more specific. | Alternative formulation - joint with the subsequent sentence on Pg 1. line 15: "Stream water temperature increases of less than 1.5°C were modelled within the first half of the century. For the period 2071-2100 a more significant increase of around 3°C in maximum, mean and minimum stream temperatures was predicted for a 20 year return period heat event." |
| 12 | Pg. 1, line 16: "to be in the region of 3 °C". In the range of 3 °C? | Yes |
| 13 | Pg. 1, line 16: "Additional riparian vegetation". Not clear how this will be accomplished, regrowth, re-vegetation, etc., please clarify. | In the present study it was not relevant whether riparian vegetation regrows or is planted. The aim was to predict the effect of a potential vegetation cover, which is not present now, but can possibly be accomplished. |
| 14 | Pg. 2, line 4: "riparian ecosystems play a superior role in climate change". Riparian ecosystem plays a superior role in climate change to what? | We agree, that the statement is vague. Alternative formulation to "Above that riparian ecosystem play a superior role in climate change adaptation in the 21st century": "Above that riparian ecosystems play a superior role in **determining the vulnerability of natural and human systems to** climate change in the 21st century (Capon et al. 2013)." |
| 15 | Pg. 2, line 11: "21st century are nearly certain". Not sure about this level of certainty. | Alternative formulation: "... increases of 3.5°C by the end of the 21st century **are expected in Austria (APCC 2014,** Gobiet et al. 2014)."

 "is expected" is the formulation used by Gobiet et al. (2014) and APCC (2014, p.84 - Figure 1.10). |
| 16 | Pg. 2, line 19: "winter half-year". Not sure about the meaning of this term winter half-year, please clarify. | In this context the winter half-year was defined as the period 1 October to 31 March. The full sentence in the manuscript is "The decrease has been observed in summer and winter half-year (Böhm et al. 2009, 2012)" though. Our suggestion is to simplify it to "summer and winter", and omit the term "half-year". |
| 17 | Pg. 2, line 23: "Long term increases of wind speeds or storm activity cannot be detected." Not clear. | Alternative formulation: "Various studies indicate that from observations no long term increase of wind speed or storm activity can be detected in Europe (e.g. Matulla et al. 2008). For the alpine region also no clear signs of increasing wind speed or extremes are projected for the future (Beniston et al. 2007)."

 Beniston, M., Stephenson, D.B., Christensen, O.B., Ferro, C.A.T., Frei, C., Goyette, S., Halsnaes, K., Holt, T., Jylhä, K., Koffi, B., Palutikof, J., Schöll, R., Semmler, T., Woth, K., 2007. Future extreme events in European climate: an exploration of regional climate model projections. Climatic Change 81, 71-95. doi:10.1007/s10584-006-9226-z

 Matulla, C., Schöner, W., Alexandersson, H., Storch, H., Wang, X.L., 2008. European storminess: late nineteenth century to present. |

| | | Climate Dynamics, 133-144. doi:10.1007/s00382-007-0333-y |
|---|---|---|
| 18 | Pg. 2, line 26: "dominant energy input causing diurnal fluctuations". Energy inputs are contributing to both diel and seasonal water temperature variability. | We agree, this is a good complement of the sentence. |
| 19 | Pg. 3, line 1: "Since 1980 Austrian river temperatures have increased on average by 1.5 °C during". Here I would be more specific, one, XX or all Austrian rivers. | Alternative formulation: "Since 1980 **all Austrian river measuring points recorded an increase of stream temperature. Averaged this amounted to an increase of** 1.5 °C during ... (APCC 2014 p. 417, BMLFUW 2011)." |
| 20 | Pg. 3, line 6: "affect discharge volume and velocity". I would delete velocity, as it is implied. | We agree. |
| 21 | Pg. 3, line 10-12: The information related to changes in sediment transport and climate change is not important, unless authors are implying that is has an impact on water temperatures, and clearly this study is not addressing this. | We do imply, that sediment transport changes impact water temperature, because they might alter bed conduction flow and flow velocity. Both parameters affect stream water temperature. We consider it important to list all influencing factors, even if this study was not taking into account all aspect in our calculations.

Alternative formulation: "Sediment changes might alter the bed conduction flow as well as flow velocity, **which can influence the magnitude and variability of stream temperature. In this article the focus was on only on the increase in air temperature caused by climate change**." |
| 22 | Pg. 3, line 33: "microthermal gradients in the river profile". Not clear. | Alternative formulation: "Apart from its influence on average stream temperature vegetation shade produces **highly spatial variable shade, which results in areas of different sun exposure and energy fluxes. These heterogeneity provides ecological niches which are important for different development stages of river fauna** (Clark et al. 1999)." |
| 23 | Pg. 4, line 16-24: Too many vague statements within this paragraph. Be more specific, how many sites within the Pinka River, which regional climate scenarios? | → see response to general remarks, comment #1+2 |
| 24 | Pg. 4, line 26-32: No need to describe the upcoming sections. Delete this whole section. | We thought this was requested by the journal within the "manuscript composition" guideline, but it can be removed. |
| 25 | Pg. 5, line 9-10: "In this region the highest temperature increases and the largest precipitation reductions in Austria have been observed (Böhm et al. 2009)." Be more specific, by how much? | Alternative formulation: "**Since 1880** in this region air temperature **rose by 2°C. Precipitation was reduced in the HISTAP region corresponding to our study region** by 10-15%, which is the largest reduction in precipitation in Austria (Auer et al. 2007, Böhm et al. 2009, Böhm et al. 2012)."

Auer, I., Böhm, R., Jurkovic, A., Lipa, W., Orlik, A., Potzmann, R., Schöner, W., Ungersböck, M., Matulla, C., Briffa, K., Jones, P., Efthymiadis, D., Brunetti, M., Nanni, T., Maugeri, M., Mercalli, L., Mestre, O., Moisselin, J.-M., Begert, M., Müller-Westermeier, G., Kveton, V., Bochnicek, O., Stastny, P., Lapin, M., Szalai, S., |

Szentimrey, T., Cegnar, T., Dolinar, M., Gajic-Capka, M., Zaninovic, K., Majstorovic, Z. and Nieplova, E.,. HISTALP—historical instrumental climatological surface time series of the Greater Alpine Region. International Journal of Climatology 27, 17–46. doi:10.1002/joc.1377, 2007.

Böhm, R.: Changes of regional climate variability in central Europe during the past 250 years, The European Physical Journal Plus, 127, doi:10.1140/epjp/i2012-12054-6, 2012.

Böhm, R., Auer, I., Schöner, W., Ganekind, M., Gruber, C., Jurkovic, A., Orlik, A. and Ungersböck, M.: Eine neue Webseite mit instrumentellen Qualitäts-Klimadaten für den Grossraum Alpen zurück bis 1760, Wiener Mitteilungen Band 216: Hochwässer: Bemessung, Risikoanalyse und Vorhersage, 2009.

| 26 | Pg. 5, line 30-33: "The average difference in stream temperature between no vegetation and maximum vegetation during the maximum heat wave of 2013 was calculated to be 3.81 ◦ C by Trimmel et al. (2016).". Here the reviewer is confused, results from the present study are being reported or this analysis has been carried out before, not clear. | We agree. This was originally meant to describe the vegetation scenarios, but the position of the sentence is misleading therefore the sentence should be omitted or moved to the Introduction.
 → see response to general remarks, comment #1+2 |
|---|---|---|
| 27 | Pg. 6, line 1-6: This information does not belong here. This information should have been presented in the introduction or in the discussion section. | We agree. This sentence was meant to describe the study region, but its position is misleading therefore the sentence should be moved to the Introduction. We also suggest to shorten it.
 → see response to general remarks, comment #1+2 |
| 28 | Pg. 6, line 7-9: Is this what is new in the present study compared to Trimmel et al. (2016), i.e., studying a reach of 49 km rather 22.5 km? | We agree. This sentence was meant to describe the study region, but its position is misleading therefore the sentence should be moved to the Introduction.
 → see response to general remarks, comment #1+2 |
| 29 | Pg. 6, line 27-28: "These comparisons showed a high consistency, so the INCA data set was used". Vague statement, please be more specific and quantitative. | Alternative formulation: "Since the local permanent meteorological stations of ZAMG were used to produce the gridded INCA data set, they are highly consistent. The comparison of the INCA data with the air temperature measured at our reference station close to the river showed a RMSE of 0.67°C and a R² of 0.99 for consecutive hourly measurements during summer half-year 2013 (1 Apr – 30 Sept). So the INCA data set was used as proxy to represent the local meteorological conditions within the catchment." |
| 30 | Pg. 7, line 11: "Stream temperature and flow volume were used as upstream boundary condition." Authors should use the term discharge or river discharge, rather than flow volume. | We agree. The term "discharge" will be used. |
| 31 | Pg. 7, line 17-20: It would be better if authors would have presented root mean square errors (RMSE) rather the R2, or presenting both, as the R2 is not very informative on a model's performance. Also, not sure about the | We agree, that these two sentences is not very useful. We also have to admit, that there was a typing error regarding the RMSE and deeply apologize for this. Apart from this the periods are confusing. We suggest following alternative formulation including corrected values:
 "Observed hourly water temperatures (12 537 values) over the |

| | reported RMSE of 0.08 ( ◦ C? maybe).

If it is an RMSE of 0.08 ◦ C, it does not fit with R2 values of 0.92 to 0.96. | period 7 July 2012 to 9 September 2014 were used to fit the model. The coefficient of determination R² between observed and predicted water temperature for this period was 0.96, the RMSE was 0.68 °C.  For the summer half-year 2013 (1 Apr – 30 Sept), the R² was 0.89, the RMSE was 0.80 °C. " |
|---|---|---|
| 32 | Pg. 7, line 22: "The substrate temperature was initialized with the upstream model boundary temperature". Not clear about the substrate temperatures, where and at  which depth? | Alternative formulation:  "Heat Source uses only one substrate temperature, which is representative for the whole sediment layer.  The depth of the sediment layer is set to 1m, which is corresponding to the available geological information of the river Pinka. The substrate temperature used in the model is set equal to the stream temperature at the uppermost model point. For each consecutive model point the substrate temperature is calculated depending on the local thermal conductivity, thermal diffusivity, layer depth, hyporheic exchange, the river morphological profile and the received solar radiation at the river bed. " |
| 33 | Pg. 7, line 27: "Tributaries are defined by their water temperature and discharge values." Vague statement. Were they measured and then used in the model? | Alternative formulation: "The discharge and water temperature of **the river Pinka at the upstream model boundary** and the main **tributaries** of the 2013 episode **were measured**. " |
| 34 | Pg. 7, line 28-29: Not exactly clear on what the boundary station means. | Alternative formulation : "The water temperature data of the remaining tributaries and their future values were synthesised using the daily fluctuations of  **the water temperature at the upstream model** boundary adding a fixed offset depending on the distance of the inflow to the **upstream model** boundary."

→ see Pg. 7, line 11 – 21, response to general remarks, comment #6 and comment #33 |
| 35 | Pg. 8, line 9-10: Information presented within these two lines and related to the climate change aspect of this study should have been clearly stated in the introduction. | The aim of the study was presented at the end of the Introduction on Pg. 4 line 22-24. But the reviewer is correct, that a restructuring of the end of the Introduction, where the scope of the study is described would greatly improve readability.
Alternative formulation: → see response to general remarks, comment #6 |
| 36 | Pg. 8, line 26-27: "The most important influences of atmospheric energy fluxes and  vegetation shade on stream temperatures are depicted in Fig. 2.". There is an issue  with this figure, as three different vegetation scenarios were presented in Figure 2a and only one heat flux scenario is presented in Figure 2b.
Also, all heat fluxes presented in Figure 2b are positive, which is not possible. Generally, some fluxes will be positive (incoming shortwave radiation); however, other will be negative (longwave radiation/evaporative flux), while sensible heat, for instance, will be both positive and negative. | The issue regarding the direction of the energy fluxes was responded during our first short comment/the response to the general remarks, comment #5. The caption of the figure could be extended with following text suggestion:
"Q_sw, Q_lw and H are directed towards the river column (positive direction),  LE is directed from the river column to the atmosphere (negative direction)."

Regarding the figure we agree, that the comparison of the energy fluxes of all vegetation scenarios are of interest and suggest to add another figure, which is included below, which shows the energy fluxes in separate subfigures for all three vegetation scenarios (Fig. X1).  Especially the change in sensible heat flux and long wave energy flux as a function of the distance to the river's source is very clear in the open sky scenario (V0) and damped by dense vegetation (V100).

Another figure showing a variant of Figure 2 including all vegetation scenarios and the latent heat flux in the original |

| | | |
|---|---|---|
| | | negative direction in one figure is included below (Fig. X2). The information about the two different evaporation methods is omitted in Fig. X2, which, we have to admit, is less important for the article than the comparison of the vegetation scenarios. |
| 37 | Pg. 9, line 1-11: All reported fluxes are positive within this paragraph (see comment above). How can authors have possibly fitted river temperatures, with such fluxes? | In this section we are distinguishing between "input" and "output". Outputs are always negative, while inputs are positive. But there will be a minus sign added to clarify this.
→ This issue was responded during our first short comment and in the response to the general remarks, comment #5. |
| 38 | Pg. 9, line 13-14: "This leads to a rapid increase in the water temperature of the cool spring water." Authors do not have the data to support such statement. | We do have measurement data, that fit the simulated data and show the same strong increase in water temperature close to the spring (Figure 2C - the measured data is plotted with an "x").

Alternative formulation: "This lead to a rapid increase in the water temperature of the cool spring water, **which is clearly seen in both measured an simulated data (Figure 2C)**." |
| 39 | Pg. 9, line 32-33: "Future boundary water temperature increases by the end of the century by up to 4.1 °C (Table 2)". Not clear. | Alternative formulation: "For the water temperature at the upstream model boundary an increases of 4.1°C for a 20 year return event of the 2085 in respect to 2013 was simulated (Table 2). |
| 40 | Pg. 10, line 9-10: "The stream temperatures increase from the upstream model boundary at DFS 13 to DFS 62 during the 2013 heat wave event was about 7 °C (Fig. 2).".
Was the water temperature increase due to tributary inflows (with different water temperatures) or due to the surrounding meteorological conditions (most likely tributary inflow)? | The increase was under the assumption of a realistic scenario, including all known parameters (tributaries, realistic vegetation, river gradient and morphology, meteorology,..). |
| 41 | Pg. 10, line 14-15: Not sure why water temperature would drop from 25.0 °C to 24.8 °C (middle period) when the climate is warming from 22.4 °C to 22.6 °C. | On Pg. 10, line 14-15 the mean and maximum water temperature of a 20 year return event and all analyzed future climate periods are presented. The corresponding values are found in Table 3. Mean air temperature was rising from 27.2°C to 28.4°C (Table 2). The climate episodes used in this study were selected using air temperature thresholds. As they simulate realistic potential episodes they differ in global radiation, wind speed and humidity (see Table 2). Lesser amount of global radiation sums can lead to lower stream temperature despite higher air temperature. Higher wind speeds triggers increased evaporation which might lead to higher energy output and lower stream temperature despite higher air temperature. This is stated in the discussion already (Pg 12, line 21-24).
While mean water temperatures don't react so strong, reduced global radiation and higher wind speeds have a stronger effect on i.e. the maximum stream temperature. The authors agree that the reaction of the maximum stream temperature should be pointed out in the discussion following Pg 12, line 24. i.e. "This was most evident in maximum water temperatures." |
| 42 | Pg. 11, line 5: "additional vegetation | Alternative formulation: |

| | becomes more distinct in the downstream sections". Not clear about additional vegetation, please clarify. | "Looking at the longitudinal distribution of water temperature along the river it can be seen that for the Pinka the benefit of **additional tree cover maximizing riparian shade** became more distinct in the downstream sections." |
|---|---|---|
| 43 | Pg. 11, line 12-32: This whole section on model uncertainties does not seem to belong in this paper. How can a reviewer assess a model uncertainties when no information was presented on the model? | → This issue is addressed in the response to general remarks, comment #6 |
| 44 | Pg. 11, line 30-32: "overhang caused changes in water temperature of +/– 0.40 ∘ C, +0.44 /–0.46 ∘ C and +0.01 /– 0.05 ∘ C respectively". It is at times difficult for the reviewer to understand which data come from the present study or Trimmel et al. (2016).

Authors should remember that this section is the results section and most of the information presented here seems to be discussion material. | This section was not intended to be a results section, it is a separate section treating the uncertainties. → This issue is addressed in the response to general remarks, comment #6. |
| 45 | Pg. 12, line 12-13: "As the air–water temperature difference – unlike the absolute temperature level – is not expected to increase, no increase in sensible heat flux can be predicted.". Not sure what authors mean, please clarify. | We agree that this formulation is confusing. We suggest to reformulate line 12-15: "Short term influences which act on the daily amplitude of the river's temperature are not expected to change in magnitude, therefore the ability of the vegetation to alter the stream's microclimate and water temperature is likely to remain the same." We suggest to restructure the Discussion section to treat the magnitude of stream temperature rise and vegetation influences in separate subsections. |

[Figure]

Figure X1: Comparison of short wave (Q_sw), long wave (Q_lw) radiation balance, latent (LW) and sensible (H) heat flux for the 4 – 8 August 2013 along the river Pinka for three vegetation scenarios no vegetation (V0), maximum vegetation(V100) and Status quo (STQ).

[Figure]

Figure X2 (Variation of Figure 2 including all vegetation scenarios but omitting the information about the two different evaporation calculation methods):

VTS levels, predicted energy fluxes: short wave (Q_sw), long wave (Q_lw) radiation balance, latent heat (LE) and sensible (H) heat flux and water temperature (WT) means for the heat wave episode of 4 - 8 August 2013 for no vegetation (V0), existing vegetation (STQ) and maximum vegetation (V100) scenario.

---

## Referee Comment (RC2) · Anonymous Referee #2 · 22 Aug 2016

**General comments**

During reading the paper it is difficult to keep the focus/aims of the paper in memory. The overall objective is formulated in the title but in the text the reader will find other (sub)aims at several positions in different sections. The differentiation between objectives and methods is not appropriate. It is hard to follow the central theme of the paper since the structure of the paper is a little bit confusing. The paper includes 4 tables and 4 figures with a lot of information content but this is not represented adequate in the text, especially in the sections "Results" and "Methods" more clear links between text and figures/tables would be helpful to understand the intention of the paper.

**Specific comments**

[Figure]

P 1, title: the overall objective is formulated here. See also p 4, lines 22ff "The aim of this study..." and p 8, lines 9ff " The focus of this study...." are mentioned in different sections. The reader should find aims and focus of a paper at the beginning of the text to keep the central theme in mind.

P 1, line 13: "and turbulent energy fluxes were analysed"

P 2, line 4: " play a superior role..." please clarify

P 2 line 19: "summer and winter half-year" please clarify- which months/from-to?

P 2, line 22: " autumn" please clarify- which months?

P 3, line 1: " Austrian river temperatures..." which rivers were regarded – discharge values or other meaningful parameters were helpful. Is the Daube representative for the study area of this paper? Why?

P 3, line 6: "indirect effects of climate change" – is it possible to quantify these uncertainties?

P 3, line 30: "energy loss" – is it possible to assess/quantify?

P 4, line 26: "Sections". An outline of different sections would be helpful at the beginning of the paper maybe combined with clearly formulated objectives and aims of the paper.

P 5, line 3: "river Pinka": which type of river represents Pinka compared to others in Austria?

P 5, line 9: "highest temperature increases and …...reductions..." - how much? Please clarify.

P 7, line 11: "flow volume" – discharge

P 7, line 26: "no deep groundwater influence" - means there is one? Significant/insignificant - please clarify

P 7, lines 27ff: "Tributaries...partly estimated...adding a fixed offset....was supplemented..." is this conform to the state of the art? Or part of model uncertainties?

P 8, line 9: "The focus..." see above

P 8, line 18: "no significant changes in vegetation cover as it was the case in other studies performed earlier in the year" – what does significant mean in this context?

P 9, lines 15-25: Are these lines part of the results or taken from literature (which one?) - please clarify and quantify the mentioned effects if possible.

P 9 line 33: "...up to 4.1 °C (Table 2): Why is "max" lower than "20a" (Table 2, P 21)?

P 11, line 8: "incoming solar radiation which"

P 11, line 10: A more detailed quantifiable description of figure 4 is desirable.

P 11, line 11: "Uncertainties..." Model uncertainties should be a section following section 2.3. In the Results-section uncertainties should be discussed referring to the relevance to quantifiable results and the author's conclusions. Discussions about the model should be conducted before.

P 12, line 12: "is not expected to increase..." – source? Please clarify.

P 12, line 26: "by other studies." Which studies? Please specify.

P 12, line 27: "For Austrian rivers..." which ones?

P 12, line 28: "An increase..." which scenarios were used? Is the Danube comparable with Pinka referring to the focus of this paper?

technical corrections P 2, line 18 and 20: 15 %

P 5, line 14: 0.46 ms-1

P 7, line 7: 50 m

---

## Author Response (AR1)

**Authors response:**

We thank both reviewers for their useful comments and feedback and for your valuable time spent reviewing our manuscript.  We gave careful consideration to all comments provided by the reviewers. Both reviewers commented that the objectives were not well presented and the structure of the paper difficult to follow.  We worked on a clearer overview and hope we could facilitate readability. We have now completed the revisions and are happy to provide a updated response to all comments. Below you find a summary of the most relevant changes made in the manuscript and a marked up version of the revised manuscript for your consideration.

**Response to Reviewer 1:**

Some responses to your general and specific comments we gave to you have changed during the preparation of the revised manuscript. These are marked yellow. Figure 2 including caption has changed as well.  Mostly the response is only clarified in correspondence with the revised version. We hope you recognize all changes as improvements.

General remarks

| No. | Comment | Response |
|---|---|---|
| 1+2 | In general, this paper was difficult to read and review because it lacked focus and clear explanation of the research actually carried out.

The objectives of the study were not clearly presented; | We agree and tried to improve this throughout the paper.

We left the aim of the present article at the end of the Introduction following a paragraph about the scientific novelty where from the objectives of the present article are derived and hope this makes the objectives  much clearer to the reader.

The delimitation between preliminary work and the present article is described in the Methods now.

"

Preliminary work has been done and published by Holzapfel and Rauch (2015), Holzapfel et al. (2015) and during a different article by Trimmel et al. (2016). Vegetation cover and river morphology was recorded continuously along the river, stream temperatures were recorded at 12 sites as well as main tributaries of the eastern Austrian river Pinka (Holzapfel and Rauch 2015, Holzapfel et al. 2015). This data was used to set up and validate the 1D energy balance and hydraulic model Heat Source (Boyd and Kasper 2003) for the river Pinka (Trimmel et al. 2016).
Further Heat Source was used to analyse the mean influence of different meteorological, hydrological and shading parameters during heat wave conditions along a 22.5 km long uniform reach (Trimmel at al. 2016). |

| | | In the present article stream temperature was simulated with the 1D energy balance and hydraulic model Heat Source (Boyd and Kasper 2003) for 51 km along a section including upstream forested regions and tributaries for each 500m along the river, which amounts to a total of 103 sites. First the longitudinal changes of energy fluxes were analysed during the maximum heat wave, which took place in eastern Austria during summer 2013. Future heat wave episodes, which are likely to occur during the climate periods 2016-2045, 2036-1065 and 2071-2100 in the study region, were selected. Regional climate scenarios, which have been produced within the ENSEMBLE project (Hewitt et al. 2004) were further processed and the meteorological data extracted. The future upstream model water temperature was simulated according the methodology of Caissie et al. (2001). Heat Source was used to simulate the stream temperature of the river Pinka for 12 future episodes and three vegetation scenarios." |
|---|---|---|
| 3 | this was a climate change type study (and the title should have reflected this) | We agree. To make it clearer the title is extended to: "Can riparian vegetation mitigate the expected rise in stream temperature due to climate change during heat waves in a pre-alpine river?" |
| 4 | A lot of results and discussion material focused around a model which was not described within the present study. As such, this aspect was not evaluated. Nevertheless, authors presented R2, RMSE and model uncertainties, etc., which was somewhat confusing to the reviewer. | The authors agree that the description of the model was not sufficient and the methods section is not well organized and difficult to follow. We included several subheadings and restructured some paragraphs to make it easier to read. We tried to include all aspects necessary to understand the results and discussion. An exhaustive description of Heat Source cannot be given though, because this is not the scope of the article. |
| 5 | The heat fluxes presented in this study were all positive, (section 3.1) which is clearly not the case in reality (not sure how authors could model river temperatures with such fluxes) | The authors fully agree that not all the energy fluxes are positive. There is a subsection called "Energy fluxes during heat waves" included in the discussion now.

On this issue we also responded within our first short comment: "In this figure the latent, sensible, short- and long wave energy flux averaged over the heat wave episode 4-8. August 2013 are shown. Extreme heat events as treated in this study are outlined by high minimum and maximum air temperatures. High minimum air |

| | | |
|---|---|---|
| | | temperatures limit radiative cooling at night, also higher air temperatures increase the sensible heat flux from the atmosphere towards the river. Under these extreme conditions long wave radiation and sensible heat flux became positive on average. Evaporation was the only energy flux, which was negative on average. The intention of Figure 2B was to better compare the magnitude of the negative latent heat flux with the magnitude of the short wave radiation balance, the magnitude of the other energy fluxes and the view to sky. This is why the latent energy flux was multiplied with (-1) and a minus sign added in the legend. In the text the term "input" and "output" was used to indicate the positive or negative direction of the energy flux. The authors however understand that this representation of the energy fluxes is misleading and will clarify this aspect." |
| 6 | The result section was difficult to follow, as authors presented both results from the present study and results from Trimmel et al. (2016). The text was almost presented as a part 2 of that paper, so reviewer was not always able to follow in the results presented were from this study or from the previous study. | We agree that the position of the section "Uncertainties" after the "Results" section was misleading. A subsection treating with the uncertainties was included at the end of the Methods section. A delimitation to preliminary work presented in a previous publication by Trimmel et al. 2016 is given in the Methods. |
| 7 | The results section also contained discussion material, and then separate discussion section was also presented. | Results are described more thoroughly now and discussion material was moved to the Discussion. |
| 8 | Finally, the reviewer is not sure of the scientific novel contribution which this present brings | We agree that a clear presentation of the scientific novelty would enhance the article. We propose to insert following paragraph following the description of the state of the art (as already posted within our second short comment): "Many studies have already addressed the influence of riparian vegetation on stream water temperature using field measurements. Other studies coped with different methods to predict stream |

temperature and few tried to answer the question on how climate change might increase stream water temperature. Mainly air temperature was used as a surrogate for stream temperature and energy flux variations at different river sections are not considered. One result or trend may however not be transferred from one river to other.  Statements of the riparian vegetation's potential to mitigate influence of climate change are only reliably valid for a given type of stream and for a given climate zone.  The novel aspect of the present study is to investigate the influence of climate change and of riparian vegetation on the same river and attempt to make a realistic forecast of the riparian vegetation's potential to mitigate climate change in a specific river."

Specific comments:

| No. | Comment | Response |
|---|---|---|
| 9 | Pg. 1, line 13-14: "and turbulent energy fluxes analysed". Not clear, something is missing here. | Referee 1 is correct, here is one word missing. It should be "and turbulent energy fluxes  **were** analysed" |
| 10 | Pg. 1, line 14: "Minor stream water temperature increases are modelled within". Authors are presenting result in the present tense; it should be in the past tense. This  applies throughout the document. | This will be corrected. |
| 11 | Pg. 1, line 14-15: "Minor stream water temperature increases are modelled within the first half of the century, but a more significant increase is predicted for the period 2071–2100". Sentence which is not saying anything, please be more specific. | Alternative formulation -  joint with the subsequent sentence on Pg 1. line 15:  "Stream water temperature increases of less than 1.5°C were modelled within the first half of the century. For the period 2071-2100 a more significant increase of around  3 °C in maximum, mean and minimum stream temperatures was predicted for a 20 year return period heat event." |
| 12 | Pg. 1, line 16: "to be in the region of 3 °C". In the range of 3 °C? | Yes. |
| 13 | Pg. 1, line 16: "Additional riparian vegetation". Not clear how this will be accomplished, regrowth, re-vegetation, etc., please clarify. | In the present study it was not relevant whether riparian vegetation regrows or is planted. The aim was to predict the effect of a potential vegetation cover, which is not present now, but can possibly be accomplished. |
| 14 | Pg. 2, line 4: "riparian ecosystems play a superior role in climate change". Riparian ecosystem plays a superior role in climate change to what? | We agree, that the statement is vague. Alternative formulation to "Above that riparian ecosystem play a superior role in climate change adaptation in the 21[st] century": "Above that riparian ecosystems play a superior role in **determining the vulnerability of natural and human systems to** climate change  in the 21st century (Capon et al. 2013)." |
| 15 | Pg. 2, line 11: "21st century are nearly certain". Not sure about this level of | Alternative formulation: "... increases of 3.5°C by the end of the 21st century **are expected in Austria (APCC 2014,** Gobiet et |

| | | al. 2014)." |
|---|---|---|
| | | "is expected" is the formulation used by Gobiet et al. (2014) and APCC (2014, p.84 - Figure 1.10). |
| 16 | Pg. 2, line 19: "winter half-year". Not sure about the meaning of this term winter half-year, please clarify. | "summer (Apr – Sept) and winter half-year (Oct – Mar)" |
| 17 | Pg. 2, line 23: "Long term increases of wind speeds or storm activity cannot be detected." Not clear. | Alternative formulation: "Various studies indicate that from observations no long term increase of wind speed or storm activity can be detected in Europe (e.g. Matulla et al. 2008). For the alpine region also no clear signs of increasing wind speed or extremes are projected for the future (Beniston et al. 2007)."

Beniston, M., Stephenson, D.B., Christensen, O.B., Ferro, C.A.T., Frei, C., Goyette, S., Halsnaes, K., Holt, T., Jylhä, K., Koffi, B., Palutikof, J., Schöll, R., Semmler, T., Woth, K., 2007. Future extreme events in European climate: an exploration of regional climate model projections. Climatic Change 81, 71-95. doi:10.1007/s10584-006-9226-z

Matulla, C., Schöner, W., Alexandersson, H., Storch, H., Wang, X.L., 2008. European storminess: late nineteenth century to present. Climate Dynamics, 133-144. doi:10.1007/s00382-007-0333-y |
| 18 | Pg. 2, line 26: "dominant energy input causing diurnal fluctuations". Energy inputs are contributing to both diel and seasonal water temperature variability. | We agree, this is a good complement of the sentence. |
| 19 | Pg. 3, line 1: "Since 1980 Austrian river temperatures have increased on average by 1.5 °C during". Here I would be more specific, one, XX or all Austrian rivers. | Alternative formulation: "stations of the Austrian hydrographic central office of different elevation, distance from source and catchment area recorded an increase of stream temperature. The data were elevation corrected using External Drift Top-Kringing (Skøien et al. 2006) and a mean trend calculated using the Mann-Kendall-Test (Burn and Hag Elnur, 2002) by BMLFUW (2011). A mean trend of 1.5 °C during summer (Jun - Aug) and 0.7 °C during winter (Dec - Feb) was calculated APCC 2014 p. 417, BMLFUW 2011)."

Skøien, J., Merz, R., and Blöschl, G., Top-Kriging – geostatistics on stream networks, Hydrolology and Earth System Sciences (HESS), 10, 277-287, 2006.

Burn, D.H., and Hag Elnur, M.A, Detection of hydrologic trends and variability, Journal of Hydrology 255, p107-122, 2002. |
| 20 | Pg. 3, line 6: "affect discharge volume and velocity". I would delete velocity, as it is implied. | We agree. |

| 21 | Pg. 3, line 10-12: The information related to changes in sediment transport and climate change is not important, unless authors are implying that is has an impact on water temperatures, and clearly this study is not addressing this. | We do imply, that sediment transport changes impact water temperature, because they might alter bed conduction flow and flow velocity. Both parameters affect stream water temperature. We consider it important to list all influencing factors, even if this study was not taking into account all aspect in our calculations, because they were not all relevant for our study region and period.
Alternative formulation: "Sediment changes might alter the bed conduction flow as well as flow velocity, which can influence the magnitude and variability of stream temperature. Artificial changes which deteriorate the situation are presently illegal in Austria as well (WRG 1959). … This article focused on only on the increase in air temperature caused by climate change." |
|---|---|---|
| 22 | Pg. 3, line 33: "microthermal gradients in the river profile". Not clear. | Alternative formulation: "Apart from its influence on average stream temperature vegetation produces highly spatial variable shade, which results in areas of different sun exposure and energy fluxes. These heterogeneity provides ecological niches which are important for different development stages of river fauna (Clark et al. 1999)." |
| 23 | Pg. 4, line 16-24: Too many vague statements within this paragraph. Be more specific, how many sites within the Pinka River, which regional climate scenarios? | → see response to general remarks, comment #1+2 |
| 24 | Pg. 4, line 26-32: No need to describe the upcoming sections. Delete this whole section. | We thought this was requested by the journal within the "manuscript composition" guideline, but it was removed and integrated in the Methods. |
| 25 | Pg. 5, line 9-10: "In this region the highest temperature increases and the largest precipitation reductions in Austria have been observed (Böhm et al. 2009)." Be more specific, by how much? | Alternative formulation: "In this region air temperature **rose by 2°C since 1880. Precipitation was reduced in the HISTALP region corresponding to our study region** by 10-15%, which is the largest reduction in precipitation in Austria (Auer et al. 2007, Böhm et al. 2009, Böhm et al. 2012)."

Auer, I., Böhm, R., Jurkovic, A., Lipa, W., Orlik, A., Potzmann, R., Schöner, W., Ungersböck, M., Matulla, C., Briffa, K., Jones, P., Efthymiadis, D., Brunetti, M., Nanni, T., Maugeri, M., Mercalli, L., Mestre, O., Moisselin, J.-M., Begert, M., Müller-Westermeier, G., Kveton, V., Bochnicek, O., Stastny, P., Lapin, M., Szalai, S., Szentimrey, T., Cegnar, T., Dolinar, M., Gajic-Capka, M., Zaninovic, K., Majstorovic, Z. and Nieplova, E.,. HISTALP—historical instrumental climatological surface time series of the Greater Alpine Region. International Journal of Climatology 27, 17–46. doi:10.1002/joc.1377, 2007.

Böhm, R.: Changes of regional climate variability in central Europe during the past 250 years, The European Physical Journal Plus, 127, doi:10.1140/epjp/i2012-12054-6, 2012.

Böhm, R., Auer, I., Schöner, W., Ganekind, M., Gruber, C., Jurkovic, A., Orlik, A. and Ungersböck, M.: Eine neue Webseite mit instrumentellen Qualitäts-Klimadaten für den Grossraum Alpen zurück bis 1760, Wiener Mitteilungen Band 216: Hochwässer: Bemessung, Risikoanalyse und Vorhersage, 2009. |
| 26 | Pg. 5, line 30-33: "The average | We agree. The sentence was omitted. |

| | | |
|---|---|---|
| | difference in stream temperature between no vegetation and maximum vegetation during the maximum heat wave of 2013 was calculated to be 3.81 ◦ C by Trimmel et al. (2016).". Here the reviewer is confused, results from the present study are being reported or this analysis has been carried out before, not clear. | |
| 27 | Pg. 6, line 1-6: This information does not belong here. This information should have been presented in the introduction or in the discussion section. | We agree. This sentence was meant to describe the study region, but its position is misleading therefore the sentence was shortened and moved to the Methods. |
| 28 | Pg. 6, line 7-9: Is this what is new in the present study compared to Trimmel et al. (2016), i.e., studying a reach of 49 km rather 22.5 km? | We agree. This sentence was meant to describe the study region, but its position is misleading therefore this information is included in the paragraph about preliminary studies and scope of the present article at the beginning of the Methods. |
| 29 | Pg. 6, line 27-28: "These comparisons showed a high consistency, so the INCA data set was used". Vague statement, please be more specific and quantitative. | Alternative formulation: "Since the local permanent meteorological stations of ZAMG were used to produce the gridded INCA data set, they are highly consistent. The comparison of the INCA data with the air temperature measured at our reference station close to the river showed a RMSE of 0.67°C and a $R^2$ of 0.99 for consecutive hourly measurements during summer half-year 2013 (1 Apr – 30 Sept). So the INCA data set was used as proxy to represent the local meteorological conditions within the catchment." |
| 30 | Pg. 7, line 11: "Stream temperature and flow volume were used as upstream boundary condition." Authors should use the term discharge or river discharge, rather than flow volume. | We agree. The term "discharge" will be used. |
| 31 | Pg. 7, line 17-20: It would be better if authors would have presented root mean square errors (RMSE) rather the R2, or presenting both, as the R2 is not very informative on a model's performance. Also, not sure about the reported RMSE of 0.08 ( ◦ C? maybe).

If it is an RMSE of 0.08 ◦ C, it does not fit with R2 values of 0.92 to 0.96. | We agree, that these two sentences are not very useful. We also have to admit, that there was a typing error regarding the RMSE and deeply apologize for this. Apart from this the periods are confusing. We suggest following alternative formulation including corrected values:
"Observed hourly water temperatures (12 537 values) over the period 7 July 2012 to 9 September 2014 were used to fit the model. The coefficient of determination $R^2$ between observed and predicted water temperature for this period was 0.96, the RMSE was 0.68 °C. For the summer half-year 2013 (1 Apr – 30 Sept), the $R^2$ was 0.89, the RMSE was 0.80 °C. " |
| 32 | Pg. 7, line 22: "The substrate temperature was initialized with the upstream model boundary temperature". Not clear about the substrate temperatures, where and at which depth? | Alternative formulation: "Heat Source uses only one substrate temperature, which is representative for the whole sediment layer. The depth of the sediment layer is set to 1m, which is corresponding to the available geological information of the river Pinka. The substrate temperature used in the model is set equal to the stream temperature at the uppermost model point. For each consecutive model point the substrate temperature is calculated depending on the local thermal conductivity, thermal diffusivity, layer depth, hyporheic exchange, the river |

| | | morphological profile and the received solar radiation at the river bed. " |
|---|---|---|
| 33 | Pg. 7, line 27: "Tributaries are defined by their water temperature and discharge values." Vague statement. Were they measured and then used in the model? | We hope the situation is clearer using this alternative formulation: "The discharge and water temperature of **the river Pinka at the upstream model boundary** and the main **5 tributaries** of the 2013 episode **were measured**. The remaining tributaries added less than 5 % discharge each. " |
| 34 | Pg. 7, line 28-29: Not exactly clear on what the boundary station means. | We hope the situation is clearer using this alternative formulation : "The water temperature data of the remaining tributaries and their future values were synthesised using the daily fluctuations of **the water temperature at the upstream model** boundary adding a fixed offset depending on the distance of the inflow to the **upstream model** boundary."

 → see comment #33 |
| 35 | Pg. 8, line 9-10: Information presented within these two lines and related to the climate change aspect of this study should have been clearly stated in the introduction. | The reviewer is correct. This information is included in the aims at the end of the Introduction. |
| 36 | Pg. 8, line 26-27: "The most important influences of atmospheric energy fluxes and vegetation shade on stream temperatures are depicted in Fig. 2.". There is an issue with this figure, as three different vegetation scenarios were presented in Figure 2a and only one heat flux scenario is presented in Figure 2b.
 Also, all heat fluxes presented in Figure 2b are positive, which is not possible. Generally, some fluxes will be positive (incoming shortwave radiation); however, other will be negative (longwave radiation/evaporative flux), while sensible heat, for instance, will be both positive and negative. | We agree.

 Regarding Figure 2 all energy fluxes and all vegetation scenarios were included in the revised version. Latent heat flux is depicted in the negative direction (directed from the river column to the atmosphere).
 The information about the two different evaporation methods is omitted in, which is less important for the article than the comparison of the vegetation scenarios. |
| 37 | Pg. 9, line 1-11: All reported fluxes are positive within this paragraph (see comment above). How can authors have possibly fitted river temperatures, | We certainly agree that not all energy fluxes are positive. In this section we are distinguishing between "input" and "output". Outputs are always negative, while inputs are positive. There was a minus sign added to clarify this. |

| | | |
|---|---|---|
| | with such fluxes? | |
| 38 | Pg. 9, line 13-14: "This leads to a rapid increase in the water temperature of the cool spring water." Authors do not have the data to support such statement. | We do have measurement data, that fit the simulated data and show the same strong increase in water temperature close to the spring (Figure 2c - the measured data is plotted with an "x").

 Alternative formulation: "This lead to a rapid increase in the water temperature of the cool spring water, **which is clearly seen in both measured an simulated data (Figure 2c)**." |
| 39 | Pg. 9, line 32-33: "Future boundary water temperature increases by the end of the century by up to 4.1 °C (Table 2)". Not clear. | Alternative formulation: "For the water temperature at the upstream model boundary an increases of 4.1°C for a 20 year return event of the 2085 in respect to 2013 was simulated (Table 2). |
| 40 | Pg. 10, line 9-10: "The stream temperatures increase from the upstream model boundary at DFS 13 to DFS 62 during the 2013 heat wave event was about 7 °C (Fig. 2).". Was the water temperature increase due to tributary inflows (with different water temperatures) or due to the surrounding meteorological conditions (most likely tributary inflow)? | The increase was under the assumption of a realistic scenario, including all known parameters (tributaries, realistic vegetation, river gradient and morphology, meteorology,..). The sentence was completed and moved below to the description of the longitudinal distribution of water temperature: "The stream temperature increase from the upstream model boundary at DFS 11 to DFS 62 during the 2013 heat wave event for the STQ scenario including all available information about the present state of the river was about 7°C (Fig. 2)." |
| 41 | Pg. 10, line 14-15: Not sure why water temperature would drop from 25.0 °C to 24.8 °C (middle period) when the climate is warming from 22.4 °C to 22.6 °C. | On Pg. 10, line 14-15 the mean and maximum water temperature of a 20 year return event and all analyzed future climate periods are presented. The corresponding values are found in Table 3. Mean air temperature was rising from 27.2°C to 28.4°C (Table 2). The climate episodes used in this study were selected using air temperature thresholds. As they simulate realistic potential episodes they differ in global radiation, wind speed and humidity (see Table 2). Lesser amount of global radiation sums can lead to lower stream temperature despite higher air temperature. Higher wind speeds triggers increased evaporation which might lead to higher energy output and lower stream temperature despite higher air temperature. This is stated in the discussion already (Pg 12, line 21-24).
 While mean water temperatures don't react so strong, reduced global radiation and higher wind speeds have a stronger effect on i.e. the maximum stream temperature. The authors agree that the reaction of the maximum stream temperature should be pointed out in the discussion following Pg 12, line 24. i.e. "This was most evident in maximum water temperatures." |
| 42 | Pg. 11, line 5: "additional vegetation becomes more distinct in the downstream sections". Not clear about additional vegetation, please clarify. | Alternative formulation:

 "Looking at the longitudinal distribution of water temperature along the river it can be seen that for the Pinka the benefit of **additional tree cover maximizing riparian shade** became more distinct in the downstream sections." |
| 43 | Pg. 11, line 12-32: This whole section on model uncertainties does not seem to belong in this paper. How can a reviewer assess a model uncertainties | We agree that the position of the chapter "Uncertainties" after the "Results" section was misleading. A subsection treating with uncertainties was included at the end of the Methods section. The model Heat Source is better described now in the Methods |

| | when no information was presented on the model? | section as well. |
|---|---|---|
| 44 | Pg. 11, line 30-32: "overhang caused changes in water temperature of +/– 0.40 ◦ C, +0.44 /–0.46 ◦ C and +0.01 /– 0.05 ◦ C respectively". It is at times difficult for the reviewer to understand which data come from the present study or Trimmel et al. (2016).

Authors should remember that this section is the results section and most of the information presented here seems to be discussion material. | This section was not intended to be a results section, it is a separate section treating the uncertainties. The uncertainties are moved as separate subsection to the Methods section now. |
| 45 | Pg. 12, line 12-13: "As the air–water temperature difference – unlike the absolute temperature level – is not expected to increase, no increase in sensible heat flux can be predicted.". Not sure what authors mean, please clarify. | We agree that this formulation is confusing. We restructured the Discussion section to treat the magnitude of stream temperature rise and vegetation influences in separate subsections. We reformulate the former Pg. 12, line 12-15: " The water temperature difference between full and no vegetation showed no clear trend for future conditions. This can be explained considering that global radiation - the main parameter, that is affected by riparian vegetation (Leach and Moore 2010, Li et al 2012) - is the main parameter that contributes to heating of the water column (Benyahya et al 2012, Hannah et al. 2008, Maheu et al. 2014) and is not expected to be affected by climate change (APCC 2014). Therefore the ability of the vegetation to alter the stream's microclimate and water temperature is likely to remain the same." |

[Figure]

Figure 2: Comparison of the calculated VTS levels, short wave (Q_sw), long wave (Q_lw) radiation balance, latent (LE) and sensible (H) heat flux and measured (measured) and simulated (WT) water temperature for the heat wave episode 4 – 8 August 2013 along the river Pinka for three vegetation scnearios. no vegetation (V0), existing vegetation (STQ) and maximum vegetation (V100).

**Response to Reviewer 2:**

Dear Referee 2

The authors thank you for the constructive and useful comments and for your valuable time spent reviewing our manuscript "Can riparian vegetation shade mitigate the expected rise in stream temperatures during heat waves in a pre-alpine river?" by H. Trimmel, P. Weihs, H. Formayer, D. Leidinger and G. Kalny. You have been addressing many issues to clarify and improve readability of the manuscript. Below we address all your general and specific comments.

General comments

| No. | Comment | Response |
|-----|---------|----------|
| 1 | During reading the paper it is difficult to keep the focus/aims of the paper in memory.

The overall objective is formulated in the title but in the text the reader will find other (sub)aims at several positions in different sections. The differentiation between objectives and methods is not appropriate. It is hard to follow the central theme of the paper since the structure of the paper is a little bit confusing. | We agree and worked on a clearer, more focused presentation of the aims.
The authors agree that the Methods section is not well organized and difficult to follow. We included several subheadings and restructured this section to make it easier to read. |
| 2 | The paper includes 4 tables and 4 figures with a lot of information content but this is not represented adequate in the text, especially in the sections "Results" and "Methods" more clear links between text and figures/tables would be helpful to understand the intention of the paper. | We agree. We tried to improve this in the revised version and added more links between text and figures/tables. |

Specific comments:

| No. | Comment | Response |
|-----|---------|----------|
| 3 | P 1, title: the overall objective is formulated here. See also p 4, lines 22ff "The aim of this study. . ." and p 8, lines 9ff " The focus of this study. . .." are mentioned in different sections. The reader should find aims and focus of a paper at the beginning of the text to keep the central theme in mind. | We worked to clarify the overall structure but still kept the aims of the study at the end of the Introduction because they are derived from the state of the art. If wished they can be moved to the beginning of the Introduction as well. |
| 4 | P 1, line 13: "and turbulent energy fluxes were analysed" | This was corrected. |
| 5 | P 2, line 4: " play a superior role. . ." please clarify | Alternative formulation:  "Above that riparian ecosystems play a superior role in determining the vulnerability of natural and human systems to climate change in the 21st century (Capon et al.  2013)." |
| 6 | P 2 line 19: "summer and winter half-year" please clarify- which months/from-to? | Alternative formulation: "summer (Apr – Sep) and winter half-year (Oct to Mar)" |

[revised manuscript text omitted]

| 14 | P 7, line 11: "flow volume" – discharge | We agree. The term "discharge" is used in the revised version. |
|---|---|---|
| 15 | P 7, line 26: "no deep groundwater influence" - means there is one? Significant/insignificant - please clarify | The groundwater influence of the Pinka in the study region is possible but unknown. During this article only simulations during low flow conditions were conducted. It it assumed, that during low flow conditions there is no influence of deep groundwater. We suggest to use a more direct formulation: "The sediment of this region is very inhomogeneous and the spatial distribution of the groundwater level is unknown (Pahr 1984). For low flow conditions it was assumed that there was no deep groundwater influence." |
| | | In the end of the description of the model Heat Source we suggest to include: "The measurements fitted the simulation very well (average hourly was RMSE 0.88 °C for all measurement stations) so we conclude that all assumption were good and the model fit to be used for predictions." |
| | | In the Discussion we suggest to add: "Ground water influence was unknown and no ground water influence was assumed in the model. Although the model performed good (RMSE 0.88) there might be some ground water influence between DFS 45 and 55 where the measurements lie below the simulation results." |
| 16 | P 7, lines 27ff: "Tributaries. . .partly estimated. . .adding a fixed offset. . ...was supplemented. . ." is this conform to the state of the art? Or part of model uncertainties? | Ideally the stream temperature and discharge of every single tributary should be measured. Practically this is very difficult for larger river sections. In our case the interpolated tributaries have less than 5 % of the discharge of the main river and are not influenced by tempered waste or cooling water. Thus we consider it part of the model uncertainties and state of the art at the same time. |
| 17 | P 8, line 9: "The focus. . ." see above | This sentence was integrated into the aims at the end of the Introduction. |
| 18 | P 8, line 18: "no significant changes in vegetation cover as it was the case in other studies performed earlier in the year" – what does significant mean in this context? | The authors admit that this sentence is confusing and suggest to omit it. To clarify the vegetation development stage we suggest to insert in the new Section 2.3.2 (Vegetation and morphology): "The riparian vegetation situation was taken after the phenological phase of leaf development was finished and leaves were already fully developed (Ellenberg 2012). |

| | | Ellenberg, H. and Leuschner, H: Vegetation Mitteleuropas mit den Alpen, 6.Auflage, Verlag Eugen Ulmer, Stuttgart, XXIV+1134pp, 2012. |
|---|---|---|
| 19 | P 9, lines 15-25: Are these lines part of the results or taken from literature (which one?) - please clarify and quantify the mentioned effects if possible. | This part was complemented with quantitative information and discussion material moved to the Discussion. |
| 20 | P 9 line 33: ". . .up to 4.1 ◦ C (Table 2): Why is "max" lower than "20a" (Table 2, P 21)? | The future climate episodes used in this study were selected using 5 day mean air temperature thresholds. As they simulate realistic potential episodes they differ in global radiation, wind speed and humidity (see Table 2). Lesser amount of global radiation sums, as it is the case during the Max event of 2085 can lead to lower stream temperature and lower maximum air temperature despite higher mean air temperature. In the revised version this is described more in detail in section 3.2:

"During the 20 year return event of 2085 on the other hand global radiation was higher than the Max event (20.9 MJ m$^{-2}$ d$^{-1}$) of this climate period (Table 2).
For the mean water temperature at the model boundary an increase of +4.1 °C for a 20 year return event of 2085 in respect to 2013 was simulated (Table 2). For the Max event of 2085, which had 2.2 MJ m$^{-2}$ d$^{-1}$ lower global radiation input a slightly lower temperature increase (+4.0 °C) was simulated (Table 3)." |
| 21 | P 11, line 8: "incoming solar radiation which" | This sentence is removed, because it is too imprecise. |
| 22 | P 11, line 10: A more detailed quantifiable description of figure 4 is desirable. | Alternative formulation: "Looking at the longitudinal distribution of water temperature along the river it can be seen that increases in mean stream temperature caused by increases of future air temperature affected all parts of the river (Fig. 4a-c).
The maximum values showed a similar distribution as the mean values on a higher level. The average difference between mean and maximum values of the STQ scenario was 3.92 °C, 3.35 °C and 3.91 °C, the maximum difference between maximum values was 5.51 °C, 4.89 °C and 5.51 °C and the standard deviation of this difference was 0.71, 0.66 and 0.71 for 2030, 2050 and 2085 respectively (Fig. 4a-c). V0 scenarios were always warmer than STQ scenarios, V100 scenarios were always cooler than the STQ scenarios. The mean difference along the river between V0 and STQ was 1.25 °C, 1.26 °C and 1.13 °C, the maximum difference was 1.81 °C, 1.85 °C and 1.66 °C, the standard deviation was 0.35, 0.36 and 0.32 for 2030, 2050 and 2085 respectively. The mean difference between STQ and V100 was 1.42 °C, 1.52 °C, and 1.26 °C, the maximum difference was 1.92 °C, 2.05 °C and 1.72 °C, the standard deviation of this difference was 0.46, 0.49 and 0.41 for 2030, 2050 and 2085 respectively (Fig. 4a-c).

Water temperature was especially sensitive to the removal of vegetation within the first 10 km (DFS 11 - 21) where there were dense forests which prevented the cool headwaters from warming (Fig. 4d). At DFS 11 - 21 temperatures increased by 1.4 °C when removal of vegetation is assumed (V0-STQ). Additional tree cover (V100) caused a reduction of -0.9 °C compared to the STQ scenario (Fig. 4d).
This can be explained by the slower flow velocities (last 30 km - DFS 32-62: 0.003 m m$^{-1}$, 0.4 m s$^{-1}$ ) in comparison to the steeper upstream |

| | | sections (first 10 km - DFS 11-21: 0.017 m m$^{-1}$, 0.6 m s$^{-1}$), which gave short wave radiation in unshaded sections more time to heat the water column.

For the Pinka the benefit of additional tree cover maximizing riparian shade became more distinct in the downstream sections (DFS 25-55) where the additional tree cover caused a change of 1.75°C while removal only caused a change of around 1.25°C (Fig. 4d)." |
|---|---|---|
| 23 | P 11, line 11: "Uncertainties. . ." Model uncertainties should be a section following section 2.3. In the Results-section uncertainties should be discussed referring to the relevance to quantifiable results and the author's conclusions. Discussions about the model should be conducted before. | This part was shortened and integrated into the Discussion and Methods, where a new subsection was created as suggested. Uncertainties relevant for the direct evaluation of results are kept in the Results section. |
| 24 | P 12, line 12: "is not expected to increase. . ." – source? Please clarify. | We agree that this formulation is confusing. We suggest to restructure the Discussion section to treat the magnitude of stream temperature rise and vegetation influences in separate subsections. We suggest following explanation: "The water temperature difference between full and no vegetation showed no clear trend for future conditions. This can be explained considering that global radiation - the main parameter, that is affected by riparian vegetation (Leach and Moore 2010, Li et al 2012) - is the main parameter that contributes to heating of the water column (Benyahya et al 2012, Hannah et al. 2008, Maheu et al. 2014) and is not expected to be affected by climate change (APCC 2014). Therefore the ability of the vegetation to alter the stream's microclimate and water temperature is likely to remain the same."

Leach JA, Moore RD (2010) Above-stream microclimate and stream surface energy exchanges in a wildfire-disturbed riparian zone. Hydrol Process n/a-n/a. doi: 10.1002/hyp.7639
Li G, Jackson CR, Kraseski KA (2012) Modeled riparian stream shading: Agreement with field measurements and sensitivity to riparian conditions. J Hydrol 428–429:142–151. doi: 10.1016/j.jhydrol.2012.01.032

Maheu A, Caissie D, St-Hilaire A, El-Jabi N (2014) River evaporation and corresponding heat fluxes in forested catchments, Hydrol Process 28:5725–5738. doi: 10.1002/hyp.10071 |
| 25 | P 12, line 26: "by other studies." Which studies? Please specify. | The sentence is changed to "The values predicted in this article were clearly above the model uncertainty and lie in the upper region of the values published by other studies (BMLFUW 2001, Dokulil 2013, Melcher et al. 2013, 2014)." and moved to the end of the new section 4.2 (Magnitudes of stream temperature rise) |
| 26 | P 12, line 27: "For Austrian rivers. . ." which ones? | "From 1980 to 2011 230 stations of the Austrian hydrographic central office of different elevation, distance from source and catchment area recorded an increase of stream temperature (BMLFUW 2011)." |
| 27 | P 12, line 28: "An increase. . ." | "Dokulil (2013) extrapolated the quadratic regression of the period |

| | | which scenarios were used? Is the Danube comparable with Pinka referring to the focus of this paper? | 1900-2006 of the river Danube near Vienna and predicted an increase of up to 3.2 °C by 2050 in respect to 1900 (0.21 °C / decade). Using linear regression the increase was only 2.3  (0.15 °C / decade), but using the linear trend beginning from 1970 the increase was 3.4° C (0.23 °C / decade). Due to the size of the river Danube daily amplitudes and extremes are not comparable to the Pinka, but trends in mean water temperature values are comparable though." |
|---|---|---|---|
| 28 | | technical corrections:

P 2, line 18 and 20: 15 %
P 5, line 14: 0.46 ms-1
P 7, line 7: 50 m | These have been corrected. |

**Summary of relevant changes made in the manuscript**

Section 1 Introduction was extended including a paragraph about scientific novelty.

Section 2 Methods: The delimitation to other studies (especially the other article by Trimmel et al. 2016) and the aims and scope of this study was given here, integrating the paragraph "sections" as well.  Subheadings were included and the subsections are rearranged. The description of the model Heat Source was extended. The begin of the study region was initially named "DFS 13" according to the location of the gauge Pinggau. Actually the model was initialized at DFS11, where water temperature was measured continuously during 2013. No tributaries entered the Pinka between DFS11 and DFS13, therefore the discharge was used from DFS 13. This is corrected in the text and in Figure 1.

Section 3 Results: Figure 2 was adapted to include all vegetation scenarios and heat fluxes. Generally figures and tables are described more quantitative and detailed. Figures and Tables are referenced more often.

Section 4 (Uncertainties in predicted stream temperature) was shortened, so redundant and not relevant sentences were deleted. We tried to include all aspects which were addressed during the article and moved this part to the Methods into a separate subsections, as recommended.

Section 5 Discussion: Subheadings were included to differentiate between change in stream temperature, influence of riparian vegetation cover on stream temperature and limitations. A discussion about energy fluxes was moved down from the results and included as subsection.

[revised manuscript text omitted]

---

## Author Response (AR2)

Review of "Can riparian vegetation shade mitigate the expected rise in stream temperatures due to climate change during heat waves in a pre-alpine river?"

Dear Prof. Ghadouani,

there were 5 major issues addressed by referee 3+4:

1) The question was risen whether a previously performed **sensitivity study might have been sufficient to answer the questions** asked in this manuscript. Unfortunately no. The sensitivity study done on single parameters is not able to predict the behaviour of a multiparameter model if a composition of parameters are changed. Each future episode varies not only in air temperature but also in global radiation, wind speed and air humidity. The consequence of different vegetation scenarios during future episodes was not predictable especially not in a quantitative way by using a simple sensitivity analysis. As we had the chance to revise the manuscript we could include some new results regarding diurnal variations and trends caused by vegetation during higher temperature level episodes.

2) The question was raised **whether climate change would cause changes in vegetation and feedback to water temperature which are not covered in the study yet**. As the river Pinka is only 4% fully natural there is only a very limited natural vegetation dynamic. Even if the species distribution is changed, this will have no foreseeable effect on the vegetation height and density. Nonetheless it is possible therefore two additional vegetation densities and one additional vegetation height were considered and shown in the revised version to be able to discuss this aspect. The outcome of this study is that even if a very high shading is assumed, which can be achieved by choosing species which are adapted to the current climate and dense plantation, the effects of riparian shade can not fully mitigate the effects of climate change.

3) There was mentioned that **discharge changes are not taken into account**. The discharge chosen is already a low flow scenario, which is the average of the daily discharges below the 5% percentile of the climate period 1981 – 2010. If the mean low flow is reduced by 15% this is a reduction of only 5% of the MQ, therefore we consider it more important whether there is low flow or not. Heat waves must not always coincide with low flow and it is difficult to predict the discharge level within a certain episode. To be able to discuss this aspect of discharge reduction on water temperatures we included a scenario of -15% of MLF discharge for the 20a 2085 climate episode. We did not include discharge issues originally, because the aim was to compare the effect of atmospheric influences on the energy balance at the river surface and its influence on water temperature to the present situation and not to compare the wide range of possible discharge situations, which would be a different topic.

4) The **distribution of percent shade, bankfull width** was asked to be described and was included together with the anthropogenic influence along the river in a new Figure. As the bankfull width only varies between 4 and 10 m this aspect was not considered so important by the authors previously.

5) It was surprising for Referee No3 to read that **a 100% removal of vegetation would have less of an effect on stream temperatures than an increase in air temperature.** This misunderstanding arises we think from the formulation we used. If we speak of removal of vegetation this is referenced to the STQ vegetation, which is not full vegetation. In many areas it is rather sparse. If we compare full (V100) and no vegetation (V0) the change is clearly greater than the change due to increase of temperature.

We addressed all general and specific comments below and the manuscript was proofread by a native speaker to improve the language.

Kind regards, the Authors

Andreas Melcher from the Instiute of Hydrology of our University was included as coauthor to our team.

Section 1 was shortened, and parts moved to the Discussion. The aims where reformulated including aspects of changing vegetation and interactions of vegetation and discharge.

Section 2 was extended including all formulas of the energy fluxes used in the manuscript. Section 2.4 strongly integrated in 2.2 and 4.3, as well as strongly reduced. The description of the present and future vegetation in the region (2.1), vegetation sampling and vegetation scenarios (2.3.2) was extended.

Section 3 was extended regarding diurnal variations and trends caused by vegetation during higher temperature level episodes. 5 additional vegetation scenarios and 1 additional discharge scenario were included.

Section 4.3 was extended including a discussion about vegetation and discharge feedback.

A list of abbreviations was included as in the Annex.

We are happy that finally we have been able to include the doi of the data underlying this study which has been published on the freshwater biodiversity data portal.
* * *
Response to Referee#3

Dear Referee#3, thank you very much for your valuable and precise comments!

General comments:

| No. | Comment | Response |
|-----|---------|----------|
| 1 | The authors appear to have responded to earlier reviewer suggestions. I find the paper fairly cohesive and understandable. The main message is that careful predictions made using the model Heat Source indicate that the river Pinka will likely warm as a consequence of global warming, and by the end of the century even full shading will be insufficient to prevent temperature increases during 20-year return events of even 2 °C. | We agree. |
| 2+3 | The two aspects that I struggled with most in the paper were understanding individual sentences (suggested edits included) and coming to grips with results that suggested a 100% removal of vegetation would have less of an effect on stream temperatures | The value 1.8°C refers to the removal of existing vegetation (STQ) of a river which is not densely vegetated in all parts. The average change from full shade (V100) to no vegetation (V0) amounts to 5.8°. (see also response to |

| | than an increase in air temperature due to climate change. This point would be clarified if there were some other variable (e.g., vegetation density, percent shade, etc.) that readers could use to better understand the available shade for the STQ runs.

Additionally, more information on the distribution of the bankfull width would be useful; if most of the river had 4m bankfull widths, I would expect that vegetation could feasibly grow to an extent that the entire stream could be shaded. If the majority of the stream had bankfull widths of 30 m, I would expect additions/removals of shade to have far less of an impact on stream temperatures. | comment 41 below).

Regarding bankfull width: The river is anthropogenically influenced most of the course. The maximum bankfull width reached is 10m. Maximum vegetation as defined in the V100 scenario shades the whole river.

Additional graphs including the changes in shading percentage (as a resultant of vegetation height, density, width and topography) and bankfull width (Figure 2) as requested. The VTS is moved to this Figure as well. Energy fluxes of different shading (Figure 7) and discharge (Figure 6) are included in the revised version. |
|---|---|---|

Specific comments:

| No. | Comment | Response |
|---|---|---|
| 4 | Page 1 Line 12: You use a passive voice in the first sentence. Start with "We simulated the influence…" | Changed accordingly in the manuscript |
| 5 | Page 1 Line 28: change to "the occurrence of many species" | Changed accordingly in the manuscript |
| 6 | Page 1 Line 30: provide a citation to support the "river continuum disruption" sentence | Citations added:

Bloisa, J. L., Williams, J. W., Fitzpatrick M. C., Jackson, S.T., and Ferrierd, S., Space can substitute for time in predicting climate-change effects on biodiversity. Proceedings of the National Academy of Sciences, 110, Nr. 23, p.9374-9379, 2013.

Matulla, C., Schmutz A., Melcher, A., Gerersdorfer, T., and Haas, P.: Assessing the impact of a downscaled climate change simulation on the fish fauna in an Inner-Alpine River, Int. J. Biometeorology., 52, 127-137, 2007. |
| 7 | Page 1 Line 31: Zoonoses are diseases that can be transmitted from animals to people. Is the statement here indicating that major fish kills | We apologize for this spelling mistake. We intended to write "zoocenosis". But as this in not a well-used term we exchanged |

| | | could result in disease transmission to people? Please clarify. | it to: "a disruption of animal communities" |
|---|---|---|---|
| 8 | Page 1 Line 33-34: This sentence is unclear. I cannot tell what it means. | As this sentence was also unclear for a previous reviewer we omit it. |
| 9 | Page 2 Line 9: change "temperatures" to "temperature" | Changed accordingly in the manuscript |
| 10 | Page 3 Line 9: change "neither groundwater" to "neither change in groundwater" | Changed accordingly in the manuscript |
| 11 | Page 3 Line 10: deleted "change" | Changed accordingly in the manuscript |
| 12 | Page 4 Line 7: change "these" to "this" | Changed accordingly in the manuscript, but the sentence moved to the section 4.3. |
| 13 | Page 5 Line 2: It is not clear on what preliminary work has been done. | Line 2 - 8 describing preliminary work is removed because the necessary aspects are described in the corresponding sections below and the focus should be on the present manuscript and not the previous work done. |
| 14 | Page 5 first paragraph: This paragraph needs to be revised. Try changing the sentences to an active voice. "Holzapfel et al. (2015) continuously recorded vegetation cover…" | See specific comment 13 |
| 15 | Page 5 Line 2: What is meant by "during a different article by Trimmell"? | See specific comment 13 |
| 16 | Page 5 Line 5: change to "these data were" | See specific comment 13 |
| 17 | Page 5 Line 7: change to "Heat Source was further used" | See specific comment 13 |
| 18 | Page 5 Line 8: What is meant by "uniform reach"? What aspects of it were uniform? In other portions of the manuscript the substrate is described as not being uniform, and the vegetation cover varies as well. Also, identify in this sentence that the Pinka is the target river. | The section was uniform terms of slope, bankfull width and discharge. Due to comments made by another reviewer the parts describing previous studies are shortened where not necessary and this part was removed. |
| 19 | Page 5 Line 27: What is the HISTALP? | HISTAP is the name of a project, which defined different regions in Austria which have distinct climate trends. As this |

| | | additional information is not necessary for the statement and can be derived from the citation the sentence is shortened to: "Precipitation was reduced in our study region by 10-15%,.. " |
|---|---|---|
| 20 | Page 6 Line 22: change "good" to "met" and after "fit" add "were appropriate" if that statement is still true. | Changed to "...we concluded that all assumptions were met and the model was appropriate to be used for predictions." |
| 21 | Page 7 Line 3: This sentence is awkward. I suggest changing it to: "… conditions at the reference station data were extracted from the regional…", add a comma after "Remo", and delete text after the closing parenthesis. Provide a citation for ECHAM 5, as it is not introduced before this point. | Citations for the global climate models were included in the manuscript. |
| 22 | Page 7 Line 5: change ", therefore" to "; therefore," | Changed accordingly in the manuscript |
| 23 | Page 7 Line 8: rephrase the statement "area encompassing the area under investigation". | "In a second step the data were spatially localized to a 1 km x 1 km grid encompassing the area under investigation using the Austrian INCA data set (Haiden et al. 2011)" |
| 24 | Page 7 Line 29: change "situation was taken" to "data were obtained". | Changed accordingly in the manuscript |
| 25 | Page 8 Line 9: change "which is corresponding" to "which corresponds" | Changed accordingly in the manuscript |
| 26 | Page 8 Line 16: change "were prevailing" to "prevailed" | Changed accordingly in the manuscript |
| 27 | Page 8 Line 19: What is meant by a "change" in discharge? Positive or negative change? The sentence indicates that any change of 0.1 $m^3/s$ will lead to an increase in stream temperature. Is this what is meant? Also, I am not convinced that the model is sensitive: a 0.1 $m^3/s$ change in discharge is a 55% increase or decrease for the upstream model boundary and still a sizable change for the downstream boundary (13%). Also, where did temperatures increase by | A decrease in discharge was meant and changed accordingly in the manuscript. A change of 0.01$m^3/s$ at the upstream model boundary was simulated with resulted in a 0.04°C increase on the average stream temperature during heat wave 2 – 8 August 2013 from DFS 26 to 48. 0.01$m^3/s$ was chosen because this was the acuracy of the gauge station. On Page 8 Line 19 the value was simply multiplied to indicate what a 4 fold |

| | | |
|---|---|---|
| | 0.4 C? Was this at the upstream boundary, downstream boundary, or at the station in the middle? | increase of stream temperature means for a higher change in discharge. The referee is correct that "very sensitive" is not correct in this context. Also it is misleading to compare m3/s with °C. Percentage values were added and admitted, that the model is not sensitive to discharge rates. In Figure 6 the effect of a discharge reduction of 15% is shown. |
| 28 | Page 8 Line 22: Please clarify what is meant by MLF. The statement "average discharge of all discharges below the 5% discharge" is not helpful. What are the time periods in question that are being used to make these assessments? Is this annual or on a daily basis? The word "were" on line 29 suggests that there are multiple MLF values that are being used. | MLF was defined as the average of all daily discharges below the 5% percentile discharge within the climate period 1981 - 2010.

On line 24 there was a spelling mistake. There is only one MLF. This and the definition was changed accordingly in the manuscript. |
| 29 | Page 9 Line 1: I am not clear on how the moving average was calculated. Over what timeframe? | The moving average is an average over 30 years which is moving. We changed it to "the moving average over a 30 year climate period " and hope it is clearer now. |
| 30 | Page 9 Line 5: Who measured the discharge and temperature during the 2013 episode? Was it the current set of authors? | The sentences was completed with: "... were measured during the 2013 episode in the field by the authors and by two permanent gauging stations." |
| 31 | Page 9 Line 6: Please clarify "boundary. adding" The sentence starting with "adding" is incomplete. | The "boundary. adding" was replaced by "with the addition of ", because this sentences were meant to belong together. |
| 32 | Page 9 Line 10: Change beginning of sentence to "As mentioned," | As reference to previous studies was removed were not necessary, the sensitivity analysis is treated here the first time so the sentence was changed accordingly in the manuscript. |
| 33 | Page 9 Line 13: "changes in water temperature": where along the river were these changes found? | " ... which caused changes in the average water temperature between DFS 26 and 48 during 2 – 8 August 2013 of … " |
| 34 | Page 9 Line 15: What is meant by "mere"? Does it mean that topographic shade contributes little to the temperature, or that it contributes more than might have been anticipated? | "mere" was meant to emphasise that this refers to the topographic shade only and not taking into account bank shade or vegetation shade. The word was omitted. |

| 35 | Page 10 Line 7: I believe this paragraph contains errors and can be cleaner. Should Fig. 2a actually be Fig 2b? Is "conduction" referring to 2f? On line 13 (should this be appended to the end of line 12?) should 2f be 2d? Finally, what is this paragraph referring to? STQ? Please clarify and check. | Thank you very much. Indeed there were some errors regarding the reference to the Figures as indicated. The paragraph has been checked and clarified. |
|----|----|----|
| 36 | Page 11 Line 33: Change "supposed" to "supposing" Make the same change in the next paragraph as well. | Thank you. The first paragraph was changed accordingly in the manuscript. For the second we used "Under conditions of maximum riverine vegetation ", to prevent repitition and hope this is ok for you. |
| 37 | Page 12 Line 5: add "for" between "fully the" | Changed accordingly in the manuscript |
| 38 | Page 12 Line 30: Why did the addition and removal of trees become roughly the same between distances 53 – 60? Why do we see the pattern mentioned? Is this due to a lack of trees along the Pinka between distances 25-53? | Yes, between 25-53 there are very few trees, therefore addition of trees has more effect than removal. Between 53 and 60 the STQ vegetation cover is balanced, so that both addition and removal have the same effect. The aspect is mentioned to show, that not in all sections removal or addition of vegetation has the same effect. |
| 39 | Page 13 Line 6: change "temperature difference" to "temperature the difference" | Changed accordingly in the manuscript |
| 40 | Page 13 Line 12: This sentence contains many qualifiers. It is difficult to understand. Can it be simplified? | We tried to improve readability by splitting the information (that was requested by previous reviewers).

"The modelled 20 year return period heat wave (20a) in the climate period 2071–2100 showed a +3.8 °C increase in air temperature with respect to the observed period.  Increases in maximum, mean and minimum stream temperatures of close to +3 °C with respect to the observed period were simulated for this episode." |
| 41 | Page 14 Line 11: This sentence relates to my general comments statement: why are these streams only warming by 1.8 C when all shade is removed? | 1) The value 1.8°C is averaged twice: First it is the average daily max of the 5 day period. Here the max increase of maximum stream temperatures is 3.7° |

| | | |
|---|---|---|
| | Yes, some studies (examining much shorter reaches) only see increases of this amount following complete canopy removal, but others see increases of even 10 C (Brown 1969 and Brown and Kryegier 1970). Again, this relates to understanding what the current shade levels are over the river. | (2085-1a, Table 4). These values are further averaged over all episodes

2) The value 1.8°C refers to the removal of vegetation of a river which is not densely vegetated in all parts. The average change from full shade to no vegetation amounts to 5.8°.

3) Brown and Krygier analysed streams of summer flow below 0.028 to 0.057m3/s, while here a river of 0.18 to 0.76m3/s analysed. Small rivers react much stronger to atmospheric influences.  Also the reduction of the absolute maxima and not the average daily maxima is over 10 °C. The changes in average daily summer maxima are one dimension smaller (0.4 – 2.8°F). The reach of Berry Creek described by Brown 1969 is comparable to the upper boundary of this study but the change is given between the beginning of the reach and the end of the 600m long reach. When  I understand the study correctly the 11°F change are not only to be accounted to the fact, that the reach is not shaded but also the the rise in temperature caused by the daily amplitude. |
| 42 | Page 14 Line 34: Change "good" to "well" | Changed accordingly in the manuscript |
| 43 | Page 15 Line 20: change "showed to aggravate" to "aggravated" or "was shown to aggravate" | This sentence was removed during the revision process to shorten the conclusion. |
* * *
Response to Referee#4

Dear Referee#4, thank you very much that you draw the attention closely to very important aspects which have not been addressed sufficiently. The aims were reformulated including aspects of changing vegetation, discharge and feedback mechanisms.

General Comments

| No. | Comment | Response |
|---|---|---|
| | | |

| 1 | This study evaluates the role of vegetation shading in mitigating the rise in stream temperature under climate change. The authors have evaluated 3 vegetation scenarios with varying degree of shading (zero, normal and maximum vegetation) using 1D energy balance and hydraulic model Heat Source (Boyd and Kasper 2003) for the river Pinka located in the eastern Austrian Alps. The Heat Source model was calibrated and validated by the lead author and results have already been published in the journal of Meteorologische Zeitschrift (Trimmel et al. 2016). Surprisingly, Trimmel et al. (2016) also evaluated the influence of shading using identical vegetation scenarios [no vegetation (V0), maximum vegetation (V100) and current condition (STQ)] along with few additional topographic shading scenarios [No topography (T0), no river bank (B0)]. The findings related to sensitivity of stream temperature to shading from the earlier paper have been summarized in section 2.4. | You are correct that too much emphasis was given on preliminary work. At many locations it is not necessary, because the manuscipt can stand on its own. We included 5 additional vegetation scenarios, analysis of amplitudes and trends. We integrated section 2.4 into the description of the model in section 2.2. |
|---|---|---|
| 2 | The authors argue and I quote "While in the previous study of Trimmel et al. (2016) only the propagation of uncertainties of input parameters on the mean stream temperature of a 22.5 km long reach during the heat episode of 2013 was analysed, here the longitudinal distribution of a more diverse section including the headwaters of the river Pinka was shown and discussed." While this is true and this paper does bring additional analysis in terms of future climate change scenarios, one may have to wonder on the novelty and scientific contribution of this paper. Can't we use the sensitivity results reported by Trimmel et al. (2016), also summarized in this paper on page 9 section 2.4, to infer the role of shading in mitigating future warming? | Unfortunately we cannot. The sensitivity analysis was performed only by changing a single value along and comparing it to a base case. This cannot be used to predict the behavior of a multiparameter problem of future episodes, which consist of the interplay of global radiation, air humidity, air temperature, wind speed and type of shade. |
| 3 | As for as mitigating the effects of future warming by shading is concern what is the mechanism of increased shading under warmer climate? How can we have maximum vegetation height and density, when air temperature increases under the climate change scenario used in this study and a constant value of discharge? What about the effects of increased riparian vegetation and air temperature on discharge? Even if you ignore the significant (10-15% as reported on page 5 | The issue of change in vegetation height and density under changing climate are adressed in chapter 2.1, 2.3.2. and 4.3. Two additional result subsections to look at the influence of different vegetation height and densities in terms of diurnal variations and trends.

The reviewer is correct that both discharge and reduced shade is an important issue, so we included a scenario of a 15% |

| | | section 2.1) changes in precipitation, vegetation and air temperature alone can modulate discharge and create a feedback with stream temperature. Even when considering the vegetation shading as end-member scenarios these feedback processes must be accounted and discussed. | discharge reduction in Figure 6 to be able to discuss this aspect.

Discharge changes were not included originally in the study, because our emphasis was on finding out more about the influence of shade itself. It is clear to us that discharge has a major effect on stream water temperature, but we intentionally left it out to reduce the variability in the episodes. |
|---|---|---|---|
| 4 | | Introduction is poorly re-written and can be condensed. Too much emphasis on discussing trends should be avoided. In the methodology section, the authors rely too much on readers' knowledge and reference to the earlier work. This paper should stand on its own. | Thank you very much for your feedback regarding the Introduction and Methodology. Some parts grew in length during the previous revision round, but we tried to move parts to the Discussion and shorten it without loosing too much content. |
| 5 | | The model used in this study should be clearly explained and well justified. | The Methods was extended to cover all energy balance components briefly. Honestly we were not sure how much information about the model is desired by the readers. Using this feedback we revised the sections. |
| 6 | | Information related to model calibration and validation should be reported as well. | The model was never calibrated by the authors. The information about validation was already included in the last version at Page 6, line 20-24. |
| 7 | | It is unclear for readers that if the authors calibrated the model or they used the calibrated model. | This aspect was described at the beginning of the Methods and in the subsection in "Modelling energ y balance and stream temperature along the river". We tried to better clarify it . |
| 8 | | How were the vegetation height and density sampled? | Section 2.3.2 Vegetation and morphology, where the sampling is described, was extended. |
| 9 | | The language of the manuscript is VERY poor and not suitable for a publication. The paragraphs lack gradual transition and often end with one sentence. | Thank you for this feedback, we will consider this and try improve the language of the manuscript as we would like for all to follow it easily. |
| 10 | | Stating how this manuscript is different from another is not a great way to start "Results" section. | To point out the distinction between preliminary work and this work was required during the last revision round, but as the sentence is placed wrong it is |

| | | omitted now. |
|----|----|----|
| 11 | Both results and discussion are very hard to follow, sorry | Much additional and quantitative information, that makes these sections difficult to read was requested during the last revision, so we have difficulties to remove them. |

Specific Comments

| No. | Comment | Response |
|----|----|----|
| 12 | Pg1, line 27: This sentence needs a reference, "Stream temperature and assemblages of fish and benthic invertebrates …". | Citations added:

Dossi, F., Leitner, P., Steindl, E. and Graf, W.: Der Einfluss der Wassertemperatur auf die benthische Evertebratenzönose in mittelgrossen Fliessgewässern am Beispiel der Flüsse Lafnitz und Pinka (Burgendland, Steiermark) in Österreich, Mitteilungsblatt für die Mitglieder des Vereins für Ingenieurbiologie, Ingenieurbiologie: Neue Entwicklungen an Fließgewässern, Hängen und Böschungen, 1/2015, 22–28, 2015.

Melcher, A., Pletterbauer, F., Guldenschuh, M., Rauch, P., Schaufler K., Seebacher, M. and Schmutz S.: Einfluss der Wassertemperatur auf die Habitatpräferenz von Fischen in mittelgroßen Flüssen, Mitteilungsblatt für die Mitglieder des Vereins für Ingenieurbiologie, Ingenieurbiologie: Neue Entwicklungen an Fließgewässern, Hängen und Böschungen, 1/2015, 15–21, 2015. |
| 13 | Pg2, lines 25-27: this sentence is too long. Please break it into two sentences. "While net short wave radiation …" | The sentence was split in two. |
| 14 | Pg2, lines 26-27: Change "air humidity" and "wind" to vapor pressure and wind speed, respectively. | Changed accordingly in the manuscript |
| 15 | Pg2, lines 28-30: Please reword and revise. | This sentence doesn't fit at this position. The shortened version of the Introduction doesn't include this sentence |
| 16 | Pg2, line 34: Move "Since 1980" to the end of the sentence. | Changed accordingly in the manuscript |
| 17 | Pg3, line 23: one sentence cannot be a paragraph. | This sentence is misleading, because in the climate episodes also global radiation, air humidity and wind speed were included therefore the sentence is omitted. |
| 18 | Pg4, line 6: Revise and reword this sentence. | The sentence was reworded and moved to the discussion.
"Apart from its influence on stream temperature vegetation can cast spatially differentiated shade, which results in areas of different sun exposure and energy balance." |

| 19 | Pg4, line 6: Again, one sentence cannot be a paragraph. | The sentence was moved down to the end of the chapter and included to the second to last paragraph. |
|---|---|---|
| 20 | Pg4, lines 18-20: adding discharge to a regression model may or may not increases the model performance. | "improves" was changed to "can improve" |
| 21 | Pg4, line 21-23: Again, this paragraph has only two sentences. | The sentence were included in the previous paragraph. |
| 22 | Pg5, line 27: "air temperature rose …" needs a reference. | Citation, which was already used at a different location added at this point: Auer et al. 2007. |
| 23 | Pg5, lines 31-33: This sentence is too long and vague. | The sentence was shortend to: "Using the deterministic model Heat Source version 9 (Boyd and Kasper, 2003; Garner 2007) the energy fluxes along the river, hydraulics and stream temperature were simulated along the River Pinka." Additional information can be found in the following section below. |
| 24 | Pg5, line 33: What do you mean by this sentence "Existing data sets and parameters obtained from Austrian authorities and the literature were completed with field surveys and measurements"? Who are Austrian authorities? Are the data sets publicly available? If so you need to provide the link. How did you complete it? | Changed to: "The generation of the input data sets is described in the following section 2.3." The responsible authorities were mentioned directly in the subsection treating with the kind of input data. Also the completion of the data is described there. We received the data directly from the authorities. |
| 25 | Pg8, line 17: What does DFS stand for? | Distance from source (DFS) was defined in section 2.1 (Page 5 line 23). There was a summary of the most frequent abbreviations included in the Annex. |
| 26 | Pg8, line 19: "A change in discharge of …" increase or decrease? | A decrease in discharge was meant and changed accordingly in the manuscript |
| 27 | Fig. 1: add geographic reference | Lat/Lon coordinates were added at the corners of the study region |
| 28 | Fig 2: very messy, legends on top of the lines | The figure was changed so no legends cover lines. |
| 29 | Fig. 3: Run statistical significance test and report results with the figure. Right now it is unclear whether STQ is significantly different from V0. | A two tailored paired students T test was run for the hourly values to determine whether the difference between STQ and V0 and STQ and V100 is significant. A p-value less than 0.0001 was received for each episode. |

**Can riparian vegetation shade mitigate the expected rise in stream temperatures due to climate change during heat waves in a pre-alpine river?**

Heidelinde Trimmel[1], Philipp Weihs[1], David Leidinger[1], Herbert Formayer[1], Gerda Kalny[2], Andreas Melcher[3]

[1]Institute of Meteorology, University of Natural Resources and Life Science (BOKU), Vienna, 1190, Vienna, Austria
[2]Institute of Soil Bioengineering and Landscape Construction (IBLB), University of Natural Resources and Life Science (BOKU), Vienna, 1190, Austria
[3]Institute of Hydrobiology and Aquatic Ecosystem Management (IHG), University of Natural Resources and Life Science (BOKU), Vienna, 1190, Vienna, Austria

*Correspondence to*: Heidelinde Trimmel (heidelinde.trimmel@boku.ac.at)

**Abstract.** Global warming has already affected European rivers and their aquatic biota, and climate models predict an increase of temperature in Central Europe over all seasons. We simulated Tthe influence of expected changes in heat wave intensity during the 21st century on waterthe temperatures of an pre-alpine Austrian riverare simulated and analysed the future mitigating effects of riparian vegetation shade on the radiant and turbulent energy fluxes using the deterministic model *Heat Source analysed were*. Modelled Sstream water temperature increaseds of less than 1.5 °C were modelled within the first half of the century. Until For the period 2071 2100 a more significant increase of around 3 °C in minimum, maximum and, mean and minimum stream temperature was predicted for a 20 year return period heat event. The result showed clearly that Additional riparian vegetation was not able to fully mitigate the predictedexpected temperature rise caused by climate change, but would be able toecould reduce maximum, mean and minimum stream water temperatures by 1 to 2 °C. The Rremoval of riparianexisting 
[revised manuscript text omitted]

~~(DFS 0-12.5)Close to the source Theists cons vegetation thecomposition ranges from commercial spruce of forests (Picea abies) close to the source which undergoes management. The middle and downstream In thethesection he t ,rivers of the and near-natural deciduous riparian vegetation -sections floodplain species of the region (typical sinclude with and willows (Salix sp.), alders Populus sppoplars (Alnus glutinosa and incana. .) and wild cherry (Prunus sp.)alder (Alnus glutinosa) ash (Fraxinus excelsior), , maples (Acer sp.),In the downstream (from DFS 34 to 61) 80% of the reach, riparian vegetation is reduced to (from DFS 34 to 61)riveronly with highly altered sections to decorative purposesfor lining the river course~~

*-sided sparse tree plantations or two- one the river coursee lining of e.g. maples (Acer sp.) or lime trees (Tilia sp.) growthhese areas are mowed on a regular basis to prevent scrub T.srees like a t.ACorylus avellanahazel (sh (Fraxinus excelsior), ), and wild cherry (Prunus avium .ra) and Elder (Sambucus niger) can be found along the whole river course.*

-

**Vegetation scenarios**

Overall, tChanges in vegetation height and density in floodplain forests in natural systems are mainly due to succession. dynamical changes are expected. will be regulated for the foreseeable future, noiver Pinka Rdecline of water table. As the the inundation periods and intensity, days since rain and including the hydrological regime via changes in riparian vegetation cover onclimate change effect of Primack (2000), Garssen et al. (2014), Rivaes et al. (2014) studied the

fringes accompanying the river, which are flooded at least annually.narrow in many areas reduced to ispotential natural floodplain forest present The tweention zone behind thesis in the transi be. The vegetationsvegetation dynamicriparian The river has been continuously straightened and regulated throughout the 20st century. Flood protection measures and land use pressure has further altered the river and The dominant tree species present along the River Pinka, *Salix alba, Alnus glutinosa* and *Fraxinus excelsior* have a European-wide distribution (San-Miguel-Ayanz et al. 2016) so they are likely to defend their habitat. Some autochthonous species (*Populus alba, Prunus avium, Salix caprea, Fraxinus excelsior, Carpinus betulus*) which were present in 2013 are favoured by warmer climates (Kiermeyer 1995, Roloff and Bärtels 2006). Non-native species like *Robinia pseudoacacia* and *Acer negundo* are already present in the study region and might enlarge their habitat at the expense of native species (Kiermeyer 1995, Roloff and Bärtels 2006). Changes in tree species in favour of warmth-loving plants from downstream regions of the Raab/Danube catchment are possible (Lexer et al. 2014). Generally changes are likely to be not only driven by climatic but also anthropogenic factors as plantation of foreign species, which is not foreseeable.

the Danube region (Birkel and Mayer 1992, Egger et al. esses, well known ind via terrestrialization proceteand hardwood wetland and a further change towards upland or zonal vegetation is expe

Taking into account all likely changes in tree species, no change in maximum vegetation height or density is predictable.

the maximum vegetation height of the riparian vegetation is not expected to change, ThereforePotential changes are most likely caused by different vegetation management strategies as intentional clearings, plantations or mowing. Four vegetation Mmanagement scenarios are chosen to estimate the impact of different levels of vegetation shade on future heat waves. This also makes it possible to quantify potential changes to warmth-loving species of reduced height and density. To estimate the influence of different shading elements the fThe fFollowing scenarios have been consideredwere used:

maximum vegetation cover (V100) andactual vegetation cover (STQ).intermediate vegetation (V0), no vegetation cover (V0), 
[revised manuscript text omitted]
 (- -21 W/m²), ), convection (+5.6) and long wave radiation (+3.7 ) (Fig.ure 7).  The difference between the two shading scenarios is less at the downstream site (hort wave: +7.5 , evaporation -10.4 , convection 0.2 , long wave 1.5 )(Figure 8). The shading affects the maximum as well as the minimum water temperature and leads to a reduction of the daily amplitude (Fig. 6 and 7). . An interesting aspect is that the peak of stream temperature occurs about 1h later when vegetation is included.  With a vegetation density reduction of 50% (VD50) the diurnal range and especially the maximum temperatures are further increased (Fig. 7). It is interesting to note, that halving vegetation height has a similar or less significant effect as reducing vegetation density by 20% (Fig. 7).

**Trends**

The trend lines where calculated by minimizing the square error. An ANCOVA (analysis of covariance) showed significant interactions between vegetation and air temperature ($p < 0.001$). The equal slope assumption failed, the equal variance test was passed. Mean, maximum and minimum stream temperatures increase as air temperature increases (Fig. 8). Under the assumption of full vegetation, the intercept of the regression line is lowest for the mean and maxima, while under the assumption of no vegetation it is lowest for the minima. The difference between the vegetation scenarios is greatest for the maxima and smallest for the minima. The slope on the other hand is smallest for the maxima and greatest for the minima. All scenarios and values show a squared Spearman's rank correlation coefficient between 0.78 and 0.93. For mean and maximum temperatures the trend line of V0 is steeper than V100 (17 %), which means, that supposing no vegetation the maximum temperatures could lead to increase at a higher rate. For the daily minima the difference in slope is even greater (30 %). The regression lines of the halved vegetation height scenario (V50) and the reduced vegetation density scenario

(V70) cross for minima, mean and maxima values. The change in slope though is small (3.6 %, 1.4 % and 5.8 % for the mean, minima and maxima respectively) and statistically not significant.

**4 Discussion**

**4.1 Energy fluxes during heat waves**

In the present article evaporative heat flux was responsible for 100 % of heat loss from river water on average. Short wave radiation balance, long wave radiation balance and sensible heat flux were 64 %, 11 % and 25 % of the total energy inputs respectively.

During summer periods of high air temperature the difference between air and water temperature increases, which can trigger intensified evaporative flux that cools the river, but can also cause sensible heat flux to heat the water column (Benyahya et al. 2012). Benyahya et al. 2012 found that evaporative heat flux accounted for 100 % of energy outputs during 7-23 June 2008 while short wave radiation balance, long wave radiation balance and sensible heat flux were 72.53 %, 24.05 % and 2.03 % of the energy input respectively.

**4.2 Magnitude of stream temperature rise**

orIn the present article fThe modelled 20 year return period heat wave (20a) in the climate period 2071–2100 showed a with +3.8 °C increase in air temperature in with respect to the observed period. and drge was assumed and values for DFS 39 extracteMLF discha.;
[revised manuscript text omitted]

---

## Author Response (AR3)

Review of "Can riparian vegetation shade mitigate the expected rise in stream temperatures due to climate change during heat waves in a human impacted pre-alpine river?"

Dear Prof. Ghadouani,

there were 2  general issues addressed by Referee#5:

1) state that the results of this study are derived from a highly altered stream system

In the title, abstract and the manuscript the fact that the study copes with a anthropogenically influenced river was added.

2) clarify how they constructed their vegetation scenarios and provide some means for readers to assess how realistically these scenarios could be realized.

In Section 2.3.2 the vegetation scenarios were described more clearly and density values added for the STQ scenario. Also in the conclusion information about the feasability of the scenarios was added.

These two points are the summary of the relevant changes made in the Manuscript. Additionally we addressed all general and specific comments below.

Kind regards,  the Authors
* * *
Response to Referee#5

Dear Referee#5, thank you very much for your valuable comments!

General comments:

| No. | Comment | Response |
|---|---|---|
| 1 | A central message for this paper, from the abstract, is that "[t]he result showed clearly that riparian vegetation was not able to fully mitigate the predicted temperature rise caused by climate change[.]" However, as I can discern from page 9 lines 15-16, the authors examined increased shade given no change in land use pressure. The river Pinka is highly altered, as the authors well document. Therefore, I believe that the authors need to highlight the caveat, possibly in the title and certainly in the abstract, that this stream is highly altered and therefore regardless of how vegetation develops | We agree that the altered state should be pointed out in title and abstract. We included in the abstract „water temperatures of a heavily impacted pre-alpine Austrian river" and „in a highly altered river system", also the titel was changed to: „Can riparian vegetation shade mitigate the expected rise in stream temperatures due to climate change during heat waves in a human impaced pre-alpine river?".

In this study the maximum vegetation scenario was meant to estimate the maximum influence that riparian vegetation can have on a river without river restoration in terms of restoring natural river banks and |

| | | |
|---|---|---|
| | or is enhanced where it currently exists, it cannot mitigate the effects of increased air temperature on stream temperature. I believe that highlighting this point will help readers understand the sort of riparian conditions under consideration here. The abstract does not indicate, except indirectly in the last sentence, the current state of the river or the limitations imposed on the V100 model. I suggest a rewording or re-examination of findings throughout the manuscript to ensure that readers understand that the results pertain to a highly altered system, unlikely to receive riparian restoration in developed areas. | allowing natural river dynamics was considered. Following this idea, not only existing forests and gardens were densified, also agricultural fields within the distance of 50m were allowed to become a riparian forest of maximum effectiveness regarding reduced water temperature warming. We have to apologize that „no relief of land use pressure" was written, which might have been misleading. What we meant, was that the river didn't get more space. This was pointed out in section 2.3.2. |
| | | Also the findings were reexamined. At section 4.3 the altered state of Pinka was already decribed: "As the River Pinka is anthropogenically influenced and will be regulated for the foreseeable future no dynamical changes and no natural succession dynamics are expected which could cause an extreme change in vegetation cover." |
| | | In Section 5 the altered state was included: "In this study the influence of expected changes in heat wave intensity during the 21st century on stream temperature in the rithron to upper potamal section of the *human impaced* eastern Austrian River Pinka were simulated." |
| | | No other places were found were it was fitting to include the regulated state of the river. |
| 2 | On page 9, please **clarify the vegetation scenarios**. The sentence on lines 8-10 seems critical, yet I cannot understand what it is trying to convey. "Potential changes are most likely caused by different vegetation management strategies as intentional clearings, plantations or mowing." Did the authors model a change in riparian characteristics by altering a mowing regime, the plantation characteristics or by allowing clearings to regrow? Or were these left unaltered, due to "no relief in land use pressure" (line 16)? In other words, what specifically was done to model V100? What areas were | We agree, that the vegetation scenarios can be clarified. |
| | | Following sentences were changed/included in Section 2.3.2: |
| | | ad **clarify vegetation scenarios/achievement of density**: „Potential changes can only be induced by different vegetation management strategies as intentional clearings, plantations or mowing." |
| | | „For V0 within a 50m buffer all vegetation parameters (vegetation height, density and overhang) were set to 0 so that no vegetation shading occurred. The V0 scenario corresponds to intentional |

left unaltered, and in what stream areas was vegetation enhanced?

Also, readers would benefit from some linkage between **vegetation density** values used in the different scenarios and observed vegetation density values. It would be useful to understand what the mean vegetation density value and range for STQ was, as well as the vegetation density value(s) at the upstream least-altered sites.

Similarly, please provide more information on the **plausibility of V100's 90%** vegetation density.

What would be required to **achieve** this level of **density**?

Is it a realistic **restoration target**?

clearings and mowing. "

„V100 was defined as: 30 m height, 8 m overhang and 90 % vegetation density within a 50m buffer which is representative for the most dense riparian forests of existing riparian vegetation (STQ) located in the Pinka catchment (Ledochowski 2014). The V100 scenario represented the maximum possible level of vegetation shade. It is achievable by suspension of clearing and mowing activities as well as additional plantations of local tree and shrub species, who grow to different heights and form a well structured shrub and tree layer. To maintain this scenario management measures like replacement plantatings and well-directed cuttings are necessary. "

ad **vegetation density**: „The average density including all land cover types was 66% (standard deviation = 17 %) and the average height was 9.4m. Only considering areas including trees larger than 15m height the average density rose to 76 % (standard deviation = 11 %), ranging from 2 to 90 %. At the sheltered headwaters (DFS 20) the vegetation density reached 0.89."

ad **plausibility** of 90%: As the vegetation density of 0.9 is presently ocurring at the present Pinka (see above) we consider its plausible to use this value for a maximum vegetation scenario.

ad **restoration target**:

It is certainly feasibly, only a matter of costs. In Section 5 we added:

"During this study no economic evaluation of the vegetation scenarios could be done. While maximum vegetation height and densities of 50 % can easily be reached without external efforts, this process can certainly be accelerated as well as high densities assured by planting additional trees. This comes at a certain cost, but it might be worth to invest. "

also at Section 5 was added:

„Even when maximum vegetation extent with maximum height and density including plantations and replacement plantings was

| | | assumed, the additional riparian vegetation was not able to fully mitigate the expected temperature rise caused by climate change." |
| --- | --- | --- |

Specific comments:

| No. | Comment | Response |
| --- | --- | --- |
| 4 | P.1 line 23: Not sure what is meant by "efficient." Replace with "effective." | Changed accordingly in the manuscript |
| 5 | P.3 line 28-29: citations needed. | The sentence was rephrased to "Particular statements about the riparian vegetation's potential to mitigate the influence of climate change are only reliably valid for a given type of stream with its unique combination of morphologic and hydrologic parameters, local climate (Sinokrot and Stefan 1993; Johnson 2003) and regional climate change (Johnson and Wilby 2015). Sinokrot, B.A. and Stefan, H.G: Stream Temperature Dynamics: Measurements and Modeling, Water Resour. Res., 29, 2299-2312, 1993. Johnson, M.F. and Wilby, R.L.: Seeing the landscape for the trees: metrics to guide riparian shade management in river catchments, Water Resour. Res., 51, 5, 3754-3769, doi:10.1002/2014WR016802, 2015. new in manuscript: Steel, E.A., Fullerton, A., Thermal Networks – Do You Really Mean It?, StreamNotes – The Technical Newsletter of the National Stream and Aquatic Ecology Center, United States Departement of Agriculture, Fort Collins, Colorado, November 2017. Johnson, S.: Stream temperature: scaling of observation and issues for modelling, Hydrol. Process., 17,497–499, 2003. |
| 6 | p.8 line 25: delete ")" | Changed accordingly in the manuscript |

| 7 | p.11 line 24: Not sure what "Fig. f2h" refers to | We apologize for this error. Changed to "Fig. 3g" |
|---|---|---|
| 8 | p.13 line 31: space between "panel" and "the" | Changed accordingly in the manuscript |
| 9 | p. 14 line 18: change "where" to "were" | Changed accordingly in the manuscript |
| 10 | p. 14 line 16: please clarify "maximum temperatures could lead to increase at a higher rate" | We agree that this sentence was confusing and shortened it to: "maximum temperatures will increase at a higher rate" |
| 11 | p. 15 line 14: change "For the V0 scenario low" to "For the V0 scenario relatively low" | Changed accordingly in the manuscript |
| 12 | p. 15 line 21: I think a ")" is missing | The missing ")" was added. |
| 13 | p. 17 line 1: what is meant by "go line"? | "go line" was replaced with "are in agreement" |
| 14 | p. 17 line 16: change "In cloud" to "Under cloudy" | Changed accordingly in the manuscript |

[revised manuscript text omitted]